# Two distinct regulatory pathways govern Cct2-Atg8 binding in the process of solid aggrephagy

Yuting Chen [1,8], Zhaojie Liu [2,8], Yi Zhang [1,8], Miao Ye [3], Yingcong Chen [1], Jianhua Gao [4], Juan Song [5], Huan Yang [1], Choufei Wu [5], Weijing Yao [1], Xue Bai [6], Mingzhu Fan [6], Shan Feng [6], Yigang Wang [3], Liqin Zhang [5], Liang Ge [7], Du Feng [2✉] & Cong Yi [1✉]

## Abstract

CCT2 serves as an aggrephagy receptor that plays a crucial role in the clearance of solid aggregates, yet the underlying molecular mechanisms by which CCT2 regulates solid aggrephagy are not fully understood. Here we report that the binding of Cct2 to Atg8 is governed by two distinct regulatory mechanisms: Atg1-mediated Cct2 phosphorylation and the interaction between Cct2 and Atg11. Atg1 phosphorylates Cct2 at Ser412 and Ser470, and disruption of these phosphorylation sites impairs solid aggrephagy by hindering Cct2-Atg8 binding. Additionally, we observe that Atg11, an adaptor protein involved in selective autophagy, directly associates with Cct2 through its CC4 domain. Deficiency in this interaction significantly weakens the association of Cct2 with Atg8. The requirement of Atg1-mediated Cct2 phosphorylation and of Atg11 for CCT2-LC3C binding and subsequent aggrephagy is conserved in mammalian cells. These findings provide insights into the crucial roles of Atg1-mediated Cct2 phosphorylation and Atg11-Cct2 binding as key mediators governing the interaction between Cct2 and Atg8 during the process of solid aggrephagy.

**Keywords** Solid Aggrephagy; Cct2-Atg8 Binding; Atg1; Atg11; Phosphorylation
**Subject Categories** Autophagy & Cell Death; Signal Transduction; Translation & Protein Quality

## Introduction

Protein aggregation is a pathological hallmark of numerous human neurodegenerative diseases, making the effective clearance of these aggregations critical for preventing and treating the occurrence and development of these diseases (Mathieu et al, 2020). Aggrephagy, a selective autophagic process, has been shown to play a vital role in mitigating protein aggregation (Lamark and Johansen, 2012). In our prior research, we identified CCT2 as a pivotal receptor for aggrephagy, responsible for clearing solid protein aggregates. CCT2 accomplishes this by directly interacting with Atg8/LC3C, with mutations in the AIM/LIR motif of CCT2 severely impairing this interaction and consequently blocking solid aggrephagy (Ma et al, 2022). Additionally, the accumulation of aggregation-prone proteins prompts CCT2 to transition from a chaperone subunit to an aggrephagy receptor by enhancing CCT2 monomer formation (Ma et al, 2022). Therefore, understanding the mechanisms underlying CCT2-mediated solid aggrephagy is of paramount significance, shedding light on the intricate molecular processes involved. This knowledge contributes to advancing our understanding of solid aggrephagy and its potential implications for neurodegenerative diseases.

Accumulating evidence highlights the critical role of selective autophagy dysfunction in the development of multiple human pathologies (Vargas et al, 2023). Selective autophagy, a receptor-dependent autophagic process, serves as a crucial mechanism for eliminating damaged organelle, protein aggregates, lipid droplets, and pathogenic invasive bacteria, playing an important role in maintaining cell homeostasis (Vargas et al, 2023). Receptors governing selective autophagy can be broadly categorized into two types: ubiquitin-binding receptors such as p62, NBR1, and Cue5, as well as cargo-localizing receptors like Atg19, Atg32, and FUNDC1(Mancias and Kimmelman, 2016). Among these, CCT2 stands out as a ubiquitin-independent cargo-localizing receptors responsible for solid aggregates. CCT2 can be divided into three domains: the equatorial domain, which contains the ATP-binding site; the apical domain, which is responsible for the recognition and binding of the substrate, and the intermediate region, which serves as a linker between the apical domain and equatorial domain (Cuellar et al, 2019). A defining feature of selective autophagy receptors is their direct binding capacity to Atg8/LC3, typically mediated by the receptor's AIM/LIR motif (Popelka and Klionsky, 2015). Additionally, in *Saccharomyces*

[1]Department of Biochemistry, and Department of Hepatobiliary and Pancreatic Surgery of the First Affiliated Hospital, Zhejiang University School of Medicine, Hangzhou, China. [2]Guangzhou Municipal and Guangdong Provincial Key Laboratory of Protein Modification and Degradation, Affiliated Cancer Hospital & Institute of Guangzhou Medical University, School of Basic Medical Sciences, Guangzhou Medical University, Guangzhou, China. [3]Xinyuan Institute of Medicine and Biotechnology, School of Life Sciences and Medicine, Zhejiang Sci-Tech University, Hangzhou, China. [4]School of Medical Technology, Jiangxi Medical College, Shangrao, China. [5]Key Laboratory of Vector Biology and Pathogen Control of Zhejiang Province, School of Life Sciences, Huzhou University, Huzhou, China. [6]Mass Spectrometry & Metabolomics Core Facility, Key Laboratory of Structural Biology of Zhejiang Province, Westlake University, Hangzhou, China. [7]State Key Laboratory of Membrane Biology, Tsinghua-Peking Center for Life Sciences, School of Life Sciences, Tsinghua University, Beijing, China. [8]These authors contributed equally as first authors: Yuting Chen, Zhaojie Liu, Yi Zhang. ✉E-mail: Fenglab@gzhmu.edu.cn; yiconglab@zju.edu.cn

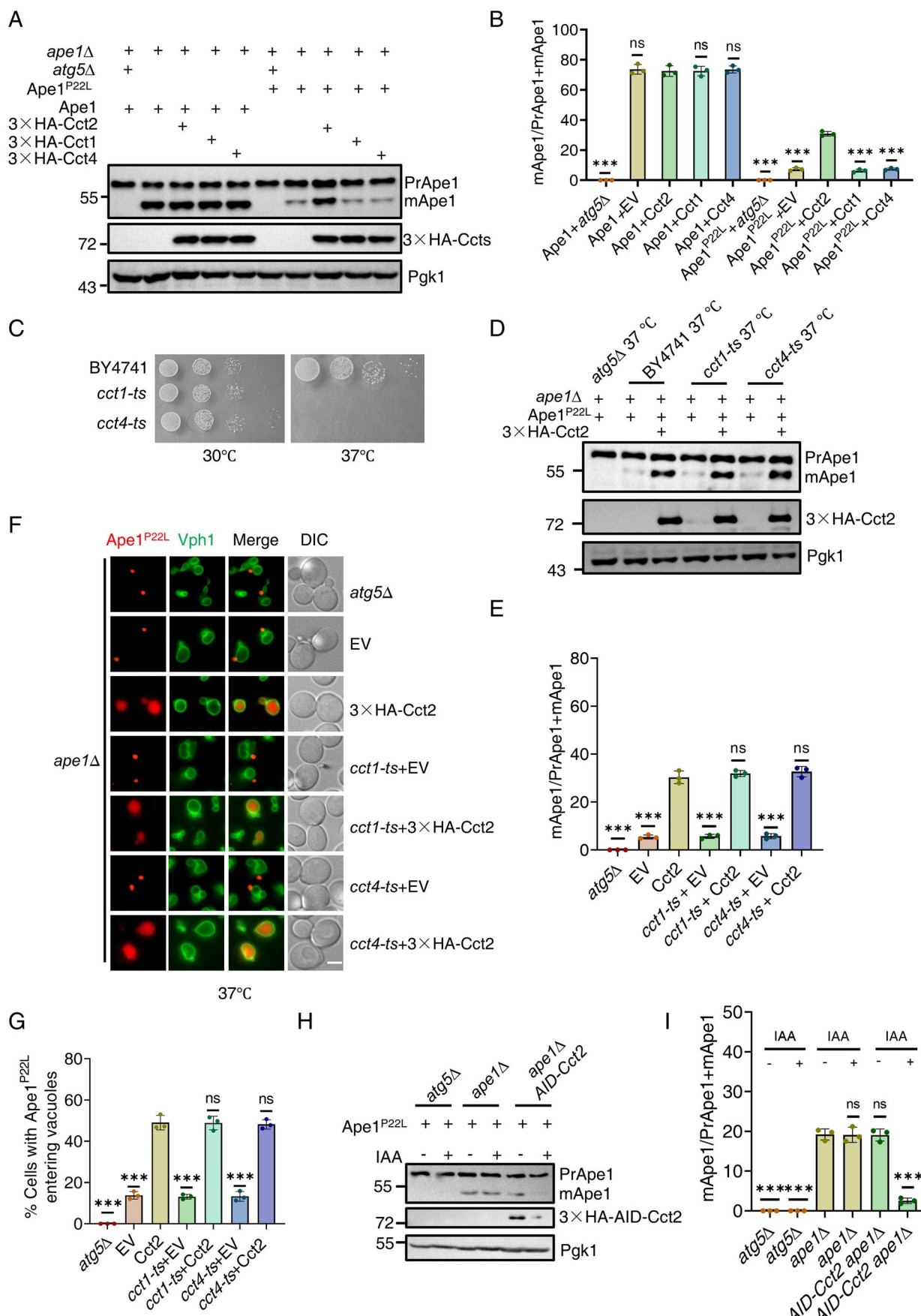

◀ **Figure 1. Cct2 functions as a solid aggrephagy receptor conserved in yeast.**

(A) *ape1Δ atg5Δ* or *ape1Δ* yeast cells co-expressing Ape1 WT or Ape1 P22L with empty vector (EV), 3×HA-Cct1, 3×HA-Cct2, or 3×HA-Cct4 were grown to log phase under nutrient-rich conditions. Samples were analyzed by immunoblot for detecting the maturation of PrApe1 into mApe1. Pgk1 served as a loading control. (B) Quantification of mApe1 to PrApe1+mApe1 ratio from (A). Data are presented as means ± SD (Data represent the results of three independent experiments). ***P < 0.001; NS not significant; two-tailed Student's *t* tests were used. P < 0.0001 (WT *atg5Δ* vs. WT Cct2); p = 0.7014 (WT EV vs. WT Cct2); P > 0.9999 (WT Cct1 vs. WT Cct2); P = 0.6838 (WT Cct4 vs. WT Cct2); P < 0.0001 (P22L *atg5Δ* vs. P22L Cct2); p = 0.0001 (P22L EV vs. P22L Cct2); P < 0.0001 (P22L Cct1 vs. P22L Cct2); P < 0.0001 (P22L Cct4 vs. P22L Cct2). (C) The indicated yeast strains were plated in 3-fold serial dilution onto YPD at 30 °C or 37 °C for 2 days. (D) *ape1Δ atg5Δ*, *ape1Δ*, *ape1Δ cct1-ts*, or *ape1Δ cct4-ts* yeast cells co-expressing Ape1 P22L with empty vector (EV) or 3×HA-Cct2 were grown to the early log phase at 30 °C and then subjected to 37 °C for 2 h. Samples were analyzed by immunoblot for detecting the maturation of PrApe1 into mApe1. Pgk1 served as a loading control. (E) Quantification of mApe1 to PrApe1+mApe1 ratio from (D). Data are presented as means ± SD (Data represent the results of three independent experiments). ***P < 0.001; NS not significant; two-tailed Student's *t* tests were used. P < 0.0001 (*atg5Δ* vs. Cct2); P = 0.0001 (EV vs. Cct2); P = 0.0001 (*cct1-ts* EV vs. Cct2); P = 0.3923 (*cct1-ts* WT vs. Cct2); P = 0.0001 (*cct4-ts* EV vs. Cct2); P = 0.2921 (*cct4-ts* WT vs. Cct2). (F) *ape1Δ atg5Δ*, *ape1Δ*, *ape1Δ cct1-ts*, or *ape1Δ cct4-ts* yeast cells co-expressing RFP-Ape1 P22L and Vph1-GFP with either an empty vector (EV) or 3×HA-Cct2 were grown to log phase at 30 °C. These yeast strains were then cultured at 37 °C for 2 h under full medium conditions. Images of cells were obtained using an inverted fluorescence microscope. Scale bar, 2 μm. (G) Cells from (F) were quantified for the vacuolar localization of RFP-Ape1 P22L. n = 300 cells were pooled from three independent experiments. Data are presented as means ± SD. ***P < 0.001; NS not significant; two-tailed Student's *t* tests were used. P < 0.0001 (*atg5Δ* vs. Cct2); P = 0.0001 (EV vs. Cct2); P < 0.0001 (*cct1-ts* EV vs. Cct2); P = 0.9552 (*cct1-ts* WT vs. Cct2); P = 0.0001 (*cct4-ts* EV vs. Cct2); P = 0.7581 (*cct4-ts* WT vs. Cct2). (H) *ape1Δ atg5Δ*, *ape1Δ*, or *ape1Δ* 3×HA-AID-Cct2 yeast cells were treated with 0.5 mM IAA for 0 h or 2 h. Samples were analyzed by immunoblot for detecting the maturation of PrApe1 into mApe1 and the levels of 3×HA-AID-Cct2 protein were detected by using Anti-HA antibody. Pgk1 served as a loading control. (I) Quantification of mApe1 to PrApe1+mApe1 ratio from (H). Data are presented as means ± SD (Data represent the results of three independent experiments). ***P < 0.001; NS not significant; two-tailed Student's *t* tests were used. P < 0.0001 (*atg5Δ* vs. *ape1Δ*); P < 0.0001 (*atg5Δ* + IAA vs. *ape1Δ*); P = 0.9397 (*ape1Δ* + IAA vs. *ape1Δ*); P = 0.9020 (AID-Cct2 vs. *ape1Δ*); P < 0.0001 (AID-Cct2 + IAA vs. *ape1Δ*). Source data are available online for this figure.

*cerevisiae*, these receptors exhibit another characteristic by directly interacting with Atg11 through the CC4 domain of Atg11 (Zientara-Rytter and Subramani, 2020). Recently, we found that both yeast and human CCT2 possess AIM/LIR motif within their apical domain, which are responsible for their interaction with Atg8/LC3C and the initiation of solid aggrephagy (Ma et al, 2022). Although the binding of CCT2 to Atg8/LC3C is crucial for solid aggrephagy, the regulatory mechanisms governing this interaction between CCT2 and Atg8/LC3C require further investigation in both yeast and mammals.

Over 40 ATG proteins have been identified to be involved in different types of autophagy in yeast (Fukuda et al, 2020). Among these, Atg1 plays a pivotal role as a protein kinase essential for both selective and non-selective autophagy (Ohsumi, 2014). In non-selective autophagy, the phosphorylation of Atg9 by Atg1 is required for the recruitment of Atg8 and Atg18 to the phagophore assembly site (PAS) (Papinski et al, 2014). Atg1 inhibits the interaction between Atg4 and Atg8-PE by phosphorylating Atg4, thereby suppressing the protease activity of Atg4 (Sanchez-Wandelmer et al, 2017). Recently, Atg1 were found to be required for the dynamic regulation of the PAS through phosphorylating Vps34 (Lee et al, 2023). In mammals, ULK1 activates Vps34 lipid kinase by phosphorylating Beclin1 and Atg14 (Russell et al, 2013; Wold et al, 2016). More recently, Stx17 was found to be phosphorylated by ULK1 to control the autophagosome maturation via FLNA (Wang et al, 2023). In selective autophagy, ULK1 has been reported to be involved in xenophagy by targeting ATG16L for phosphorylation (Alsaadi et al, 2019). p62 was found to be phosphorylated by ULK1 to promote selective autophagy under proteotoxic stress conditions via regulating its binding to ubiquitinated proteins (Lim et al, 2015). However, the role and mechanism of the kinase activity of Atg1/ULK1 in CCT2-mediated solid aggrephagy remains unknown.

In this study, we revealed that Cct2 is a new substrate of Atg1 and directly interacts with Atg11. Our research findings indicate that Atg1-mediated Cct2 phosphorylation and the binding of Atg11 to Cct2 are positively correlated with the interaction between Cct2 and Atg8. Importantly, this mechanism is also conserved in

mammalian cells. Given the crucial role of the binding of CCT2 with Atg8/LC3C in the aggrephagy, these findings raise the possibility that screening for regulators or drugs that affect the interaction between CCT2 and Atg8/LC3C could potentially be used to modulate the activity of solid aggrephagy.

# Results

## Cct2 functions as a solid aggrephagy receptor conserved in yeast

Our previous research revealed that yeast Cct2 directly interacts with Atg8 through its AIM motif and that the overexpression of Cct2 can enhance the processing of solid aggrephagy (Ma et al, 2022). Given that Cct2 is a subunit of the TRiC complex, we first explored whether the overexpression of other TRiC complex subunits, such as Cct1 and Cct4, could similarly promote the maturation of Ape1-P22L. Western blot results showed that while the overexpression of Cct2 promoted Ape1-P22L maturation, the overexpression of Cct1 and Cct4 did not have the same effect (Fig.1A,B). This suggests that specifically, Cct2 overexpression among the TRiC complex subunits promotes the maturation of Ape1-P22L.

Next, we explored whether Cct2 promotes Ape1-P22L maturation independently of the TRiC complex function. To this end, we introduced an overexpressed Cct2 plasmid into temperature-sensitive Cct1 (*cct1-ts*) and Cct4 (*cct4-ts*) yeast strains (Ben-Aroya et al, 2008). As shown in Fig. 1C, *cct1-ts* and *cct4-ts* yeast strains exhibit growth patterns similar to the wild-type yeast strain BY4741 at 30 °C but are lethal at 37 °C. Furthermore, cct1-ts and cct4-ts mutants disrupt the integrity of the TRiC complex, which was confirmed by a reduction of α-tubulin at 37 °C (Fig. EV1A) (Ma et al, 2022; Grantham et al, 2006). Subsequent Western blot assays demonstrated that overexpressed Cct2 still promotes Ape1-P22L maturation when Cct1 and Cct4 are inactive (Fig. 1D,E). Consistently, imaging data and statistical analysis indicate that over-expressed Cct2 facilitates the vacuolar localization of Ape1-P22L in

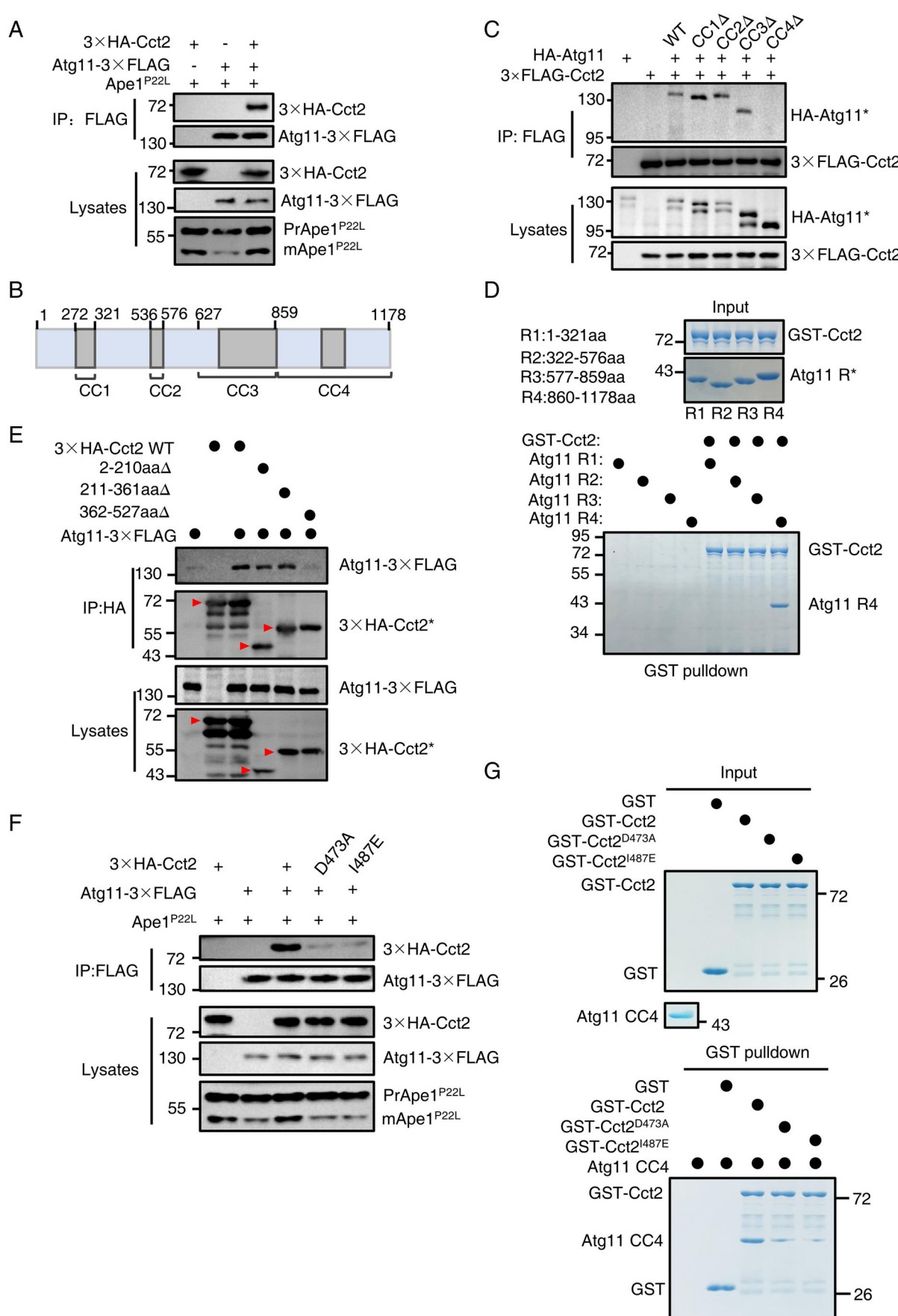

◀  **Figure 2.  Atg11 directly binds to Cct2.**

(A) *ape1Δ* cells co-expressing 3×HA-Cct2 and Atg11-3×FLAG with Ape1 P22L were grown to the early log phase under nutrient-rich conditions. Cell lysates were immunoprecipitated with anti-FLAG agarose beads and then analyzed by western blot using anti-HA antibody. The data are representative of three independent experiments. (B) Secondary structure diagram of Atg11. (C) *atg11Δ* cells co-expressing an empty vector (EV), WT HA-Atg11, HA-Atg11-CC1Δ, HA-Atg11-CC2Δ, HA-Atg11-CC3Δ, or HA-Atg11-CC4Δ with 3×FLAG-Cct2 were grown to the early log phase. Cell lysates were immunoprecipitated with anti-FLAG agarose beads and then analyzed by western blot using anti-HA antibody. The data are representative of two independent experiments. (D) In vitro GST pulldowns were performed using Atg11 R1, R2, R3, or R4 with GST-Cct2 purified from *E. coli*. Protein samples were separated by SDS-PAGE and detected using Coomassie blue staining. The data are representative of two independent experiments. (E) Cells co-expressing an empty vector, WT 3×HA-Cct2, 3×HA-Cct2(2-210aaΔ, D1Δ), 3×HA-CCT2(211-361aaΔ, D2Δ), or 3×HA-CCT2(362-527 aaΔ, D3Δ) with Atg11-3×FLAG were grown to the log phase. Cell lysates were immunoprecipitated with anti-HA agarose beads and then analyzed by western blot using anti-FLAG antibody. The red arrows indicate the bands corresponding to the wild-type (WT) and specific mutants of the 3×HA-Cct2 protein. The data are representative of two independent experiments. (F) *ape1Δ* cells co-expressing an empty vector (EV), WT 3×HA-Cct2, 3×HA-Cct2 D473A, or 3×HA- Cct2 I487E with Atg11-3×FLAG were grown to the early log phase in the presence of Ape1 P22L under nutrient-rich conditions. Cell lysates were immunoprecipitated with anti-FLAG agarose beads and then analyzed by western blot using anti-HA antibody. The data are representative of three independent experiments. (G) In vitro GST pulldowns were performed using Atg11 CC4 with GST, GST-Cct2, GST-Cct2 D473A, or GST-Cct2 I487E purified from *E. coli*. Protein samples were separated by SDS-PAGE and detected using Coomassie blue staining. The data are representative of two independent experiments. Source data are available online for this figure.

the absence of active Cct1 and Cct4 (Fig. 1F,G). In vitro pull-down assays additionally revealed that Atg8 binds exclusively to Cct2 and not to Cct1 or Cct4 (Fig. EV1B). Consistently, co-immunoprecipitation (co-IP) assays indicated that Atg8 binds only to free Cct2 and does not interact with other subunits of the TRiC complex (Fig. EV1C). These results strongly suggest that the regulation of solid aggrephagy by Cct2 in yeast is highly conserved with that in mammalian cells, and Cct2's involvement in solid aggrephagy does not depend on the TRiC complex function.

Subsequently, we investigated whether endogenous Cct2 is acting in aggregate turnover. To test this, we used a degron system to induce the degradation of Cct2 (Nishimura et al, 2009). Western blot data revealed that the degradation of Cct2 by IAA treatment significantly reduces the maturation of Ape1-P22L under nutrient-rich conditions (Fig. 1H,I), indicating that Cct2 is required for aggregate turnover. Subsequently, we explored whether Cct2 can be degraded by autophagy in the presence of Ape1 P22L under nutrient-rich medium. Fluorescence microscopy results revealed that Cct2-GFP enters the yeast vacuole in *atg15Δ* or *pep4Δ* cells, whereas the entry of Cct2-GFP into the vacuole is completely inhibited in *atg2Δ atg15Δ* cells (Fig. EV1D,E), indicating that Cct2 is a degradation substrate of autophagy. Collectively, these data suggest that the function of Cct2 as an aggrephagy receptor is conserved in yeast.

## Atg11 directly binds to Cct2

To better understand the roles and functions of Cct2 in the process of solid aggrephagy, we first investigated whether Cct2 could directly interact with other autophagy-related proteins beyond Atg8 and whether these interactions play a role in solid aggrephagy. In *Saccharomyces cerevisiae*, as a selective autophagy receptor, Cct2 also should bind with Atg11 in addition to Atg8 (Yorimitsu and Klionsky, 2005; Mochida et al, 2015). Consequently, we probed the potential association between Cct2 and Atg11. Co-IP experiments revealed that Cct2 can interact with Atg11 in the presence or absence of Ape1-P22L under nutrient-rich conditions (Fig. 2A; Appendix Fig. S1A). Previous study has highlighted the crucial role of the Atg11 CC4 domain in mediating its association with Cvt pathway receptor Atg19 (Yorimitsu and Klionsky, 2005). Consistently, co-IP results further substantiated this by confirming that Atg11 CC4 is required for its binding to Cct2 (Fig. 2B,C). Furthermore, to demonstrate the regulatory role of Atg11 CC4 in

regulating the interaction between Atg11 and Cct2, we performed in vitro GST pulldown assay and found that Atg11 CC4 can directly interact with Cct2 (Fig. 2D).

Next, we sought to identify which critical region or specific amino acid(s) on Cct2 are responsible for its interaction with Atg11. Structurally, Cct2 is compartmentalized into three domains, designated as D1, D2, D3, with Cct2 D3 previously established as vital for the association of Cct2 with Atg8/LC3C (Ma et al, 2022). We generated a series of individual deletions at 2–210 aa, 211–361 aa, and 362–527 aa and used these deletion variants for co-IP assays. Transformation of these HA-Cct2 variants in yeast cells expressing Atg11-3×FLAG resulted in a defect in the binding of Atg11 to Cct2 in cells harboring HA-Cct2 362-527aaΔ, indicating that the indispensable role of Cct2 D3 in the Cct2-Atg11 binding (Fig. 2E). Following this, we constructed a set of deleted mutants or random amino acid conversions within this critical region to assess their impact on the binding efficiency of Atg11 with Cct2, our results showed a significant reduction in Atg11-Cct2 binding with the Cct2 D473A or Cct2 I487E mutants in the presence or absence of Ape1 P22L under nutrient-rich conditions (Fig. 2F; Appendix Fig. S1B–E), highlighting that their binding was dependent on these residues in Cct2. Subsequently, we subjected the Cct2 mutants to GST pulldown assays to assess their Atg11 binding, further confirming that Atg11-Cct2 binding was nearly completely lost in the Cct2 D473A or Cct2 I487E mutants (Fig. 2G). In conclusion, these findings suggest that Cct2 directly interacts with Atg11, with the residues D473 and I487 of Cct2 playing a crucial role in the Cct2-Atg11 interaction.

## Cct2 binding to Atg11 is essential for its interaction with Atg8

We then used these mutants to investigate whether Atg11-Cct2 binding was required for solid aggrephagy. To further confirm that Ape1 P22L and GFP-47Q forms solid aggregates, we performed a fluorescence recovery after photobleaching (FRAP) assay and observed minimal fluorescence recovery for both Ape1 P22L and GFP-47Q after bleaching (Fig. EV2A–D) (Ma et al, 2022), indicating the formation of solid aggregates. To assess the impact of Cct2-Atg11 binding on solid aggrephagy, we transformed empty vector, HA-Cct2 WT, HA-Cct2 D473A, or HA-Cct2 I487E mutant plasmids into the *ape1Δ* yeast strain expressing RFP-Ape1 P22L. Imaging and statistical analysis revealed a notable decrease in the

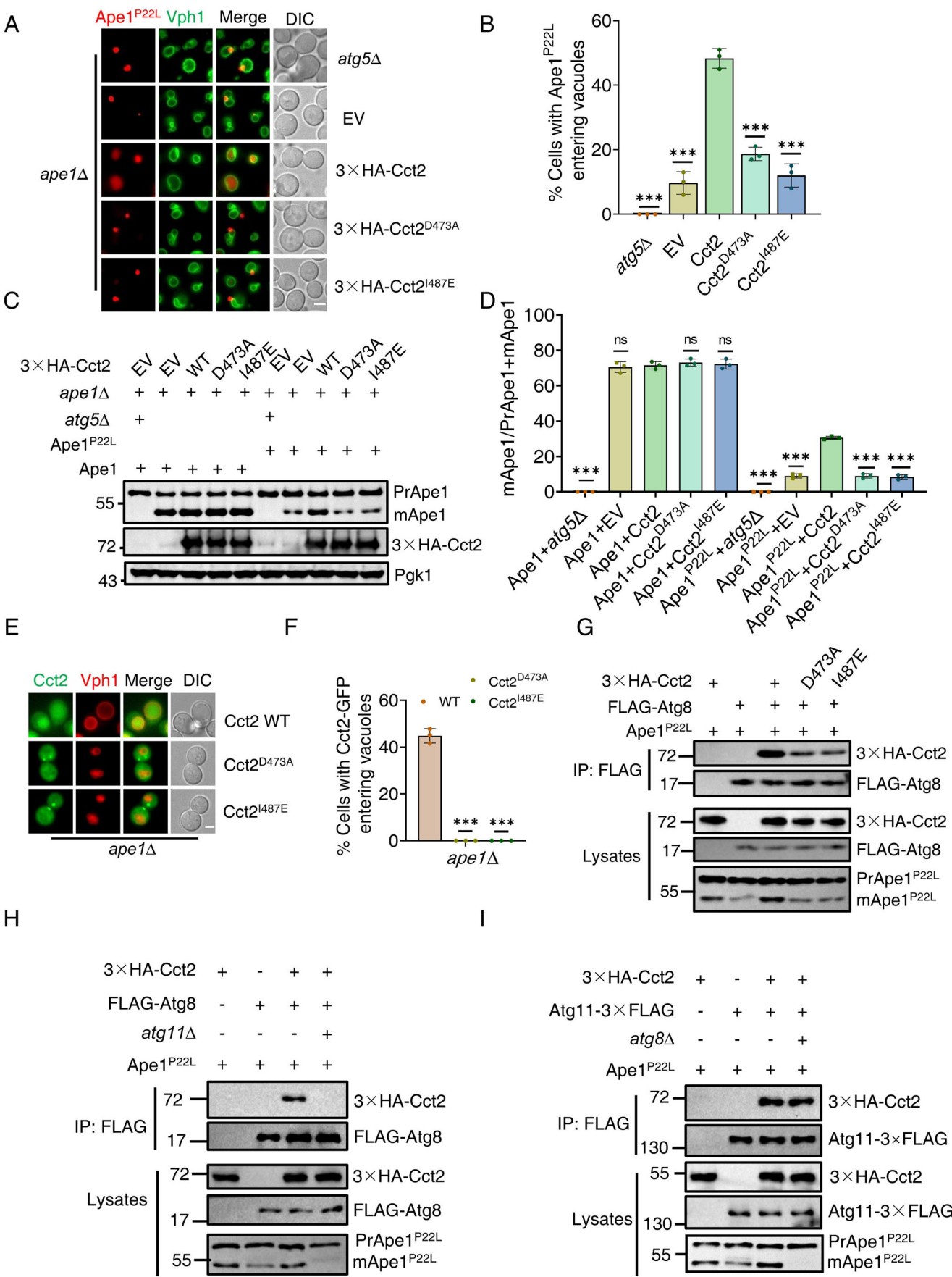

**Figure 3. The binding Cct2 with Atg11 is required for its association with Atg8.**

(A) *ape1Δ* or *ape1Δ atg5Δ* yeast cells co-expressing RFP-Ape1 P22L, Vph1-GFP with empty vector (EV), WT 3×HA-Cct2, or 3×HA-Cct2 variants were grown to log phase under nutrient-rich conditions. Images of cells were obtained using an inverted fluorescence microscope. Scale bar, 2 μm. (B) Cells from (A) were quantified for the vacuolar localization of RFP-Ape1 P22L. $n = 300$ cells were pooled from three independent experiments. Data are presented as means ± SD. \*\*\*$P < 0.001$; two-tailed Student's *t* tests were used. $P < 0.0001$ (*atg5Δ* vs. Cct2); p = 0.0001 (EV vs. Cct2); $P = 0.0002$ (Cct2 D473A vs. Cct2); $P = 0.0002$ (Cct2 I487E vs. Cct2). (C) *ape1Δ atg5Δ* or *ape1Δ* yeast cells co-expressing WT Ape1 or Ape1 P22L with empty vector (EV), 3×HA-Cct2 WT, or the indicated 3×HA-Cct2 variants were grown to log phase under nutrient-rich conditions. Samples were analyzed by immunoblot for detecting the maturation of PrApe1 into mApe1. Pgk1 served as a loading control. (D) Quantification of mApe1 to PrApe1 ratio from (C). Data are presented as means ± SD (Data represent the results of three independent experiments). \*\*\*$P < 0.001$; NS not significant; two-tailed Student's *t* tests were used. $P < 0.0001$ (WT *atg5Δ* vs. WT Cct2); $P = 0.6407$ (WT EV vs. WT Cct2); $P = 0.4033$ (WT Cct2 D473A vs. WT Cct2); $P = 0.7400$ (WT Cct2 I487E vs. WT Cct2); $P < 0.0001$(P22L *atg5Δ* vs. P22L Cct2); $P < 0.0001$(P22L EV vs. P22L Cct2); $P < 0.0001$(P22L Cct2 D473A vs. P22L Cct2); $P < 0.0001$(P22L Cct2 I487E vs. P22L Cct2). (E) *ape1Δ* Ape1 P22L yeast cells co-expressing Vph1-mCherry with Cct2-GFP, Cct2 [D473A]-GFP, or Cct2 [I487E]-GFP were grown to $OD_{600} = 0.8$, and then further cultured in nutrient-rich medium for 6 h. Images of cells were obtained using an inverted fluorescence microscope. Scale bar, 2 μm. (F) Cells from (E) were quantified for the vacuolar localization of Cct2-GFP. $n = 300$ cells were pooled from three independent experiments. Data are presented as means ± SD. \*\*\*$P < 0.001$; two-tailed Student's *t* tests were used. $P < 0.0001$(Cct2 D473A vs. WT); $P < 0.0001$(Cct2 I487E vs. WT). (G) *ape1Δ atg8Δ* cells co-expressing an empty vector (EV), 3×HA-Cct2, 3×HA-Cct2 [D473A], or 3×HA-Cct2 [I487E] with 3×FLAG-Atg8 in the presence of Ape1 P22L were grown to the log phase under nutrient-rich medium. Cell lysates were immunoprecipitated with anti-FLAG agarose beads and then analyzed by western blot using anti-HA antibody. The data are representative of three independent experiments. (H) *ape1Δ* or *ape1Δ atg11Δ* cells co-expressing 3×HA-Cct2 and FLAG-Atg8 were grown to log phase in the presence of Ape1 P22L under nutrient-rich conditions. Cell lysates were immunoprecipitated with anti-FLAG agarose beads and then analyzed by western blot using anti-HA antibody. The data are representative of three independent experiments. (I) *ape1Δ* or *ape1Δ atg8Δ* cells co-expressing 3×HA-Cct2 and Atg11-3×FLAG were grown to log phase in the presence of Ape1 P22L under nutrient-rich conditions. Cell lysates were immunoprecipitated with anti-FLAG agarose beads and then analyzed by western blot using anti-HA antibody. The data are representative of three independent experiments. Source data are available online for this figure.

vacuolar localization of Ape1 P22L in the presence of HA-Cct2 [D473A] or HA-Cct2 [I487E], while HA-Cct2 promoted the vacuolar localization of Ape1 P22L under nutrient-rich conditions (Fig. 3A,B). In line with the imaging data, the prApe1 maturation assay found that HA-Cct2 [D473A] or HA-Cct2 [I487E] mutants severely impaired the maturation of PrApe1 P22L, while these mutants had no effect on the PrApe1 maturation compared to HA-Cct2 under nutrient-rich conditions (Fig. 3C,D). This suggests that Cct2-Atg11 binding is specifically required for the maturation of PrApe1 P22L. Further transformation of the same plasmids into a yeast strain expressing GFP-47Q with Cup1 promoter resulted in impaired cleavage of GFP-47Q with HA-Cct2 [D473A] or HA-Cct2 [I487E] compared to HA-Cct2 in the presence of $Cu^{2+}$ under nutrient-rich conditions (Fig. EV2E,F). Furthermore, the deficiency of the Cct2-Atg11 blocked the vacuolar localization of Cct2-GFP in the presence of Ape1 P22L under nutrient-rich medium (Fig. 3E, F). Collectively, these data indicate that Cct2-Atg11 binding is required for solid aggrephagy.

Next, we sought to investigate the potential molecular mechanism underlying the Cct2-Atg11 binding in solid aggrephagy. Considering the crucial role of Atg11 in selective autophagy and the necessity of Cct2-Atg8 binding for solid aggrephagy (Ma et al, 2022; Zientara-Rytter and Subramani, 2020), we investigated whether the absence of Cct2 interaction with Atg11 affects Cct2-Atg8 binding. We co-transformed FLAG-Atg8 with HA-Cct2, HA-Cct2 [D473A], or HA-Cct2 [I487E] mutants into *atg8Δ apeα1Δ* yeast cells expressing the Ape1 P22L plasmid. Co-IP assays revealed that the deficiency of Cct2 interaction with Atg11 significantly inhibits Cct2-Atg8 binding in the presence of Ape1 P22L under nutrient-rich conditions (Fig. 3G). However, an in vitro GST pulldown assay revealed no significant change in the binding of Cct2 and Atg8 in the GST-Cct2 D473A or GST-Cct2 I487E mutants compared to WT GST-Cct2 (Fig. EV2G), indicating that Atg11 may mediate the binding of Atg8 and Cct2 in yeast cells. To test it, we transformed HA-Cct2 and FLAG-Atg8 plasmids into wild-type or *atg11Δ* yeast strains. Co-IP assays indicated that the binding of Cct2-Atg8 was nearly entirely inhibited in the absence of *ATG11* (Fig. 3H), highlighting the critical role of Atg11 as a key mediator of the Cct2-

Atg8 binding. To explore whether other autophagy-related proteins are involved in the interaction between Cct2 and Atg8, we knocked out *ATG2* and *ATG14* in yeast cells co-expressing HA-Cct2 and FLAG-Atg8, respectively. Co-IP experiments showed that the deletion of *ATG2* or *ATG14* did not impair the binding of Cct2 with Atg8 under nutrient-rich conditions in the presence of Ape1 P22L (Fig. EV2H,I). However, the knockout of *ATG8* did not affect the binding of Atg11 with Cct2 in the presence of Ape1 P22L under nutrient-rich conditions (Fig. 3I). Furthermore, in vitro GST pulldown assays indicated that the Cct2 AIM mutant does not affect the binding of Cct2 to Atg11 CC4 domain (Fig. EV2J), indicating that Cct2 AIM specifically regulates its association with Atg8 in yeast. We subsequently explored whether Atg11 is required for the turnover of aggregates. The cleavage of Ape1 P22L and GFP-47Q assays showed that Atg11 play a crucial role in solid aggrephagy under nutrient-rich conditions (Fig. EV3A–D).

Given the important role of the interaction between Atg11 and Cct2 in solid aggrephagy, we investigated whether the interaction between Cct2 and Atg11 is required for the colocalization of Atg11 with Ape1 P22L. First, we examined whether the aggregate Ape1 P22L can colocalize with Atg1, Atg8, Atg11, and Cct2. Imaging and statistical analysis indicated that Ape1 P22L has colocalization with Cct2 and the autophagy-related proteins (Fig. EV3E,F). However, when the interaction between Atg11 and Cct2 was disrupted, the colocalization of Ape1 P22L with Atg11 significantly decreased, suggesting that the interaction between Atg11 and Cct2 is required for recruiting the solid aggregate to the PAS (Fig. EV3G,H). Taken together, these data suggested that Atg11 acts as a key regulator for Cct2-Atg8 binding and that its association with Cct2 is required for solid aggrephagy by regulating the binding of Cct2-Atg8.

## Cct2 is a novel substrate of Atg1

Given that post-translational modification plays an important role in selective autophagy (Xie et al, 2015), we sought to investigate the involvement of post-translational modification of Cct2 in solid aggrephagy. Since Atg1 is a protein kinase and both Atg1 and Cct2 directly associate with Atg8 and Atg11, we aimed to determine

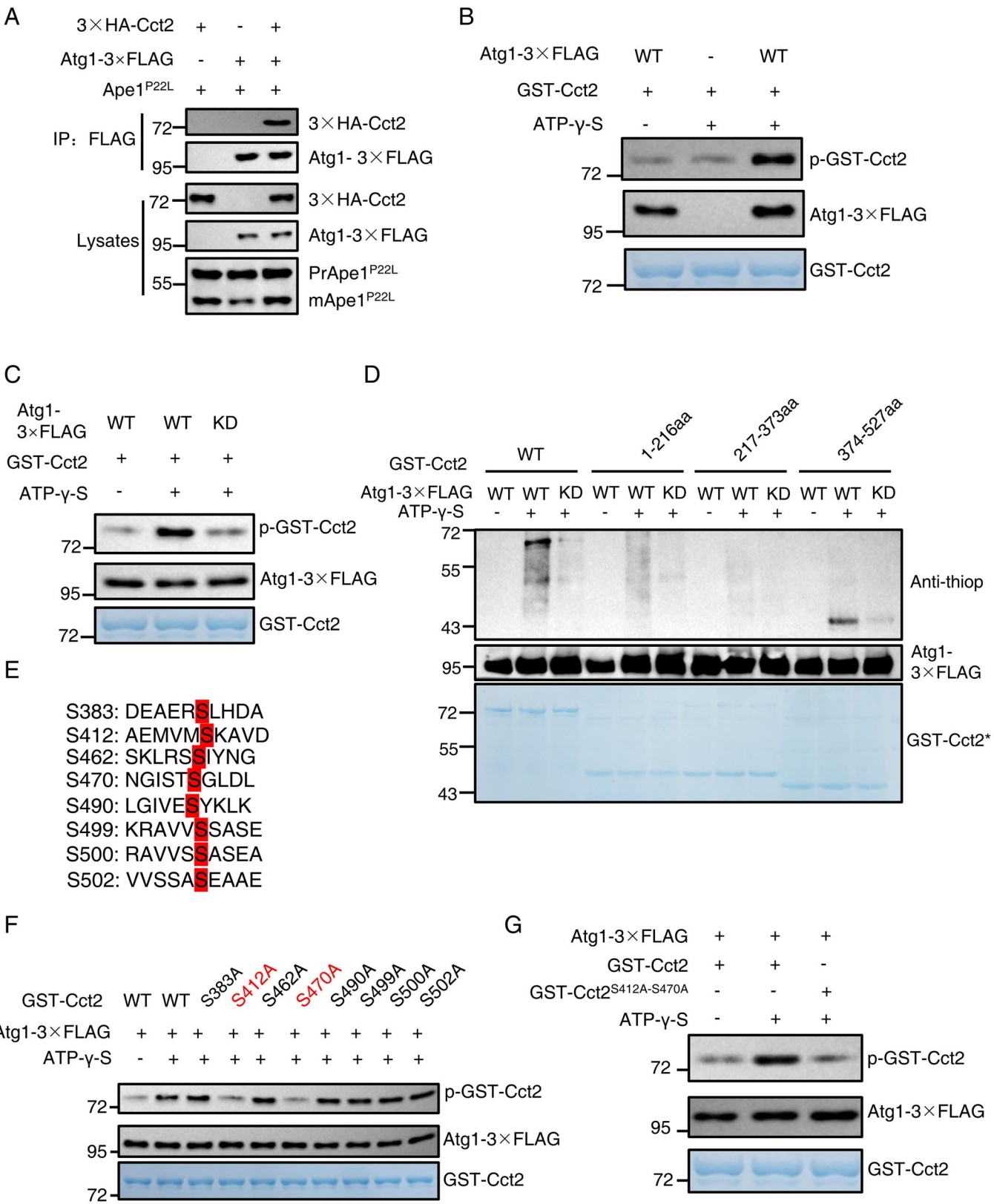

◀ **Figure 4. Cct2 is a novel substrate of Atg1.**

(A) *ape1Δ* cells co-expressing 3×HA-Cct2 and Atg1-3×FLAG were grown to the log phase in the presence of Ape1 P22L under nutrient-rich conditions. Cell lysates were immunoprecipitated with anti-FLAG agarose beads and then analyzed by western blot using anti-HA antibody. The data are representative of three independent experiments. (B) In vitro kinase assays were performed using the GST-Cct2 purified from *E. coli* as substrates and WT Atg1-3×FLAG purified from yeast cells expressing Ape1 P22L under nutrient-rich conditions as the protein kinase. Phosphorylation of GST-Cct2 was detected using anti-thioP antibody. The data are representative of two independent experiments. (C) In vitro kinase assays were performed using the GST-Cct2 purified from *E. coli* as substrates with WT (wild-type) or KD (kinase-dead) Atg1-3×FLAG purified from yeast cells expressing Ape1 P22L under nutrient-rich conditions as a protein kinase. Phosphorylation of GST-Cct2 was detected using anti-thioP antibody. The data are representative of two independent experiments. (D) In vitro kinase assays were performed using the GST-Cct2 or its variants purified from *E. coli* as substrates with wild-type (WT) Atg1-3×FLAG or Atg1 kinase dead (KD) purified from yeast cells expressing Ape1 P22L under nutrient-rich conditions as the protein kinase. Phosphorylation of GST-Cct2 was detected using anti-thioP antibody. The data are representative of two independent experiments. (E) Eight typical Atg1 phosphorylation sites on GST-Cct2 D3 domain. (F) An in vitro kinase assays were performed using GST, GST-Cct2, or eight indicated GST-Cct2 variants were purified from *E. coli* as substrates and purified Atg1-3×FLAG from yeast cells expressing Ape1 P22L under nutrient-rich conditions as a protein kinase. The phosphorylation of Cct2 and these variants were detected by anti-thioP antibody. The data are representative of two independent experiments. (G) In vitro kinase assays were performed using the GST-Cct2 or GST-Cct2 S412A-S470A purified from *E. coli* as substrates with Atg1-3×FLAG purified from yeast cells expressing Ape1 P22L under nutrient-rich conditions as a protein kinase. Phosphorylation of Cct2 and Cct2 S412A-S470A were detected using anti-thioP antibody. The data are representative of two independent experiments. Source data are available online for this figure.

whether Atg1 can phosphorylate Cct2 (Yorimitsu and Klionsky, 2005; Nakatogawa et al, 2012). Co-IP assays were performed and revealed that Atg1 can associate with Cct2 in the presence of Ape1 P22L under nutrient-rich conditions (Fig. 4A), suggesting Cct2 may be a substrate of Atg1 (Kim et al, 2011). Subsequently, we cloned the *CCT2* gene into an *E. coli* expression vector pGEX-4T-1 and performed in vitro kinase assays to examine whether Cct2 can indeed be phosphorylated by Atg1 (Allen et al, 2007). The results showed that Cct2 can be phosphorylated by Atg1 (Fig. 4B). To further validate the essential role of Atg1 kinase activity in the phosphorylation of Cct2, we used purified wild-type (WT) and kinase-dead (KD) Atg1 (Atg1 $^{D211A}$) as protein kinase with Cct2 as a substrate in the in vitro kinase assays (Yao et al, 2023b). Our findings demonstrated that WT Atg1, but not KD Atg1, could phosphorylate Cct2 (Fig. 4C), further supporting the identification of Cct2 as a novel substrate of Atg1.

We proceeded to identify the phosphorylation site(s) of Cct2 by Atg1. Given that Cct2 is comprised of three functional domains, namely D1, D2, and D3, our focus was to determine which domain(s) of Cct2 is phosphorylated by Atg1. In vitro kinase assays showed that Cct2 D3 was specifically phosphorylated by Atg1 (Fig. 4D). Subsequently, we conducted an analysis to pinpoint the precise site(s) on Cct2 that are phosphorylated by Atg1. Sequence analysis of yeast Cct2 D3 domain revealed eight potential Atg1 consensus sites (Papinski et al, 2014), namely Ser383, Ser412, Ser462, Ser470, Ser490, Ser499, Ser500, and Ser502 (Fig. 4E). To investigate further, we generated serine to alanine conversion mutants for each of these eight sites. Notably, in vitro kinase assays demonstrated that the S412A or S470A mutation resulted in a substantial reduction in Cct2 phosphorylation (Fig. 4F). Furthermore, our findings indicated that a double mutant version, S412A-S470A, of the Cct2 D3 region almost completely abolished phosphorylation (Fig. 4G). Simultaneously, we also examined the phosphorylation modification of these two residues in vivo. To accomplish this, we purified 3×FLAG-Cct2 from wild-type or *atg1Δ* yeast cells in the presence of Ape1 P22L under nutrient-rich conditions using anti-FLAG agarose beads. The subsequent LC–MS showed that Cct2 in wild-type yeast cells is phosphorylated at S412 and S470, whereas in *atg1Δ* cells, phosphorylation at S412 and S470 was absent (Dataset EV1). Collectively, these findings demonstrate that Atg1 phosphorylates Cct2 at the S412 and S470 residues during the process of solid aggrephagy.

## Atg1-mediated Cct2 phosphorylation is required for solid aggrephagy via regulating the association of Cct2 with Atg8 and Atg11

Next, we investigated the necessity of Atg1-mediated Cct2 phosphorylation for solid aggrephagy. To this end, we separately transformed HA-Cct2, HA-Cct2 $^{S412A-S470A}$ (2A), or empty vector plasmids into *ape1Δ* yeast cells expressing RFP-Ape1 P22L. Subsequent imaging and statistical analysis indicated a significant reduction in the vacuolar localization of Ape1 P22L in the presence of HA-Cct2-2A, whereas HA-Cct2 facilitated the vacuolar localization of Ape1 P22L under nutrient-rich conditions (Fig. 5A,B). Consistent with these observations, the prApe1 maturation assay, performed by transforming HA-Cct2, HA-Cct2-2A, or empty vector plasmids into *ape1Δ* yeast cells expressing Ape1 or Ape1 P22L, showed that the HA-Cct2-2A mutant had little impact on the maturation of PrApe1 but notably reduced the maturation of Ape1 P22L under nutrient-rich medium (Fig. 5C,D). To further confirm this observation, we separately transformed HA-Cct2, HA-Cct2-2A, or empty vector plasmids into yeast cells expressing GFP-poly 47Q plasmids. The GFP-47Q cleavage analysis indicated that the free GFP level in the HA-Cct2-2A mutation decreased to a level similar to that in the empty vector under nutrient-rich conditions (Fig. EV4A,B). Taken together, these data indicated that the phosphorylation of Cct2 by Atg1 is required for solid aggrephagy.

Consequently, we investigated how the phosphorylation of Cct2 by Atg1 regulates the solid aggrephagy. Considering that the binding of Cct2 with Atg8 and Atg11 is required for solid autophagy, we separately transformed HA-Cct2 or HA-Cct2-2A into yeast cells expressing FLAG-Atg8 or FLAG-Atg11. As shown in Figs. 5E and EV4C, HA-Cct2-2A significantly impaired the association of Cct2 with both Atg8 and Atg11 compared to HA-Cct2 in the presence of Ape1 P22L under nutrient-rich conditions. Additionally, we investigated whether the deletion of *ATG1* affects the interaction of Cct2 with both Atg8 and Atg11. Co-IP assays showed a significant decrease in Cct2 binding to both Atg8 and Atg11 in *atg1Δ* cells compared to wild-type cells under nutrient-rich conditions in the presence of Ape1-P22L (Figs. 5F and EV4D). To rule out the possibility that the impaired binding of Cct2 to Atg8 and Atg11 was due to structural changes caused by Cct2-2A mutant, we expressed and purified the Cct2-2A mutant protein from *E. coli*. In vitro GST pulldown assays indicated that, compared

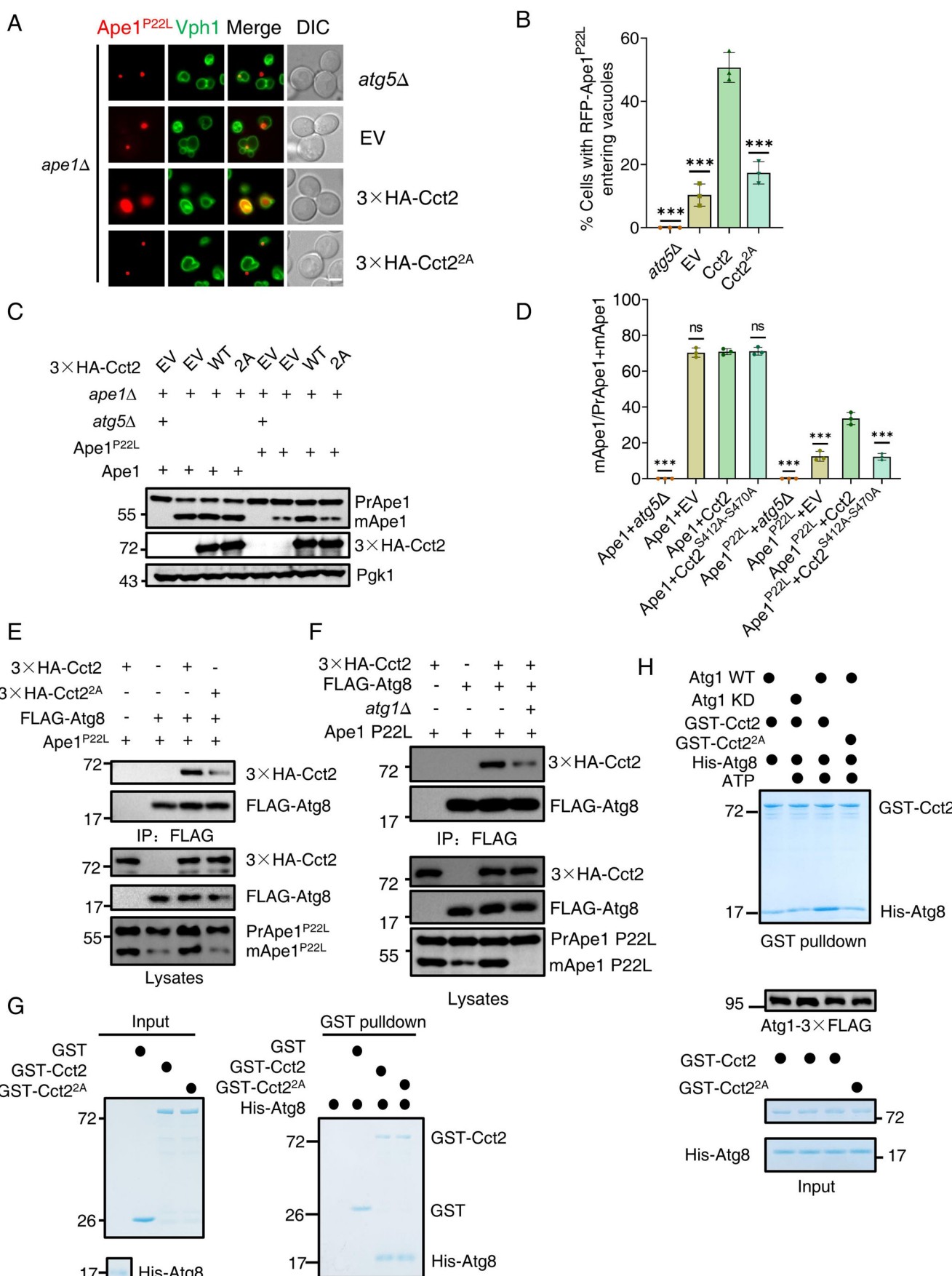

◄  **Figure 5.  Atg1-mediated Cct2 phosphorylation regulates the binding of Cct2 with Atg8 and Atg11.**

(A) *ape1Δ* or *ape1Δ atg5Δ* yeast cells co-expressing RFP-Ape1 P22L, Vph1-GFP with empty vector (EV), 3×HA-Cct2 WT or 3×HA-CCT2 ^S412A-S470A were grown to the log phase under nutrient-rich conditions. Images of cells were obtained using an inverted fluorescence microscope. Scale bar, 2 μm. (B) Cells from (A) were quantified for the vacuolar localization of RFP-Ape1 P22L. $n = 300$ cells were pooled from three independent experiments. Data are presented as means ± SD. ***$P < 0.001$; two-tailed Student's *t* tests were used. $P < 0.0001$ (*atg5Δ* vs. Cct2); $P = 0.0003$(EV vs. Cct2); $P = 0.0006$(Cct2 2A vs. Cct2). (C) *ape1Δ atg5Δ* or *ape1Δ* yeast cells co-expressing Ape1 WT or Ape1 P22L with empty vector (EV), wild-type 3×HA-Cct2 (WT), or 3×HA-Cct2 ^S412A-S470A (2A) were grown to the log phase under nutrient-rich conditions. Samples were analyzed by immunoblot for detecting the maturation of PrApe1 into mApe1. Pgk1 served as a loading control. (D) Quantification of mApe1 to PrApe1 ratio from (C). Data are presented as means ± SD (Data represent the results of three independent experiments). ***$P < 0.001$; NS, not significant; two-tailed Student's *t* tests were used. $P < 0.0001$ (WT *atg5Δ* vs. WT Cct2); $P = 0.7945$ (WT EV vs. WT Cct2); $P = 0.8705$ (P22L Cct2 S412A-S470A vs. WT Cct2); $P < 0.0001$ (P22L *atg5Δ* vs. P22L Cct2); $P = 0.0012$ (P22L EV vs. P22L Cct2); $P = 0.0007$ (P22L Cct2 S412A-S470A vs. P22L Cct2). (E) *ape1Δ atg8Δ* cells co-expressing an empty vector (EV), 3×HA-Cct2, 3×HA-Cct2 ^S412A-S470A (2A) with FLAG-Atg8 were grown into the log phase under nutrient-rich conditions. Cell lysates were immunoprecipitated with anti-FLAG agarose beads and then analyzed by western blot using anti-HA antibody. The data are representative of three independent experiments. (F) *ape1Δ atg1Δ atg8Δ* cells co-expressing an empty vector (EV), 3×HA-Cct2, 3×HA-Cct2 ^S412A-S470A (2A) with FLAG-Atg8 were grown to the log phase under nutrient-rich conditions. Cell lysates were immunoprecipitated with anti-FLAG agarose beads and then analyzed by western blot using anti-HA antibody. The data are representative of three independent experiments. (G) In vitro GST pulldowns were performed using His-Atg8 with GST, GST-Cct2, GST-Cct2 ^S412A-S470A (2A) purified from *E. coli*. Protein samples were separated by SDS-PAGE and detected using Coomassie blue staining. The data are representative of two independent experiments. (H) In vitro kinase assays were performed using GST-Cct2 WT or 2A (S412A-S470A) purified from *E. coli* as substrates, with WT or KD Atg1-3×FLAG purified from yeast cells expressing Ape1 P22L under nutrient-rich conditions as the protein kinase. After that, in vitro GST pulldowns were performed using phosphorylated products enriched by GST beads in vitro with His-Atg8 protein purified from *E. coli*. Protein samples were separated by SDS-PAGE, and then detected using Coomassie blue staining. The data are representative of three independent experiments. Source data are available online for this figure.

to the wild-type Cct2, the Cct2-2A mutant protein did not impair the binding of Cct2 to Atg8 and Atg11 (Figs. 5G and EV4E), indicating that the Cct2 2A mutation does not alter its structure. Taken together, these findings suggest that Atg1-mediated Cct2 phosphorylation is required for its association with Atg8 and Atg11.

To further confirm that Atg1-mediated Cct2 phosphorylation indeed promotes Cct2 binding with Atg8 and Atg11, we performed in vitro kinase reactions followed by pull-down assays. We found that Cct2 phosphorylation by Atg1 promotes its binding to both Atg8 and Atg11 in vitro (Figs. 5H and EV4F). Additionally, we performed competitive binding experiments between Atg11 and Atg8 with Cct2. We found that although Atg1-mediated Cct2 phosphorylation increases its binding to both Atg8 and Atg11, the binding affinity of Cct2 to Atg11 CC4 is stronger than that to Atg8 (Fig. EV4G). Cumulatively, these results precisely suggest that Cct2 can enhance its binding with Atg8 and Atg11, and subsequently initiate aggrephagy, when phosphorylated by Atg1.

## The requirement for both Atg1-mediated Cct2 phosphorylation and Atg11 in the binding of Cct2-Atg8 is conserved in mammals

The conservation of CCT2 as a solid aggrephagy receptor from yeast to humans prompted us to examine whether human CCT2 could be phosphorylated by ULK1 (the homolog of Atg1 in mammals) (Ma et al, 2022). Consistent with the results observed in yeast, we found that CCT2 exhibited association with ULK1 in the presence of SOD1 G93A GFP under nutrient-rich conditions (Fig. EV5A). Subsequent in vitro kinase assays confirmed that ULK1 also phosphorylates CCT2 (Fig. 6A). In mammals, CCT2 consists of three conserved domains (Ma et al, 2022). In lines with the results from yeast, in vitro kinase assays indicated that the D3 domain of mammalian CCT2 can be phosphorylated by ULK1 (Fig. 6B). Through analysis of the conserved phosphorylation motif of Atg1/ULK1, we identified six consensus motifs in CCT2 D3 domain (Fig. EV5B). We generated serine-to-alanine conversion mutants for each of these six sites. In vitro kinase assays indicated that S458A, S470A, and S506A mutations led to significantly

decreased phosphorylation of CCT2 (Figs. 6C and EV5C). Furthermore, we tested the phosphorylation modification of CCT2 in vivo. To this end, we purified HA-CCT2 from wild-type or ULK1 KO MEF cells in the presence of SOD1 G93A GFP under nutrient-rich medium using anti-HA agarose beads. LC–MS analysis of the protein sample identified that only the S458 residue was phosphorylated in wild-type MEF cells, whereas S458 is not phosphorylated in ULK1 KO MEF cells (Dataset EV2). Taken together, these results suggest that ULK1 phosphorylates CCT2 at the S458 residues.

To assess the significance of CCT2 phosphorylation by ULK1 in aggrephagy, we transformed HEK293T cells expressing SOD1 G93A-GFP or GFP-α-Synuclein A53T with HA-CCT2, HA-CCT2 mVL(I)L (LIR motif mutant), HA-CCT2 ^S458A or empty vector plasmids. Western blot results showed that HA-CCT2 ^S458A significantly impairs the degradation of these solid aggregates (Fig. 6D–G), indicating that ULK1-mediated phosphorylation of CCT2 is required for aggrephagy in mammals. Additionally, we investigated whether the degradation of the corresponding wild-type proteins depends on CCT2. Western blotting confirmed that the degradation of these wild-type proteins is not dependent on CCT2 (Fig. EV5D–G), indicating that CCT2 specifically regulates aggrephagy.

Subsequently, we investigated how ULK1-mediated phosphorylation of CCT2 regulates aggrephagy. Given that Atg1-mediated CCT2 phosphorylation is required for the interaction between Cct2 and Atg8 in yeast, and CCT2-LC3C binding is crucial for aggrephagy, we tested whether the phosphorylation of CCT1 by ULK1 regulates its binding to LC3C. FLAG-CCT2 or FLAG-CCT2 ^S458A plasmids were transfected into HEK293T cells expressing HA-LC3C. The results indicated that, compared with HA-CCT2, HA-CCT2 ^S458A significantly decreased the association of CCT2 with LC3C (Fig. 6H,I). We then employed a ULK1 KO MEF cell line to examine whether ULK1 regulates the interaction between CCT2 and LC3C. Co-IP assays confirmed that ULK1 KO significantly decreases the binding of CCT2 to LC3C in the presence of SOD1 G93A-GFP under nutrient-rich conditions (Fig. 6J–L). Additionally, an in vitro GST pulldown assay revealed that the CCT2 ^S458A protein did not affect the binding of CCT2 with

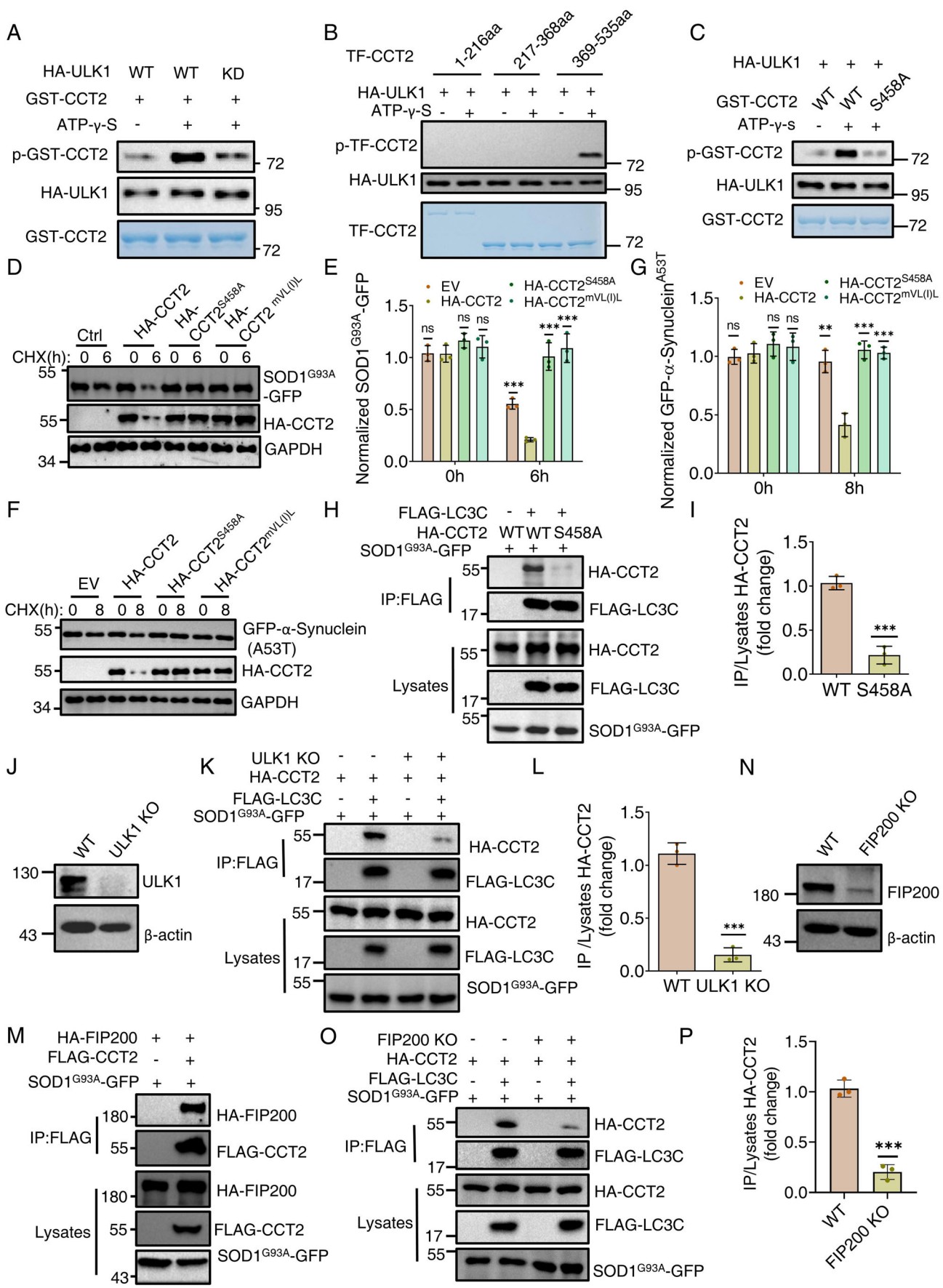

Figure 6. The requirement for both Atg1-mediated Cct2 phosphorylation and Atg11 in the binding of Cct2-Atg8 is conserved in mammals.

(A) In vitro kinase assays were performed using the GST-hCCT2(human) purified from *E. coli* as substrates with WT (wild-type) or KD (kinase-dead) HA-ULK1 purified from SOD1 G93A-GFP expressed HEK293T cells in the full medium as the protein kinase. Phosphorylation of GST-CCT2 was detected using anti-thioP antibody. The data are representative of two independent experiments. (B) In vitro kinase assays were performed using the GST-CCT2 1–216 aa (D1), 217–368 aa (D2), or 369–535 aa (D3) purified from *E. coli* as substrates with HA-ULK1 purified from SOD1 G93A-GFP expressed HEK293T cells under nutrient-rich conditions as the protein kinase. Phosphorylation of GST-CCT2 D1, D2 or D3 was detected using anti-thioP antibody. The data are representative of two independent experiments. (C) An in vitro kinase assays were performed using GST-hCCT2, or GST-hCCT2 S458A was purified from *E. coli* as substrates and purified HA-ULK1 from SOD1 G93A-GFP expressed HEK293T cells under nutrient-rich medium as the protein kinase. The phosphorylation of hCCT2 and hCCT2 S458A were detected by anti-thioP antibody. The data are representative of two independent experiments. (D) The turnover of SOD1 G93A-GFP in a CHX chase assay with an empty vector, HA-CCT2, HA-CCT2 S458A, or HA-CCT2-mVL(I)L (LIR motif mutant) expression in Hela cell lines after 24 h transfection. The cells were permeabilized with digitonin before immunoblot analysis. Western blot analysis was performed to assess SOD1 G93A-GFP degradation. GAPDH served as a loading control. (E) Quantification of normalized SOD1 G93A-GFP from (D). Data are presented as means ± SD (Data represent the results of three independent experiments). ***$P < 0.001$; **$P < 0.01$; NS not significant; two-tailed Student's *t* tests were used. $P = 0.9610$ (EV 0 h vs. CCT2 0 h); $P = 0.1110$ (CCT2 S58A 0 h vs. CCT2 0 h); $P = 0.4411$ (mVL(I)L 0 h vs. CCT2 0 h); $P = 0.0004$ (EV 6 h vs. CCT2 6 h); $P = 0.0005$ (CCT2 S58A 6 h vs. CCT2 6 h); $P = 0.0004$ (mVL(I)L 6 h vs. CCT2 6 h). (F) The turnover of GFP-α-Synuclein (A53T) in a CHX chase assay with an empty vector, HA-CCT2, HA-CCT2 S458A, or HA-CCT2-mVL(I)L (LIR motif mutant) expression in Hela cell lines after 24 h transfection. The cells were permeabilized with digitonin before immunoblot analysis. Western blot analysis was performed to assess GFP-α-Synuclein (A53T) degradation. GAPDH served as a loading control. (G) Quantification of normalized GFP-α-Synuclein (A53T) (F). Data are presented as means ± SD (Data represent the results of three independent experiments). ***$P < 0.001$; **$P < 0.01$; NS not significant; two-tailed Student's *t* tests were used. $P = 0.6472$ (EV 0 h vs. CCT2 0 h); $P = 0.3573$ (CCT2 S58A 0 h vs. CCT2 0 h); $P = 0.5323$ (mVL(I)L 0 h vs. CCT2 0 h); $P = 0.0025$ (EV 8 h vs. CCT2 8 h); $P = 0.0009$ (CCT2 S58A 8 h vs. CCT2 8 h); $P = 0.0007$ (mVL(I)L 8 h vs. CCT2 8 h). (H) HEK293T cell lines were co-transfected with FLAG-LC3C, HA-CCT2 (WT/S458A), and SOD1 G93A-GFP. Cell lysates were immunoprecipitated with anti-FLAG agarose beads and then analyzed by western blot using an anti-HA antibody. (I) Quantification of IP/Lysates HA-CCT2 from (H). Data are presented as means ± SD (Data represent the results of three independent experiments). ***$P < 0.001$; two-tailed Student's *t* tests were used. $P = 0.0004$ (S458A vs. WT). (J) Western blot validation of ULK1 KO effect in MEF cell lines. The data are representative of two independent experiments. (K) WT or ULK1 KO MEF cell lines were co-transfected with FLAG-LC3C, HA-CCT2, and SOD1 G93A-GFP. Cell lysates were immunoprecipitated with anti-FLAG agarose beads and then analyzed by western blot using an anti-HA antibody. (L) Quantification of IP/Lysates HA-CCT2 from (K). Data are presented as means ± SD (Data represent the results of three independent experiments). ***$P < 0.001$; two-tailed Student's *t* tests were used. $P = 0.0002$ (ULK1 KO vs. WT). (M) HEK293T cell lines were co-transfected with FLAG-CCT2, HA-FIP200, and SOD1 G93A-GFP. Cell lysates were immunoprecipitated with anti-FLAG agarose beads and then analyzed by western blot using an anti-HA antibody. The data are representative of three independent experiments. (N) Western blot validation of FIP200 KO effect in Hela cell lines. The data are representative of two independent experiments. (O) WT or FIP200 KO Hela cell lines were co-transfected with FLAG-LC3C, HA-CCT2, and SOD1 G93A-GFP. Cell lysates were immunoprecipitated with anti-FLAG agarose beads and then analyzed by western blot using an anti-HA antibody. (P) Quantification of IP/Lysates HA-CCT2 from (O). Data are presented as means ± SD (Data represent the results of three independent experiments). ***$P < 0.001$; two-tailed Student's *t* tests were used. $P = 0.0002$ (FIP200 KO vs. WT). Source data are available online for this figure.

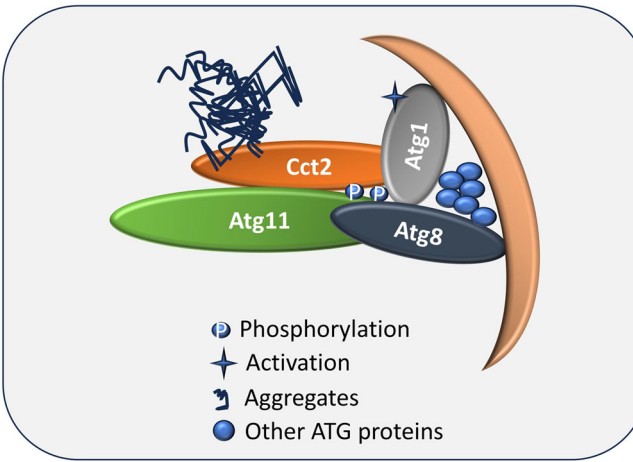

**Figure 7. Model for the regulation of Cct2-Atg8 binding by Atg1-mediated Cct2 phosphorylation and the binding of Cct2 to Atg11.**

During the process of solid aggrephagy, Atg1-mediated Cct2 phosphorylation and the binding of Cct2 to Atg11 are required for initiating solid aggrephagy by regulating the association of Cct2 with Atg8. P phosphorylated.

LC3C compared to HA-CCT2 (Fig. EV5H). These findings collectively suggest that ULK1-mediated phosphorylation of CCT2 is required for aggrephagy by regulating its association with LC3C.

Given that Atg11 binds with Cct2 and is required for the association of Cct2 with Atg8 in yeast, we next investigated whether

the mammalian homolog of Atg11, FIP200 (Zientara-Rytter and Subramani, 2020), can bind to CCT2 and regulate its association with LC3C. Co-IP assays revealed that FIP200 can indeed bind with CCT2 (Fig. 6M), and FIP200 KO significantly reduces the interaction between CCT2 and LC3C (Fig. 6N–P), whereas WIPI2 KO does not affect the binding of LC3C to CCT2 (Fig. EV5I–K). Taken together, these findings suggest that the regulatory roles of Atg11 and Atg1-mediated Cct2 phosphorylation in the binding of Cct2 to Atg8 are conserved in mammals.

## Discussion

In this study, we provide strong evidence that Cct2 plays a critical role in solid aggrephagy by interacting directly with Atg11 and subsequently regulating its association with Atg8. Our findings highlight the significance of specific domains and residues within Cct2 that are crucial for its interaction with Atg11, shedding light on the molecular basis of this regulatory mechanism. Furthermore, we identified Cct2 as a novel substrate of Atg1, emphasizing the importance of post-translational modification of Cct2 in solid aggrephagy. Importantly, we demonstrated the conservation of this regulatory mechanism in mammals, indicating the evolutionary significance of Atg11/FIP200 and Atg1/ULK1-mediated phosphorylation of Cct2 in the binding Cct2-Atg8/LC3C and subsequently aggrephagy (Fig. 7).

Our previous research revealed that Cct2/CCT2 binds to Atg8/LC3C through its AIM/LIR motif, and their interaction is a crucial step during the process of solid aggrephagy (Ma et al, 2022). In this study, we found that the phosphorylation of Cct2/CCT2 by

Atg1/ULK1 and the interaction between the adaptor protein Atg11/FIP200 and Cct2/CCT2 serve as key regulators governing the Cct2/CCT2-Atg8/LC3C binding, contributing to the solid aggrephagy. This indicates that the binding of CCT2 to Atg11 and the Atg1/ULK1-mediated CCT2 phosphorylation are prerequisites and important regulators for the interaction between CCT2 and Atg8, serving as necessary conditions for initiating solid aggrephagy. These findings broaden our understanding of the regulation of the binding of CCT2 with Atg8/LC3C. Moreover, these discoveries suggest that enhancing the interaction between CCT2 and Atg8/LC3C can be achieved by regulating the activity of these key factors, thereby promoting solid aggrephagy. Additionally, we aim to identify other unknown factors that may exist in the cell, which play important roles in regulating the interaction between CCT2 and Atg8/LC3C.

Moving forward, we noticed that the phosphorylation sites of CCT2 by Atg1 or ULK1 are not conserved in yeast and mammals. However, both the Atg1 and ULK1 phosphorylation region on CCT2 are located within the D3 domain of CCT2, and the phosphorylation sites are close to the AIM/LIR motif of CCT2. We hypothesize that this discrepancy may be caused by the different CCT2-associated proteins in yeast and mammals or by slight differences in the three-dimensional structure of yeast and mammalian CCT2. In fact, some studies have found that Atg1/ULK1 phosphorylates the same substrates but at different phosphorylation sites in yeast and mammals. For example, Atg4 has been reported to be phosphorylated by Atg1/ULK1 in both yeast and mammals. However, Atg4 is phosphorylated by Atg1 at Ser307 in yeast cells (Sanchez-Wandelmer et al, 2017), while in mammals, Atg4 is phosphorylated at the Ser316 residue by ULK1 (Pengo et al, 2017). Although the phosphorylation site of Atg4 by Atg1/ULK1 differs between yeast and mammals, the phosphorylation of Atg4 by Atg1/ULK1 has a similar function, inhibiting its catalytic activity and promoting the growth of the phagophore into an autophagosome. Similarly, although the phosphorylation sites of CCT2 by Atg1/ULK differ in yeast and mammals, the phosphorylation of CCT2 by Atg1/ULK1 initiates solid aggrephagy by regulating the association of CCT2 with Atg8/LC3C. Additionally, previous studies have reported that Atg1 substrates have specific phosphorylation motifs, although we cannot exclude the possibility that Atg1 phosphorylates Cct2 with non-classical motifs. However, our in vitro kinase and CCT2 $^{S412A-S470A}$ mutant experiments confirm that S412 and S470 are indeed the two primary phosphorylation sites on Cct2 by Atg1.

In this work, we revealed that Atg1-mediated Cct2 phosphorylation and Atg11-Cct2 binding are required for the initiation of solid aggrephagy via regulating the interaction between Atg8 and Cct2. This discovery expands the mechanistic understanding of Cct2-mediated solid aggrephagy. Despite these advances, it remains uncertain whether there are other post-translational modifications in Cct2 that also affect its activity in solid aggrephagy, or whether it is a substrate of other protein kinases. Our MS results showed that Cct2 has other phosphorylation sites and may undergo acetylation or methylation modifications. Our future and ongoing studies will further illustrate the functions of post-translational modification in driving Cct2-mediated solid autophagy.

# Methods

## Reagents and tools table

| Reagent/resource | Reference or source | Identifier or catalog number |
|---|---|---|
| **Experimental models** | | |
| BY4741 (*S. cerevisiae*) | Lab stock | |
| cct1-ts (*S. cerevisiae*) | Prof. Philip Hieter | |
| cct4-ts (*S. cerevisiae*) | Prof. Philip Hieter | |
| DH5α (*E. coli*) | Tsingke | TSC-C14 |
| BL21 (*E. coli*) | Tsingke | TSC-E06 |
| HEK-293 cells (*H. sapiens*) | ATCC | CRL-1573 |
| Hela cells (*H. sapiens*) | Prof. Wei Liu | |
| FIP200 KO Hela cells (*H. sapiens*) | Prof. Wei Liu | |
| WIPI2 KO Hela cells (*H. sapiens*) | Prof. Wei Liu | |
| MEF cells (*M. musculus*) | Prof. Min Li | |
| ULK1 KO MEF cells (*M. musculus*) | Prof. Min Li | |
| **Recombinant DNA** | | |
| pGEX-4T-1-Cct1 | This work | |
| pGEX-4T-1-Cct2 | Lab stock | |
| pGEX-4T-1-Cct4 | This work | |
| pGEX-4T-1-Cct2 D473A | This work | |
| pGEX-4T-1-Cct2 I487E | This work | |
| pGEX-4T-1-Cct2 mVLL | Lab stock | |
| pGEX-4T-1-Cct2 S412A | This work | |
| pGEX-4T-1-Cct2 S470A | This work | |
| pGEX-4T-1-Cct2 S412A-S470A(2A) | This work | |
| pGEX-4T-1-CCT2 | This work | |
| pGEX-4T-1-CCT2 S458A | This work | |
| pGEX-4T-1-LC3 | This work | |
| pCold-TF-CCT2 | This work | |
| pRS313-Ape1 | Lab stock | |
| pRS313-Ape1 P22L | Lab stock | |
| pRS315-Cct2-GFP | This work | |
| pRS315- Vph1-GFP | This work | |
| pRS315-Pro$_{GPD1}$-3×HA-Cct2 | Lab stock | |
| pRS315-Pro$_{GPD1}$-3×HA-Cct2 D473A | This work | |
| pRS315-Pro$_{GPD1}$-3×HA-Cct2 I487E | This work | |
| pRS315-Pro$_{GPD1}$-3×HA-Cct2 S412 A | This work | |
| pRS315-Pro$_{GPD1}$-3×HA-Cct2 S470A | This work | |
| pRS315-Pro$_{GPD1}$-3×HA-Cct2 S412A-S470A(2A) | This work | |
| pRS315-Pro$_{GPD1}$-3×HA-Cct1 | This work | |
| pRS315-Pro$_{GPD1}$-3×HA-Cct4 | This work | |
| pRS316-HA-Atg11 | Lab stock | |
| pRS316-Pro$_{Cup1}$-GFP-47Q | Lab stock | |

| Reagent/resource | Reference or source | Identifier or catalog number |
|---|---|---|
| pRS316-3×FLAG-Atg8 | Lab stock | |
| pRS313-RFP-Ape1 | Lab stock | |
| pRS313-RFP-Ape1 P22L | This work | |
| pRS316-GFP-Atg8 | Lab stock | |
| SOD1 G93A-EGFPN1 | Prof. Liang Ge | |
| SOD1-EGFPN1 | This work | |
| pCI-α-Synuclein A53T-EGFPN1 | Prof. Zhen Zhong | |
| pCI-α-Synuclein-EGFPN1 | Prof. Zhen Zhong | |
| FUGW-HA-CCT2 | Prof. Liang Ge | |
| FUGW-HA-CCT2 mVL(I)L | Prof. Liang Ge | |
| FUGW-HA-CCT2 S458A | This work | |
| pcDNA5-HA-ULK1 | Prof. Qiming Sun | |
| pcDNA5-HA-FIP200 | Prof. Qiming Sun | |
| pcDNA5-FLAG-ULK1 | This work | |
| pcDNA5-FRT-FLAG-LC3C | This work | |
| FUGW-FLAG-CCT2 | This work | |
| **Antibodies** | | |
| Rabbit anti-Pgk1 | Nordic Immunology | NE130/7S |
| Mouse anti-GFP | Roche | 11814460001 |
| Mouse anti-FLAG | Sigma | F1804 |
| Mouse anti-HA | Abmart | M20003L |
| Rabbit anti-Ape1 | GL Biochem Ltd., Shanghai | |
| Rabbit anti-thiophosphate ester | Abcam | ab92570 |
| Mouse anti-α-tubulin | EASYBIO | BE4067 |
| Mouse anti-β-actin | Biodragon | B1029 |
| Mouse anti-GAPDH | Proteintech | 60004-1-Ig |
| Rabbit anti-WIPI2 | Cell Signaling Technologies | 8567S |
| Rabbit anti-ULK1 | Cell Signaling Technologies | 4776S |
| Rabbit anti-FIP200 | Cell Signaling Technologies | 12436S |
| goat anti-mouse IgG1, human ads-HRP | SouthernBiotech | 1070-05 |
| goat anti-rabbit, human ads-HRP | SouthernBiotech | 4010-05 |
| **Chemicals, enzymes and other reagents** | | |
| Endonuclease Dpn1 | LABLEAD | F5585S |
| Endonuclease XbaI | LABLEAD | F5581S |
| Endonuclease SalI | LABLEAD | F5566S |
| Endonuclease SacI | LABLEAD | F5565S |
| Endonuclease NdeI | LABLEAD | F5551S |
| Endonuclease HindIII | LABLEAD | F5539S |
| Taq Master Mix | Vazyme | P222-01 |
| Super-Fidelity DNA Polymerase | Vazyme | P521-d1 |

| Reagent/resource | Reference or source | Identifier or catalog number |
|---|---|---|
| Yeast nitrogen base (without amino acids and ammonium sulfate) | BD | 233510 |
| Hygromycin B | YEASEN | 60225ES10 |
| G418 | YEASEN | 60220ES08 |
| Primary antibody dilution buffer | Beyotime | P0023A |
| Fetal Bovine Serum | VISTECH | SE200-ES |
| EBSS medium | Solarbio | H2025 |
| DMEM | Gibco | 12100061 |
| Trypsin | YEASEN | 40101ES25 |
| Penicillin-Streptomycin | Beyotime | ST488S |
| Indole-3-acetic acid (IAA) | Sigma | I2886 |
| **Software** | | |
| GraphPad Prism 9 | https://www.graphpad.com | |
| ImageJ | https://imagej.nih.gov/ij/index.html | |
| **Other** | | |
| Inverted fluorescence microscope | Olympus | IX83 |
| Chemiluminescent imaging system | Beijing Sage Creation Science Co.,Ltd | MiniChemi 910 Plus |

## Yeast strains and growth conditions

The yeast strains used in this study were confirmed via polymerase chain reaction (PCR) (Vazyme, P505-d1) or western blot analysis with specific antibodies, as listed in Table EV1 (Janke et al, 2004). Western blot detection was performed using the MiniChemi Chemiluminescence imager (SAGECREATION, Beijing). Additionally, all plasmids utilized were verified either through sequencing or western blotting. Yeast cells were cultured at 30 °C in either YPD medium (1% yeast extract, 2% glucose, and 2% peptone) or synthetic medium (SD; 0.17% yeast nitrogen base without amino acids and ammonium sulfate, 0.5% ammonium sulfate, 2% glucose, and the corresponding auxotrophic amino acids and vitamins). For induction of autophagy, cells were first grown to mid-log phase in YPD or SD medium and were subsequently transferred to nitrogen starvation medium (SD − N; 0.17% yeast nitrogen base without amino acids and ammonium sulfate, 2% glucose).

## Cell culture, transfection, and co-immunoprecipitation

HeLa, HEK293T, and MEF cells were cultured in Dulbecco's modified Eagle's (DMEM) medium with 5% $CO_2$ at 37 °C, supplemented with 10% fetal bovine serum and 1% penicillin/streptomycin. Upon reaching a cell density of ~80%, the cells were transfected with PEI. Following transfection with the corresponding plasmids for 24 h, the cells were collected, and lysis buffer (20 mM Tris-HCl pH 7.5, 150 mM NaCl, 1 mM EDTA, 0.5% NP40)

with protease inhibitors was added. The sample was kept on ice for 30 min, and the lysate was separated by centrifugation. The resulting supernatant was incubated with the designated agarose while rotating at 4 °C for 4 h. After the incubation period, the agarose was washed three times for 10 min each with lysis buffer. Following the wash, 2×SDS loading buffer was added to the agarose and then boiled at 95 °C for 5 min for SDS-PAGE detection.

## CHX chase assay

Cells were transfected with the designated plasmids and incubated for the indicated times. Following transfection, cells were treated with 50 mg/ml cycloheximide (CHX) and collected at each specified time point for immunoblot analysis. To detect the aggregated (non-soluble) form of the mutant proteins, cells were permeabilized with 40 mg/ml digitonin diluted in PBS on ice for 5 min and then washed with PBS before being collected for immunoblot analysis.

## Fluorescence recovery after photobleaching (FRAP)

FRAP experiments were performed using a Nikon C2 confocal microscope. Yeast cells expressing RFP-Ape1 p22L were bleached for 110 s at 561 nm with 100% laser intensity during the logarithmic growth phase. For GFP-47Q yeast cells, $Cu^{2+}$ was added to induce solid aggregate formation over 12 h, followed by bleaching for 20 s at 488 nm with 100% laser intensity. Recovery was recorded for the specified durations, and the fluorescence intensity of the photobleached area was normalized to the intensity of the unbleached area.

## Microscopy, western blots, and immunoprecipitation

Yeast strains expressing the specified fluorescent tag were cultured to the early log phase in a nutrient-rich medium. Cellular imaging was conducted using an inverted fluorescence microscope (IX83; Olympus). Yeast protein extraction, immunoblotting, and immunoprecipitation experiments were performed according to previously established protocols (Yao et al, 2023a).

## Auxin treatment

Yeast strains expressing the indicated proteins fused with the AID tag were grown to the early logarithmic phase. They were then treated with 0.5 mM IAA (indole-3-acetic acid; Sigma, I2886) for 2 h. The degradation of Cct2 was detected using an anti-HA antibody. An equal amount of dimethyl sulfoxide (DMSO) was added to the corresponding strains as a control.

## In vitro GST or Ni-NTA pulldown assay

For the in vitro Ni-NTA or GST pulldown assay, 5 nM of the purified protein with either a His or GST tag was combined with 10 μl Ni-NTA or GST beads in 500 μl of binding buffer A. The binding buffer A contained 50 mM Tris.HCl pH 7.5, 0.5 M NaCl, 1 mM DTT, 1% Triton X-100, 1% PMSF, and 20 mM iminazole (excluding iminazole for GST-tagged proteins). This mixture was incubated on a rotor at 4 °C for 2 h. Subsequently, the beads were washed once with binding buffer A and twice with binding buffer B, which consisted of 1 mM EDTA, 1 mM EGTA, 150 mM NaCl, and

1% Triton X-100 in 1×PBS. Next, the beads were exposed to Atg11 CC4 or its variants in binding buffer B for 3 h on a rotor at 4 °C. Following three washes with binding buffer B, 50 μl of 2×SDS loading buffer was added. The proteins separated by SDS-PAGE were then stained using Coomassie brilliant blue dye.

## In vitro phosphorylation assay and alkylation protocol

$atg1\Delta$ $ape1\Delta$ cells expressing either wild-type (WT) or kinase-dead (KD) Atg1-3×FLAG plasmids with Ape1 P22L were grown to the log phase under nutrient-rich conditions. Subsequently, 50 $OD_{600}$ yeast cells were lysed, and the resulting supernatant was subjected to immunoprecipitation using anti-FLAG agarose beads (Sigma, A2220). The purified WT or KD Atg1-3×FLAG was incubated with 5 μg of purified proteins from *E. coli* and 1.5 μl of 10 mM ATP-γ-S (Sigma, A1388) in Atg1 kinase buffer (30 mM HEPES·KOH pH 7.4, 5 mM NaF, 10 mM $MgCl_2$, 1 mM DTT) at 30 °C for 30 min. Following this, 2 μl of 50 mM p-nitrobenzyl mesylate (PNBM) (Abcam, ab138910) was added and allowed to react for an additional hour at 30 °C. The reaction was terminated by boiling in protein loading buffer for 5 min. Finally, the phosphorylation level of substrates was evaluated using an anti-thiophosphate ester antibody (Abcam, ab92570).

## In vitro binding assays of phosphorylated GST-Cct2 WT/ 2A with His-Atg8/Atg11 CC4

$atg1\Delta$ $ape1\Delta$ cells expressing either WT or KD Atg1-3×FLAG with Ape1 P22L were grown to the log phase under nutrient-rich conditions. Following this, 50 $OD_{600}$ yeast cells were harvested, and cell lysates were immunoprecipitated using anti-FLAG agarose beads (Sigma, A2220). The enriched WT or KD Atg1-3×FLAG was then incubated with 5 μg of purified GST-Cct2 (either WT or 2 A variant from *E. coli*) and 10 μM ATP in Atg1 kinase buffer (30 mM HEPES-KOH pH 7.4, 5 mM NaF, 10 mM MgCl2, 1 mM DTT) at 30 °C for 30 min. Next, the reaction product was transferred into a new EP tube. GST beads and 500 μl of binding buffer (50 mM Tris pH 7.5, 0.5 M NaCl, 1 mM DTT, 1% Triton X-100, 1% PMSF) were added, and the mixture was incubated at 4 °C for 1 h. The supernatant was then discarded, and 10 mM purified His-Atg8/ Atg11 CC4 from *E. coli* and 500 μl of interaction buffer (1 mM EDTA, 1 mM EGTA, 150 mM NaCl, 1% Triton X-100 in 1×PBS) were added and allowed to react for 2 h on a rotor at 4 °C. Finally, after three washes with interaction buffer, 50 μl of 2× SDS loading buffer was added to the beads. The proteins were separated by SDS-PAGE and stained with Coomassie Brilliant Blue dye.

## Mass spectrometric analysis

For LC–MS/MS analysis, peptide separation was performed over a 65-min gradient elution at a flow rate of 0.300 μl/min using a Thermo EASY-nLC1200 integrated nano-HPLC system, directly connected to a Thermo Q Exactive HF-X mass spectrometer. The analytical column was a custom-made fused silica capillary column (75 μm ID, 150 mm length; Upchurch, Oak Harbor, WA) packed with C-18 resin (300 Å, 3 μm; Varian, Lexington, MA). Mobile phase A consisted of 0.1% formic acid, while mobile phase B was 100% acetonitrile with 0.1% formic acid. The mass spectrometer operated in data-dependent acquisition mode, utilizing Xcalibur

4.1 software to acquire a single full-scan mass spectrum in the Orbitrap (ranging from 400 to 1800 $m/z$, with a resolution of 60,000), followed by 20 data-dependent MS/MS scans using 30% normalized collision energy. Each mass spectrometry result was analyzed with Thermo Xcalibur Qual Browser and Proteome Discoverer for subsequent database searching.

## Quantitative and statistical analysis

No blinding was done in this study. Western blot images from three independent experiments were analyzed and quantified using ImageJ software. Statistical analyses were performed using Graph-Pad Prism 9 software (GraphPad Software Inc.). The maturation of Ape1 was determined by calculating the ratio of the band intensities of mature Ape1 (mApe1) to the total Ape1 (mApe1 + precursor Ape1, PrApe1). To assess the degradation rate of the fused protein, GFP intensities were normalized by dividing by the sum of GFP and GFP-47Q. All quantitative analyses are presented as mean values with standard deviations (SD) shown as error bars. Statistical significance was evaluated using two-tailed Student's t-tests and is presented as the mean ± SD, based on the results of three independent experiments. The levels of significance are indicated as follows: \*\*\*$P < 0.001$; \*\*$P < 0.01$; \*$P < 0.05$; and ns (not significant).

# Data availability

The mass spectrometry proteomics data have been deposited to the ProteomeXchange Consortium via the PRIDE repository (ftp://massive.ucsd.edu/v08/MSV000095858/) with the dataset identifier MassIVE MSV000095858.

The source data of this paper are collected in the following database record: biostudies:S-SCDT-10_1038-S44319-024-00275-7.

# Peer review information

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

## Acknowledgements

We would like to thank Prof. Wei Liu (Zhejiang University) for providing
FIP200 KO and WIPI2 KO Hela cells, Prof. Min Li (Sun Yat-sen University) for
providing ULK1 KO MEF cells, and Prof. Philip Hieter (University of British
Columbia) for providing *cct1-ts* and *cct4-ts* yeast strains. We also thank Dr.
Jiayin Zhang and Prof. Peidong Han from Zhejiang University School of
Medicine for FRAP analysis, Dr. Cheng Ma from the Core Facilities, Zhejiang
University School of Medicine for their technical support, and the Mass
Spectrometry & Metabolomics Core Facility at the Center for Biomedical
Research Core Facilities of Westlake University for sample analysis. This
research was supported by the National Key Research and Development
Program of China (2021YFC2600104) to Liqin Zhang; the National Natural
Science Foundation of China (Grant No: 32100600) to Weijing Yao; and the
National Natural Science Foundation of China (32122028, 92254307, and
32070739) and Zhejiang Provincial Natural Science Foundation of China
(Grant No: LR21C070001) to Cong Yi.

## Author contributions

**Yuting Chen**: Conceptualization; Resources; Data curation; Software; Formal
analysis; Validation; Investigation; Visualization; Methodology; Writing—
original draft. **Zhaojie Liu**: Conceptualization; Resources; Data curation;
Software; Formal analysis; Validation; Investigation; Visualization;
Methodology; Writing—original draft. **Yi Zhang**: Conceptualization; Resources;
Data curation; Software; Formal analysis; Validation; Investigation;
Visualization; Methodology; Writing—original draft. **Miao Ye**: Resources; Data
curation; Software; Formal analysis; Validation; Visualization; Methodology.
**Yingcong Chen**: Resources; Data curation; Software; Formal analysis;
Validation; Visualization; Methodology. **Jianhua Gao**: Resources; Software;
Methodology. **Juan Song**: Resources; Software; Validation; Visualization;
Methodology. **Huan Yang**: Resources; Software; Validation; Visualization.
**Choufei Wu**: Resources; Software; Methodology. **Weijing Yao**: Data curation;
Supervision; Funding acquisition; Methodology. **Xue Bai**: Resources;
Methodology. **Mingzhu Fan**: Resources; Methodology. **Shan Feng**: Resources;
Methodology. **Yigang Wang**: Resources; Supervision; Methodology. **Liqin
Zhang**: Resources; Funding acquisition; Methodology. **Liang Ge**: Resources;
Methodology. **Du Feng**: Conceptualization; Resources; Supervision;
Investigation; Project administration; Writing—review and editing. **Cong Yi**:
Conceptualization; Resources; Formal analysis; Supervision; Funding
acquisition; Investigation; Writing—original draft; Project administration;
Writing—review and editing.

Source data underlying figure panels in this paper may have individual
authorship assigned. Where available, figure panel/source data authorship is
listed in the following database record: biostudies:S-SCDT-10_1038-S44319-
024-00275-7.

## Disclosure and competing interests statement

The authors declare no competing interests.

# Expanded View Figures

**Figure EV1.** **(corresponding to Fig. 1). Cct2 specifically binds to Atg8 and is degraded by autophagy in yeast.**

(A) BY4741, *cct1-ts*, or *cct4-ts* yeast cells expressing Atg11-3×FLAG were grown to log phase at 30 °C and then subjected to 37 °C for 2 h. Samples were analyzed by immunoblot for detecting the expression of Atg11-3×FLAG and α-Tubulin. Pgk1 served as a loading control. The data are representative of three independent experiments. (B) GST pulldowns were performed by using purified GST, GST-Cct1, GST-Cct2, or GST-Cct4 with His-Atg8 protein from *E. Coli*. The data are representative of three independent experiments. (C) *ape1Δ* cells co-expressing HA-Cct1, HA-Cct2, HA-Cct4, HA-Cct5, or HA-Cct8 with FLAG-Atg8 and Ape1 P22L were grown to the log phase under nutrient-rich conditions. Cell lysates were immunoprecipitated with anti-FLAG agarose beads and then analyzed by western blot using anti-HA antibody. The data are representative of three independent experiments. (D) *ape1Δ*, *ape1Δ atg8Δ*, *ape1Δ atg11Δ*, *ape1Δ atg15Δ*, *ape1Δ atg2Δ atg15Δ*, or *ape1Δ pep4Δ* cells co-expressing Cct2-GFP and Vph1-mCherry in the presence of Ape1 P22L were grown to $OD_{600} = 0.8$. These yeast strains were then cultured in nutrient-rich medium for 6 h. Images of cells were obtained using an inverted fluorescence microscope. Scale bar, 2 μm. (E) Cells from (D) were quantified for the vacuolar localization of Cct2-GFP. $n = 300$ cells were pooled from three independent experiments. Data are presented as means ± SD. ***$P < 0.001$; two-tailed Student's *t* tests were used. $P < 0.0001$(*atg8Δ* vs. WT); $P < 0.0001$(*atg11Δ* vs. WT); $P < 0.0001$(*atg15Δ* vs. WT); $P < 0.0001$(*atg2Δ*& *atg15Δ* vs. WT); $P < 0.0001$(*pep4Δ* vs. WT) in (E).

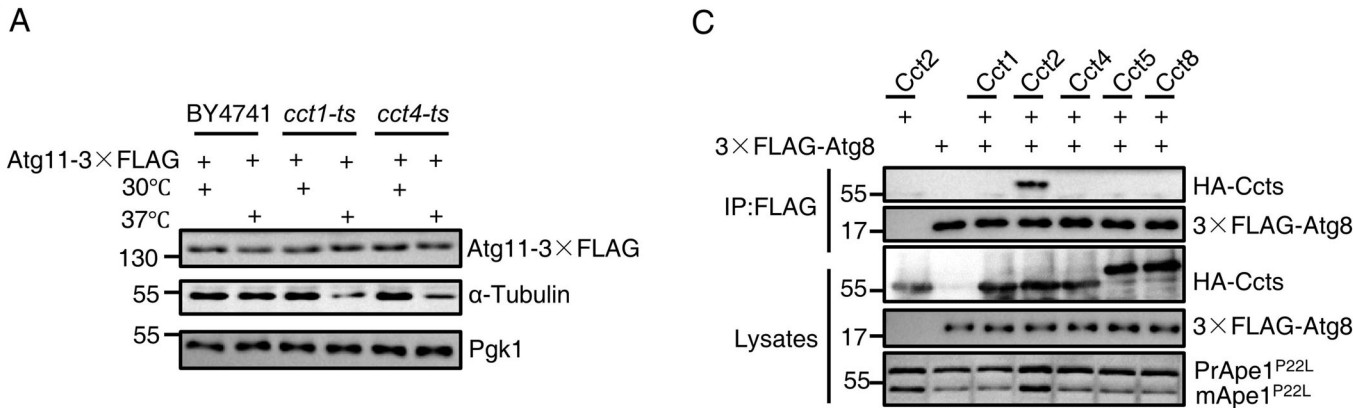

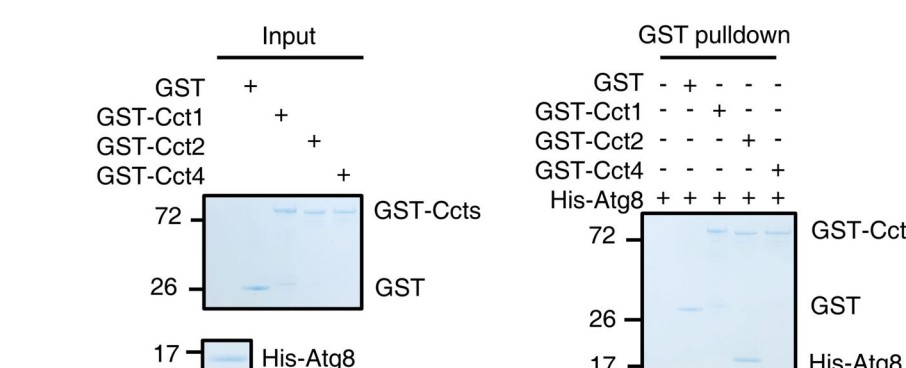

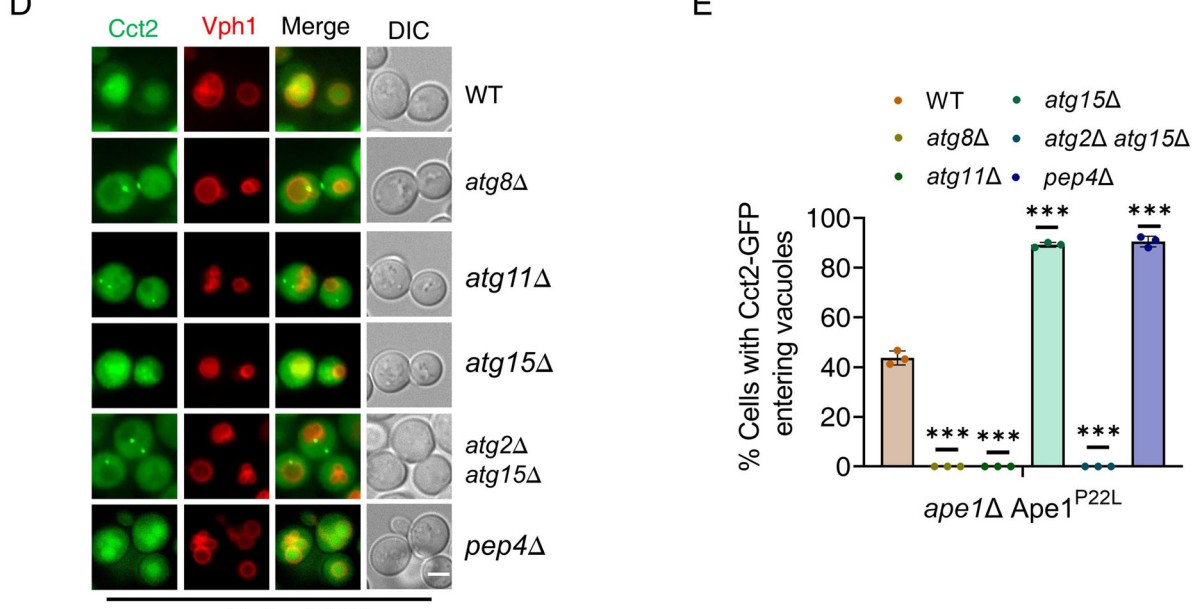

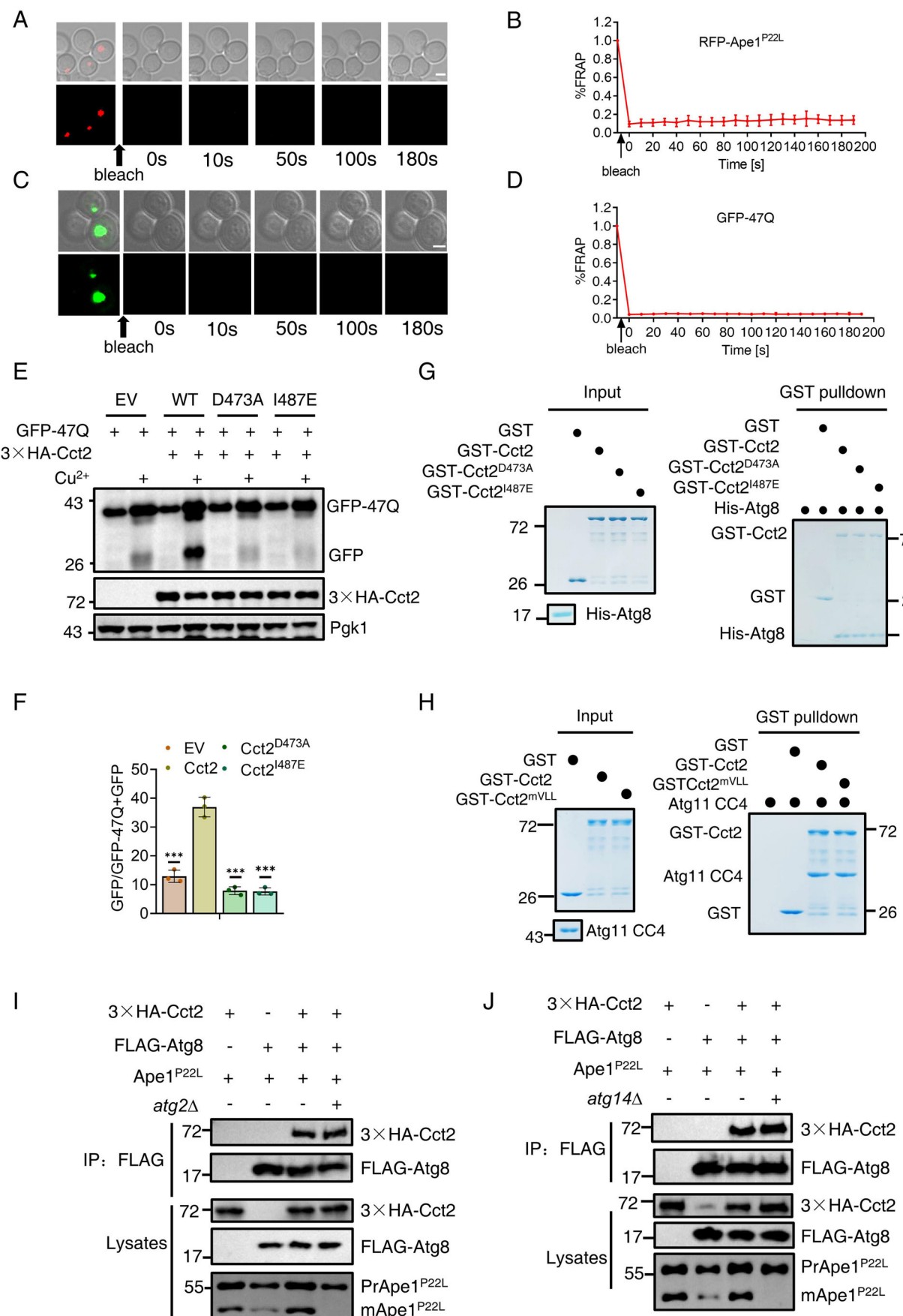

◀ **Figure EV2.** (corresponding to Fig. 3). **The binding of Cct2 to Atg11 is required for solid aggrephagy.**

(A) *ape1Δ* yeast cells expressing RFP-Ape1 P22L were grown to log phase. FRAP analysis was performed at the indicated time points. Scale bar, 2 μm. (B) Quantification of the normalized RFP-Ape1 P22L fluorescence signal (mean ± SD) in (A) (>30 cells from three independent experiments). (C) BY4741 yeast cells expressing $Cu^{2+}$-inducible GFP-47Q plasmids were cultured in nutrient-rich medium for 12 h to induce the formation of solid aggregates. FRAP analysis was performed at the indicated time points. Scale bar: 2 μm. (D) Quantification of the normalized GFP-47Q fluorescence signal (mean ± SD) in (C) (>30 cells from three independent experiments). (E) Yeast cells co-expressing $Cu^{2+}$-inducible GFP-47Q plasmids with an empty vector (EV), 3×HA-Cct2, or 3×HA-Cct2 variants were grown to an $OD_{600} = 0.6$. Subsequently, 0.1 mM $CuSO_4$ was added to the cells to induce GFP-47Q to form solid aggregates for 12 h. Samples were analyzed by immunoblot for the cleavage of GFP-47Q. Pgk1 served as a loading control. (F) Quantification of GFP to GFP-47Q + GFP ratio from (E). Data are presented as means ± SD (Data represent the results of three independent experiments). ***$P < 0.001$; two-tailed Student's *t* tests were used. $P = 0.0005$ (EV vs. Cct2); $P = 0.0002$ (D473A vs. Cct2); $P = 0.0002$ (I487E vs. Cct2). (G) In vitro GST pulldowns were performed using His-Atg8 with GST, GST-Cct2, GST-Cct2 D473A, or I487E purified from *E. coli*. Protein samples were separated by SDS-PAGE and detected using Coomassie blue staining. The data are representative of two independent experiments. (H) In vitro GST pulldowns were performed using Atg11 CC4 with GST, GST-Cct2, GST-Cct2 mVLL (AIM motif mutant) purified from *E. coli*. Protein samples were separated by SDS-PAGE and detected using Coomassie blue staining. The data are representative of two independent experiments. (I, J) *ape1Δ*, *atg2Δ ape1Δ*, or *atg14Δ ape1Δ* cells co-expressing 3×HA-Cct2 with FLAG-Atg8 in the presence of Ape1 P22L were grown to the log phase under nutrient-rich conditions. Cell lysates were immunoprecipitated with anti-FLAG agarose beads and then analyzed by western blot using anti-HA antibody. The data are representative of three independent experiments.

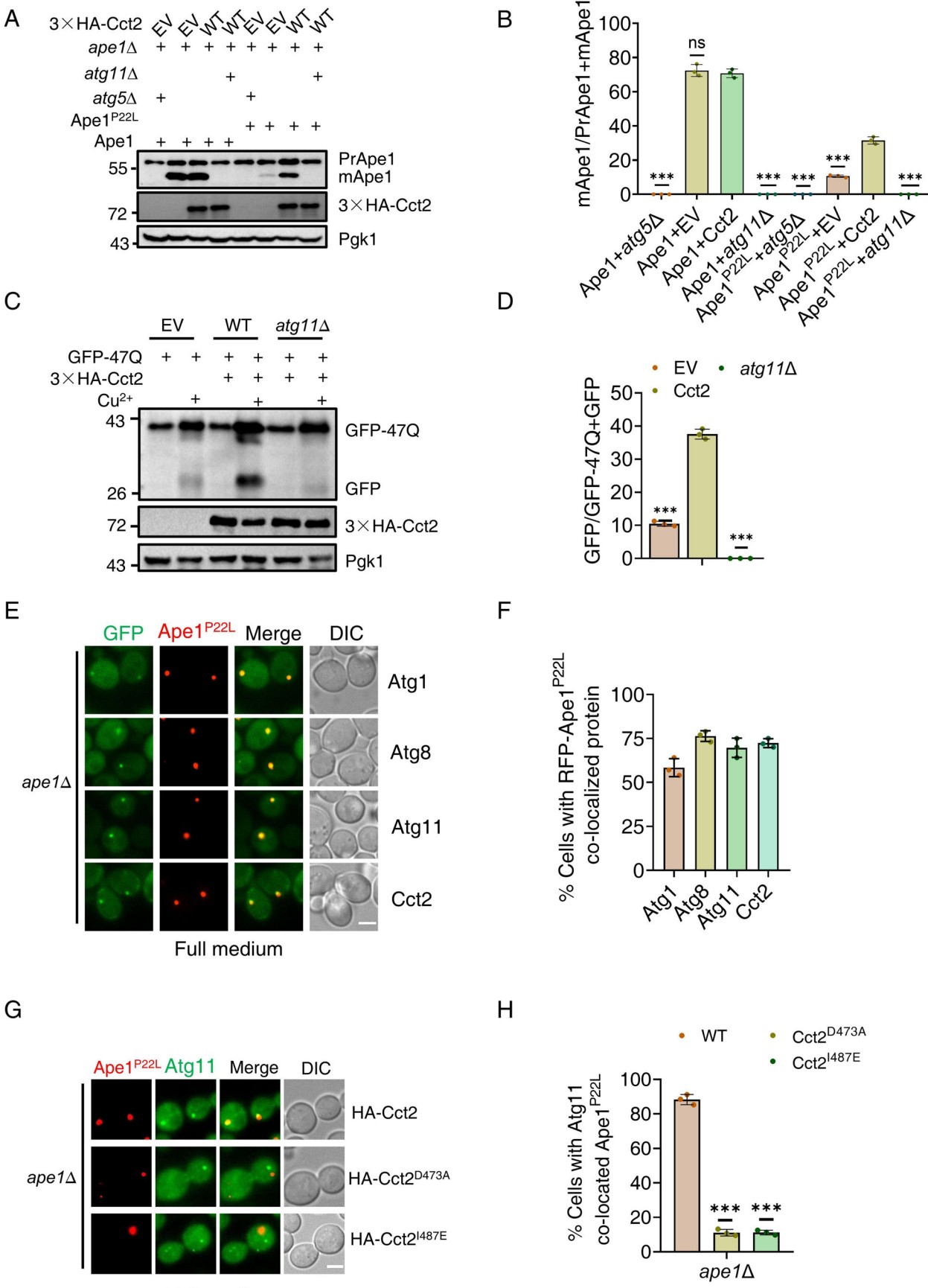

**Figure EV3.  (corresponding to Fig. 3). Atg11 is required for aggregates turnover and Ape1 P22L co-localizes with autophagy-related proteins and Cct2.**

(A) The indicated yeast cells were grown to the log phase in nutrient-rich medium. Samples were analyzed by immunoblot for detecting the maturation of PrApe1 into mApe1. Pgk1 served as a loading control. (B) Quantification of mApe1 to PrApe1+mApe1 ratio from (A). Data are presented as means ± SD (Data represent the results of three independent experiments). ***$P < 0.001$; NS, not significant; two-tailed Student's $t$ tests were used. $P < 0.0001$ (WT *atg5Δ* vs. WT Cct2); $P = 0.5412$ (WT EV vs. WT Cct2); $P < 0.0001$ (WT *atg11Δ* vs. WT Cct2); $P < 0.0001$ (P22L *atg5Δ* vs. P22L Cct2); $P < 0.0001$ (P22L EV vs. P22L Cct2); $P < 0.0001$ (P22L *atg11Δ* vs. P22L Cct2). (C) The indicated yeast cells were grown to an $OD_{600} = 0.6$. Subsequently, 0.1 mM $CuSO_4$ was added to the cells to induce GFP-47Q to form solid aggregates for 12 h. Samples were analyzed by immunoblot for the cleavage of GFP-47Q. Pgk1 served as a loading control. (D) Quantification of GFP to GFP-47Q + GFP ratio from (C). Data are presented as means ± SD (Data represent the results of three independent experiments). ***$P < 0.001$; two-tailed Student's $t$ tests were used. $P < 0.0001$ (EV vs. Cct2); $P < 0.0001$ (*atg11Δ* vs. Cct2). (E) *ape1Δ* yeast cells co-expressing RFP-Ape1 P22L with Atg1-2×GFP, GFP-Atg8, Atg11-2×GFP, or Cct2-GFP were grown to the log phase in nutrient-rich medium. Images of cells were obtained using an inverted fluorescence microscope. Scale bar, 2 μm. (F) Cells from (E) were quantified for the number of cells in which RFP-Ape1 P22L colocalized with the indicated GFP-fused proteins. $n = 300$ cells were pooled from three independent experiments. Data are shown as mean ± SD. (G) *ape1Δ* yeast cells co-expressing RFP-Ape1 P22L and Atg11-2×GFP with 3×HA-Cct2, 3×HA-Cct2 D473A, or 3×HA-Cct2 I487E were grown to the log phase in nutrient-rich medium. Images of cells were obtained using an inverted fluorescence microscope. Scale bar, 2 μm. (H) Cells from (G) were quantified for the number of cells in which RFP-Ape1 P22L colocalized with Atg11-2×GFP. $n = 300$ cells were pooled from three independent experiments. Data are shown as mean ± SD. ***$P < 0.001$; two-tailed Student's $t$ tests were used. $P < 0.0001$ (D473A vs. WT); $P < 0.0001$ (I487E vs. WT).

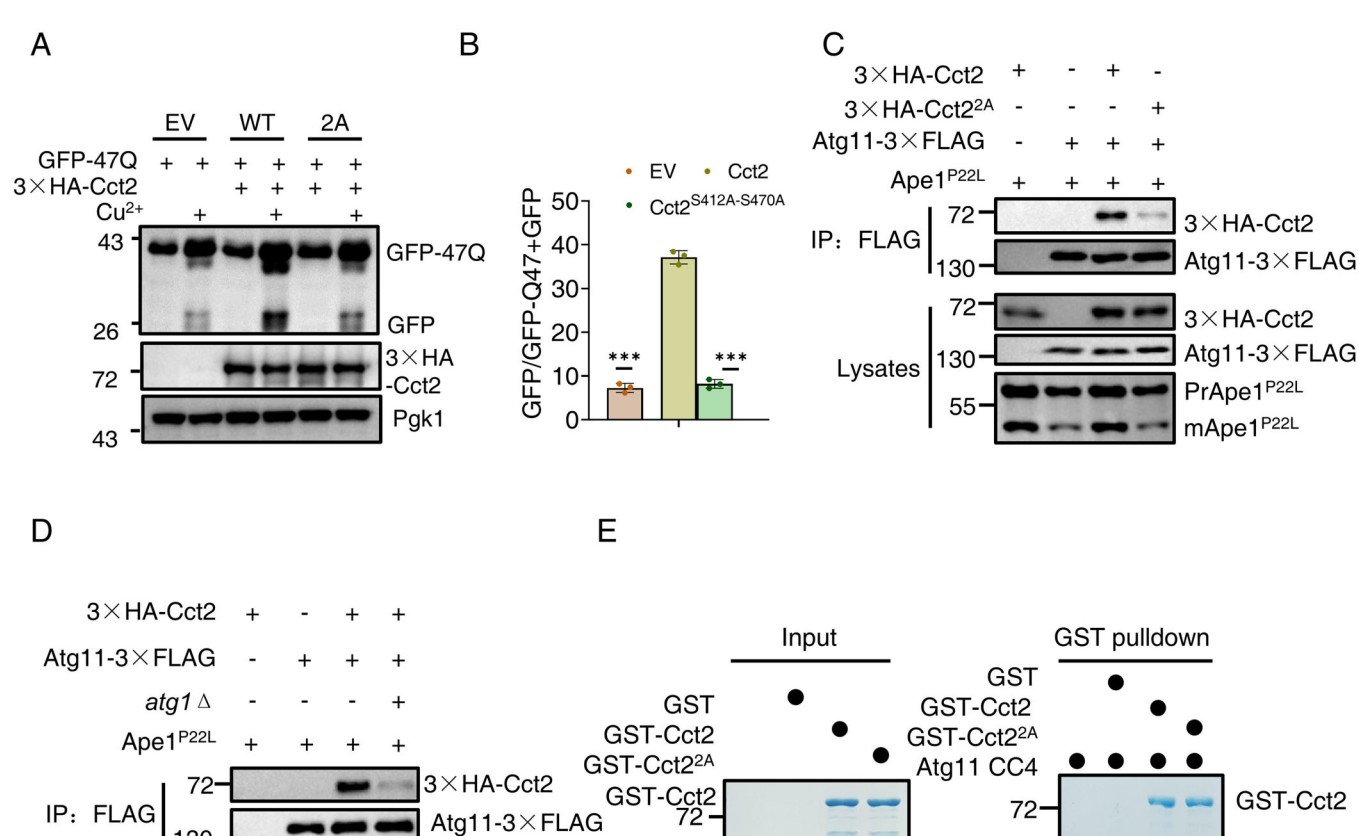

◀ **Figure EV4.   (corresponding to Fig. 5). The phosphorylation of Cct2 by Atg1 is required for the cleavage of GFP-47Q and the binding of Cct2-Atg11.**

(A) Yeast cells co-expressing $Cu^{2+}$-inducible GFP-47Q plasmids with an empty vector (EV), 3×HA-Cct2, or Cct2 [S412A-S470A] (2A) were grown to an $OD_{600} = 0.6$. Subsequently, 0.1 mM $CuSO_4$ was added to the cells to induce GFP-47Q to form solid aggregates over 12 h. Samples were analyzed by immunoblot for the cleavage of GFP-47Q. Pgk1 served as a loading control. (B) Quantification of GFP to GFP-47Q + GFP ratio from (A). Data are presented as means ± SD (Data represent the results of three independent experiments). ***$P < 0.001$; two-tailed Student's $t$ tests were used. $P < 0.0001$ (EV vs. Cct2); $P < 0.0001$ (S412A-S470A vs. Cct2). (C) *ape1Δ* cells co-expressing an empty vector, 3×HA-Cct2, or 3×HA-Cct2 [S412A-S470A] with Atg11-3×FLAG in the presence of Ape1 P22L were grown to the log phase under nutrient-rich medium. Cell lysates were immunoprecipitated with anti-FLAG agarose beads and then analyzed by western blot using anti-HA antibody. The data are representative of three independent experiments. (D) *ape1Δ* or *ape1Δ atg1Δ* cells co-expressing 3×HA-Cct2 with Atg11-3×FLAG in the presence of Ape1 P22L were grown to the log phase under nutrient-rich conditions. Cell lysates were immunoprecipitated with anti-FLAG agarose beads and then analyzed by western blot using anti-HA antibody. The data are representative of three independent experiments. (E) In vitro GST pulldowns were performed using Atg11 CC4 domain with GST, GST-Cct2, or GST-Cct2 [S412A-S470A] purified from *E. coli*. Protein samples were separated by SDS-PAGE and detected using Coomassie blue staining. The data are representative of two independent experiments. (F) In vitro phosphorylation assays were performed using GST-Cct2 WT or 2A(S412A-S470A) purified from *E. coli* as substrates, with WT or KD Atg1-3×FLAG, purified from yeast cells, in the presence of Ape1 P22L under nutrient-rich conditions as the protein kinase. After that, in vitro GST pulldowns were performed using phosphorylated products enriched by GST beads in vitro with Atg11 CC4 protein purified from *E. coli*. Protein samples were separated by SDS-PAGE, and then detected using Coomassie blue staining. The data are representative of three independent experiments. (G) In vitro phosphorylation assays were performed using GST-Cct2 WT or 2A(S412A-S470A) purified from *E. coli* as substrates, with WT or KD Atg1-3×FLAG, purified from yeast cells, in the presence of Ape1 P22L under nutrient-rich conditions as the protein kinase. After that, in vitro GST pulldowns were performed using phosphorylated products enriched by GST beads in vitro with the same amount of His-Atg8 protein and Atg11 CC4 protein purified from *E. coli*. Protein samples were separated by SDS-PAGE, and then detected using Coomassie blue staining. The data are representative of three independent experiments.

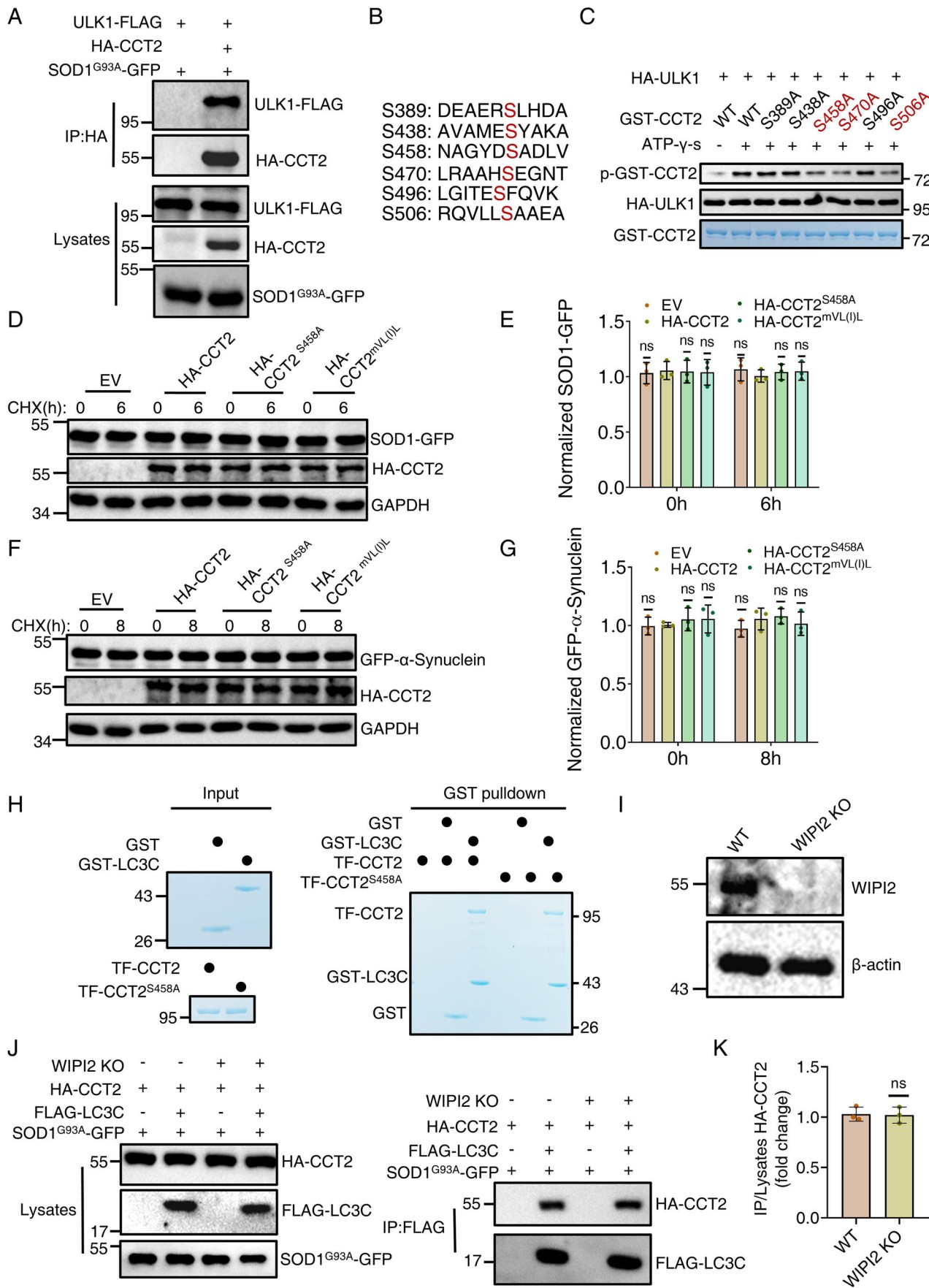

◀ **Figure EV5.  (corresponding to Fig. 6). CCT2 is a substrate of ULK1 and WIPI2 does not participate in the binding of LC3C-CCT2 in mammalian cells.**

(A) HEK293T cell lines were co-transfected with HA-CCT2, ULK1-FLAG, and SOD1 G93A-GFP. Cell lysates were immunoprecipitated with anti-HA agarose beads and then analyzed by western blot using an anti-FLAG antibody. The data are representative of three independent experiments. (B) Six typical ULK1 phosphorylation sites on GST-CCT2 D3 domain. (C) An in vitro kinase assays were performed using GST, GST-hCCT2, or six indicated GST-hCCT2 variants was purified from *E. coli* as substrates and purified HA-ULK1 from SOD1-G93A-GFP expressed HEK293T cell in nutrient-rich medium as protein kinase. The phosphorylation of Cct2 and these variants were detected by anti-thioP antibody. The data are representative of two independent experiments. (D) Turnover of SOD1-GFP in CHX chase assay with empty vector (EV), HA-CCT2, HA-CCT2 $^{S458A}$, or HA-CCT2-mVL(I)L (LIR motif mutant) expression in HeLa cell lines at 24 h after transfection. The protein sample were analyzed by western blot. GAPDH served as a loading control. (E) Quantification of normalized SOD1-GFP from (D). Data are presented as means ± SD (Data represent the results of three independent experiments). NS, not significant. Two-tailed Student's $t$ tests were used. $P = 0.7644$ (EV 0 h vs. CCT2 0 h); $P = 0.9012$ (S458A 0 h vs. CCT2 0 h); $P = 0.8480$ (mVL(I)L 0 h vs. CCT2 0 h); $P = 0.4273$ (EV 6 h vs. CCT2 6 h); $P = 0.5161$ (S458A 6 h vs. CCT2 6 h); $P = 0.4892$ (mVL(I)L 6 h vs. CCT2 6 h) (F) Turnover of GFP-α-Synuclein in CHX chase assay with empty vector (EV), HA-CCT2, HA-CCT2 $^{S458A}$, or HA-CCT2-mVL(I)L (LIR motif mutant) expression in HeLa cell lines at 24 h after transfection. The protein sample were analyzed by western blot. GAPDH served as a loading control. (G) Quantification of normalized GFP-α-Synuclein (F). Data are presented as means ± SD (Data represent the results of three independent experiments). NS, not significant. Two-tailed Student's $t$ tests were used. $P = 0.8375$ (EV 0 h vs. CCT2 0 h); $P = 0.4615$ (S458A 0 h vs. CCT2 0 h); $P = 0.5141$ (mVL(I)L 0 h vs. CCT2 0 h); $P = 0.2845$ (EV 8 h vs. CCT2 8 h); $P = 0.7403$ (S458A 8 h vs. CCT2 8 h); $P = 0.6388$ (mVL(I)L 8 h vs. CCT2 8 h). (H) In vitro GST pulldowns were performed using GST-LC3C domain with TF-hCCT2 or TF-hCCT2$^{S458A}$ purified from *E. coli*. Protein samples were separated by SDS-PAGE and detected using Coomassie blue staining. The data are representative of two independent experiments. (I) Western blot validation of WIPI2 KO effect in Hela cell lines. The data are representative of two independent experiments. (J) WT or WIPI2 KO Hela cell lines were co-transfected with FLAG-LC3C, HA-CCT2, and SOD1 G93A-GFP. Cell lysates were immunoprecipitated with anti-FLAG agarose beads and then analyzed by western blot using an anti-HA antibody. (K) Quantification of IP/lysates HA-CCT2 from (J). Data are presented as means ± SD (Data represent the results of three independent experiments). NS, not significant; two-tailed Student's $t$ tests were used. $P = 0.8785$ (WIPI2 KO vs. WT).

