## [Peer Review File · EMBO Reports]

Two distinct regulatory pathways govern Cct2-Atg8 binding in the process of solid autophagy

Cong Yi, Yu Chen, Zhaojie Liu, Yi Zhang, Miao Ye, Ying Chen, Jianhua Gao, Juan Song, Huan Yang, Chou Wu, Wei Yao, Xue Bai, Mingzhu Fan, Shan Feng, Yigang Wang, Liqin Zhang, Liang Ge, and Du Feng

Corresponding author(s): Cong Yi (yiconglab@zju.edu.cn) , Du Feng (Fenglab@gzhmu.edu.cn)

Review Timeline:

Submission Date:	16th Jan 24
Editorial Decision:	4th Mar 24
Appeal Received:	4th Jul 24
Editorial Decision:	13th Aug 24
Revision Received:	15th Aug 24
Editorial Decision:	28th Aug 24
Revision Received:	30th Aug 24
Accepted:	13th Sep 24

Editor: *Martina Rembold*

Transaction Report:

Dear Prof. Yi

Thank you for the submission of your research manuscript to EMBO Reports. We have now received the enclosed reports on it. As you will see, the referees acknowledge that the findings are potentially interesting, but they also raise a number of important concerns that challenge the key findings of your study. As it stands, it remains unclear whether the interaction between Cct2 and Atg11/Atg8 is indeed regulated by Atg1-mediated phosphorylation, whether it is induced by the presence of aggregates rather than by starvation, whether it plays a role in aggrephagy rather than macroautophagy and whether Cct2 functions independently of the TriC complex in this context.

Due to the nature of the criticisms that question the key conclusions of your study, the amount of work likely to be required to address them, and the uncertain outcome of these experiments, I am afraid that we do not feel it would be productive to call for a revised version of your manuscript at this stage.

Given the potential interest of your findings, we would, however, have no objections to consider a resubmission of the manuscript in the future if you were able to address all main concerns of the reviewers as highlighted above and in their reports. I would like to stress though that such a manuscript would be treated as a new submission and would be evaluated again, also with respect to the literature and the novelty of your findings at the time of resubmission.

I apologize that I cannot be more positive at this point. I hope, however, that the referee comments are going to be helpful in strengthening your indeed very interesting initial observations and I will be happy to discuss any additional data on this topic with you in the future.

Yours sincerely

Referee #1:

The authors have previously reported that Cct2 acts as an autophagy receptor in degrading solid aggregates via autophagy and that it interacts directly with Atg8 both in yeast and mammals, and in vitro and in vivo (Ma et al., Cell 2022).

In this new work, the authors show that Cct2 also interacts directly with Atg11 and that this interaction depends on D473, D487, S412 and S470 in vivo.

Claimed is furthermore that this interaction requires phosphorylation of residues S412 and S470, that the responsible kinase is Atg1, and that this regulation is conserved in mammals. However, this claim on phosphoregulation lacks in vivo evidence. Also, the regulated assembly of the autophagy factors on aggregates is claimed but not shown.

Convincingly shown is that Cct2 directly interacts with Atg11 in vitro, and this interaction is convincingly confirmed in vivo. Not sufficiently addressed but claimed are the phosphoregulation by Atg1 and the assembly of Cct2, Atg11, Atg8 and Atg1 on aggregates. Without showing these later two points convincingly, the novelty of the work is limited and the study is not suited for publication in EMBO Reports.

Specific points:

1. Figure 3 and 4 and supplement: The authors claim that Atg1 kinase phosphorylates Cct2 and that this phosphorylation is required for Cct2 binding to Atg8 and Atg11 in vivo. However, the experiments are only done in vitro, and this claim lacks in vivo evidence. Several kinases share a similar consensus to Atg1, and many kinases are in vitro not very specific, as long as the consensus is more or less there. To claim that Atg1 is the kinase acting on Cct2, it needs to be shown:

- by mass spec that these sites are lost in an atg1-delta or atg1-kinase dead situation, and
- that an atg1-delta/kinase dead results in the loss of the Cct2-Atg8 and Cct2-Atg11 interactions.

2. All in vivo verification of Cct2 interactions with Atg11, Atg8 and Atg1 is done in a setup lacking aggregates, therefore only represents a general ability of these proteins to interact in the absence of cargo, which questions the significance of the findings for the proposed model. The authors, however, claim from these results an association of all these factors with aggregates. Such an association has not been shown and needs to be clarified by microscopy. To make this claim they need to show:

- that Cct2, Atg11, Atg8 and Atg1 colocalize with aggregates (either Q47 or Ape1P22L),
- that Cct2 colocalization with aggregates remains in atg11-delta cells, but Atg8 is lost, and
- that the Cct2-D473A or I487E mutant results in loss of Atg11 colocalization to the aggregates.

3. In figures 4 and S4 the authors show that the Cct2-2A mutant decreases Atg8 and Atg11 binding in vivo, however, when

assessing this interaction in vitro, there is no defect (Figure 4F and S4D). If phosphorylation and therefore the negative charge on these serine residues was essential to promote these interactions, then an alanine mutant should abolish this interaction also in vitro. The authors' argument that this in vitro - in vivo difference shows that the Cct2-2A structure is not impaired, does not make sense at all! It rather indicates that the DIRECT interaction between Cct2 and Atg11/Atg8 is not affected by these two residues, which speaks against their phosphoregulation model.

4. The mammalian part shows similar problems in the authors' claims as for the yeast part, e.g. if ULK1 is required needs to be shown by addressing the CCT2-LC3C interaction in an ULK1 knockdown, and the presence of S458 phosphorylation needs to be verified by mass spec to be lost in the absence of ULK1.

5. As solid aggregates the authors use Ape1-P22L and Q47. From their results, they claim that the turnover of these depends on Atg11 binding to Cct2. However, for both aggregates, they don't show that these aggregates fail to be turned over in an atg11-delta situation. They only did the experiments in a Cct2 point mutant, which might also lack binding to factors other than Atg11. Therefore, the aggregate degradation experiments need to be repeated in atg11-delta cells.

6. Fig S2A and S4A: How was vacuolar localization judged without a co-stain? The pictures shown don't allow a localization judgment of Ape1-P22L. As Ape1-P22L can get extremely big due to the strongly decreased Atg19 binding, which leads to higher oligomerization of Ape1 (Shintani et al. Dev Cell 2002, Bertipaglia et al. EMBO Rep 2016, Yamasaki et al. Mol Cell 2020), it could be misleading and a vacuolar co-stain should be used. As the effects observed by microscopy seem much more severe than the very mild effect found by western blots, this is essential to clarify.

7. Another general issue is that Cct2 is overexpressed in all experiments, and this is done on top of the endogenous Cct2, both for WT and mutants. Whether endogenous Cct2 is acting in aggregate turnover has neither been addressed in the previous work nor here. As Cct2 is essential, a temperature-sensitive mutant or degron system could be used to clarify this point. Otherwise, the authors cannot claim that Cct2 acts as a bona fide autophagy receptor, and need to state properly throughout the text that their observation "is enhanced aggregate turnover upon Cct2 overexpression."

8. Most western blots are not quantified, and only shown n=1. It needs to be stated if this has been repeated and if the same result has been observed (minimum n=2). Ideally, the repetitions could be shown in the supplement, which would make the rather mild effects more convincing.

Minor points:

9. The authors use inconsistent naming of CCT2, sometimes capital CCT2 (e.g. Fig 3G), sometimes Cct2 (e.g. Fig 3F). The latter is the yeast nomenclature, the previous is the mammalian. They should do this correctly throughout the manuscript, in the figures, the text and the legends.

10. Unclear is why the authors do some experiments under starvation or torin treatment, although they look at selective autophagy and most experiments seem to be done under nutrient-rich conditions. This should be explained. If effects are only observed under starvation and not under rich conditions, they need to rule out bulk autophagy in these experiments.

11. The authors state: "...an increase in the binding of CCT2 with Atg1 under nitrogen starvation conditions (Fig. 3A), suggesting the potential phosphorylation of CCT2 by Atg1 in response to stress." This argument is strange as kinases usually don't stably interact with their substrate, and should at least not be used to support the claim that Atg1 phosphorylates Cct2.

12. The authors use GFP-Q47 as a solid aggregate. They cite their previous paper for this, however, in their previous work, it has not been shown that GFP-Q47 is indeed solid. They need to cite the original literature for this if existent, or otherwise show it experimentally.

Referee #2:

The authors' groups previously discovered autophagic degradation of solid protein aggregates, which involves the chaperonine subunit CCT2 as a receptor in both yeast and mammals; CCT2 binds to both the aggregates and Atg8-family proteins anchored to the autophagosomal membrane and thereby mediates their sequestration into the autophagosome. In this study, the authors further addressed the regulation of this solid aggregate pathway. They first showed in yeast that similar to most of the yeast autophagy receptors, CCT2 interacts with Atg11 and that this interaction facilitates the CCT2-Atg8 interaction. In addition, they found that CCT2 is phosphorylated by the autophagy initiation kinase Atg1 and that this phosphorylation enhances the CCT2 interactions with Atg11 and Atg8. These interactions and phosphorylation of CCT2 were shown to be important for solid aggregate turnover. Finally, the authors showed that mammalian CCT2 is also phosphorylated by the Atg1 homolog ULK1, and this phosphorylation is important for the CCT2 interaction with the Atg8 homolog LC3C and thus for solid aggregate turnover. Thus this study reveals how solid aggregate turnover is regulated but the following issues should be addressed to strengthen the authors' conclusions.

Major comments:

1. The authors showed that the CCT2-Atg11 interaction facilitates the CCT2-Atg8 interaction, but the underpinning mechanism remains unknown. To assess whether the former interaction directly increases the latter, the author should perform in vitro pull-down assay for the CCT2-WT/CCT2-D473A/I487E-Atg8 interaction with or without Atg11. On the other hand, another possibility would be that autophagosome formation on solid aggregates, which requires the former interaction, promotes the latter via the production of Atg8-PE in the expanding phagophore. If so, deletion of core ATG genes such as ATG2 would also decrease the CCT2-Atg8 interaction in vivo even in the presence of Atg11.
2. The authors showed that Atg1-mediated phosphorylation of CCT2 promotes the CCT2 interactions with both Atg11 and Atg8. However, they insist that CCT2 phosphorylation increases its interaction with Atg8. Although without further dissecting the effect of CCT2 phosphorylation on these interactions; there remains the possibility that CCT2 phosphorylation by Atg1 enhances the CCT2-Atg11 interaction, which promotes the CCT2-Atg8 interaction. The authors should clarify this point. Does abolishing CCT2 phosphorylation decrease the CCT2-Atg8 interaction in the absence of Atg11 in vivo? How CCT2 phosphorylation by Atg1 affect the CCT2-Atg8 and CCT2-Atg11 interactions in in vitro pull-down assay?
3. In the authors' model, free CCT2 (not assembled into the chaperonine complex) acts as a solid aggregate receptor. Is free CCT2 more efficiently phosphorylated by Atg1 compared with that in the complex?
4. The authors should examine whether CCT2 interacts with FIP200 in mammalian cells and if this is the case how the interaction affects the CCT2-LC3C interaction.

Minor comments:

1. Page 6, lines 26-27, "We proceeded to... its interaction with Atg11": This sentence should be rewritten.
2. Page 8, lines 7-12, "Considering...with Atg8 (Fig. 2E).": This sentence should be rewritten.
3. Figure 2B: The graph labels should be corrected.
4. The authors should describe why they chose LC3C among several Atg8-family proteins in mammalian cells.

Referee #3:

CCT2 was recently discovered to be a solid aggregate receptor that functions independently of the chaperonin TCP-1 ring complex (TRiC). Selective autophagy receptors (SARs) are known to bind to members of the ATG machinery and thereby autophagosome formation is initiated around the cargo that is bound by the SARs. In this manuscript, the authors mostly investigated the role of Cct2 in a yeast model. They show that Cct2 is phosphorylated by Atg1 on 2 sites (Ser 412 and 470) and interacts with Atg11. The phosphorylation and subsequent Atg11 binding regulate the binding to Atg8. Furthermore, the authors showed that a similar mechanism is conserved in mammalian cells. The manuscript advances the knowledge of CCT2 as a selective autophagy receptor.

Nevertheless, there are three major concerns about the interpretation of the data (that will be further detailed below): First, the authors did not establish that CCT2 works independently from the TriC complex in yeast. Second, the authors claim to investigate solid aggregate, although the substrates they test are mostly soluble mutant proteins. Third, most of the interaction experiments were performed in the absence of a solid aggregate cargo and often even under starvation conditions. Therefore, the provided data does not allow to call the selective form of autophagy, observed here, solid aggregate or that the modulations of CCT2 happen during solid aggregate.

Major points:

Figure 1: Other subunits of the TriC (e.g., CCT1, 3 or 6) needs to be tested to show that CCT2 works independently (also in yeast).

Can the authors exclude that Atg11 is a substrate of the TriC?

Figure S2, S4: how did the authors determine the localization of the vacuole without staining? It needs to be indicated how the quantification was done (automatically or manual)

Figure 2A, B: the effect of the WT-CCT2 vs the mutants (or KO) seems rather marginal, especially given that for the WT-CCT2, the expression levels are higher compared to the mutants. Do the authors still detect this defect when the results are normalized against the CCT2 expression levels?

To show that CCT2 is a solid aggregate receptor in yeast (independently of TriC), the authors need to show that CCT2 is delivered to the vacuole and not the Atg11 binding mutants, and that deletions of other TriC subunits do not impair the delivery of ApeP22L.

Figure 2C, D: similar controls as suggested for figure 2A, B should also be tested here. The authors analyzing here yeast cells that have been starved for 1-2 days, thereby massively inducing bulk autophagy. How can the authors conclude that the effect is specific for selective autophagy? These experiments need to be repeated under non-starving conditions.

Figure 3: The authors claim that they have identified the two phosphorylation sites in CCT2 by Atg1 during solid aggregatephagy. This interpretation of these data bears multiple problems. First, the authors have not tested all the phosphorylation sites. They focused on the D3 region and did not consider the other 2 regions, where there was phosphorylation detected (albeit less strong). Second, in the dataset table S1 there are many more phosphorylation sites detected and the description why exactly the mentioned 2 phosphorylation sites are important is missing. Third, there is no evidence that Atg1 is phosphorylating CCT2 *in vivo* during solid aggregatephagy. On the contrary, the authors show that CCT2 is phosphorylated during starvation induced autophagy *in vivo*, not solid aggregatephagy. To prove that Atg1 is the responsible kinase *in vivo*, the CCT2 phosphorylation needs to be tested in the absence of Atg1. And to be able to claim that this phosphorylation happens during solid aggregatephagy specifically, the experiment must be done under non-starved conditions with an appropriately aggregated cargo.

Figure 4C, D: WT-CCT2, the expression levels of the WT-CCT2 are much higher compared to the mutants. Do the authors still detect this defect when the results are normalized against the CCT2 expression levels? The authors analyzing here yeast cells that have been starved for 1-2 days, thereby massively inducing bulk autophagy. How can the authors conclude that the effect is specific for selective autophagy? These experiments need to be repeated under non-starving conditions.

Figure 4E, S4C: The authors must not talk about solid aggregatephagy when the experiment was performed during starvation conditions. This is misleading. Moreover, in the experiments described in Figure 4, there is even no solid aggregatephagy cargo present showing that the interaction/phosphorylation occurs independently of the substrate. Therefore, to claim that the Atg1 mediated CCT2 phosphorylation is solid aggregatephagy specific, the authors need to show the selectivity only in the presence of a substrate and/or re-phrase their conclusion.

Figure 5D-G: For all the solid aggregatephagy substrates (Q47, FUS, alpha-syn, ...in yeast and HEK) the authors did not show that they analyse the aggregated (non-soluble) form of the mutant proteins. The protocol in the methods section does also not allow to judge whether they examine the insoluble or soluble fraction. The authors must clearly show that CCT2 has no effect on the WT protein versions of their mutated counterparts. Furthermore, if the authors cannot clearly show that aggregated species are analysed, then the phrasing solid aggregatephagy needs to be changed.

Many interaction experiments between CCT2 and the autophagy machinery proteins were done in the absence of solid aggregatephagy cargos or during starvation conditions. Can the authors exclude that CCT2 is a substrate of autophagy and how are these interactions then to be explained?

Quantification for all IP results are missing and if not done, repeats of these experiments need to be provided.

Minor points:

The dataset tables should be better explained.

In yeast, the authors show that Atg1 phosphorylation of Cct2 regulates the binding of Cct2 and Atg11 and Atg11 then binds Atg8. Can the authors speculate whether in mammalian cells such an adaptor protein that links CCT2 and LC3 does also exist (e.g., FIP200) or is the LC3-CCT2 interaction direct?

** As a service to authors, EMBO Press provides authors with the ability to transfer a manuscript that one journal cannot offer to publish to another journal, without the author having to upload the manuscript data again. To transfer your manuscript to another EMBO Press journal using this service, please click on Link Not Available

Dear Martina,

We are grateful for the opportunity to resubmit our manuscript and for the valuable suggestions provided by the reviewers. As suggested, we have performed extensive follow-up experiments to address their questions and comments. Please find below our point-by-point responses to each of the reviewer's points. Thank you once again for allowing us to address these issues.

Point-by-point responses to the reviewers:

Referee #1: The authors have previously reported that Cct2 acts as an autophagy receptor in degrading solid aggregates via autophagy and that it interacts directly with Atg8 both in yeast and mammals, and in vitro and in vivo (Ma et al., Cell 2022). In this new work, the authors show that Cct2 also interacts directly with Atg11 and that this interaction depends on D473, D487, S412 and S470 in vivo. Claimed is furthermore that this interaction requires phosphorylation of residues S412 and S470, that the responsible kinase is Atg1, and that this regulation is conserved in mammals. However, this claim on phosphoregulation lacks in vivo evidence. Also, the regulated assembly of the autophagy factors on aggregates is claimed but not shown. Convincingly shown is that Cct2 directly interacts with Atg11 in vitro, and this interaction is convincingly confirmed in vivo. Not sufficiently addressed but claimed are the phosphoregulation by Atg1 and the assembly of Cct2, Atg11, Atg8 and Atg1 on aggregates. Without showing these later two points convincingly, the novelty of the work is limited and the study is not suited for publication in EMBO Reports.

Specific points:

1. Figure 3 and 4 and supplement: The authors claim that Atg1 kinase

phosphorylates

Cct2 and that this phosphorylation is required for Cct2 binding to Atg8 and Atg11 in vivo. However, the experiments are only done in vitro, and this claim lacks in vivo evidence. Several kinases share a similar consensus to Atg1, and many kinases are in vitro not very specific, as long as the consensus is more or less there. To claim that Atg1 is the kinase acting on Cct2, it needs to be shown: - by mass spec that these sites are lost in an *atg1*-delta or *atg1*-kinase dead situation, and- that an *atg1*-delta/-kinase dead results in the loss of the Cct2-Atg8 and Cct2-Atg11 interactions.

Response: We fully agree with the reviewer’s insightful comments. As suggested, we purified 3×FLAG-Cct2 using anti-FLAG agarose beads from wild-type and *atg1Δ* yeast cells in the presence of Ape1-P22L under nutrient-rich conditions. Mass spectrometry (MS) results showed that 3×FLAG-Cct2 in wild-type yeast cells is phosphorylated at S412 and S470, whereas in *atg1Δ* cells, phosphorylation at S412 and S470 was absent (Dataset S1). Additionally, we investigated whether the deletion of *ATG1* affects the interaction of Cct2 with both Atg8 and Atg11. Co-immunoprecipitation (Co-IP) assays revealed a significant decrease in Cct2 binding to both Atg8 and Atg11 in *atg1Δ* cells compared to wild-type cells, in the presence of Ape1-P22L under nutrient-rich conditions (Figure R1, now Figure 5F and S5D). Taken together, these findings confirm that Atg1-mediated Cct2 phosphorylation is required for its association with Atg8 and Atg11.

Figure R1: Atg1 regulates the association of Cct2 with Atg8 and Atg11. (A) *ape1Δ atg8Δ* or *atg1Δ ape1Δ atg8Δ* cells co-expressing 3×HA-Cct2 with FLAG-Atg8 in the presence of Ape1 P22L were grown to the log phase under nutrient-rich conditions. Cell lysates were immunoprecipitated with anti-FLAG agarose beads and then

analyzed by western blot using anti-HA antibody. The data are representative of three independent experiments. **(B)** *ape1* Δ or *atg1* Δ *ape1* Δ cells co-expressing 3 \times HA-Cct2 with Atg11-3 \times FLAG in the presence of Ape1 P22L were grown to the log phase under nutrient-rich conditions. Cell lysates were immunoprecipitated with anti-FLAG agarose beads and then analyzed by western blot using anti-HA antibody. The data are representative of three independent experiments.

2. All in vivo verification of Cct2 interactions with Atg11, Atg8 and Atg1 is done in a setup lacking aggregates, therefore only represents a general ability of these proteins to interact in the absence of cargo, which questions the significance of the findings for the proposed model. The authors, however, claim from these results an association of all these factors with aggregates. Such an association has not been shown and needs to be clarified by microscopy. To make this claim they need to show:
- that Cct2, Atg11, Atg8 and Atg1 colocalize with aggregates (either Q47 or Ape1P22L), - that Cct2 colocalization with aggregates remains in *atg11*-delta cells, but Atg8 is lost, and- that the Cct2-D473A or I487E mutant results in loss of Atg11 colocalization to the aggregates.

Response: We thank the reviewer for the insightful comments. As suggested, we investigated the colocalization of Cct2, Atg11, Atg8, and Atg1 with Ape1 P22L. Image data and statistical analysis reveal that RFP-Ape1 P22L colocalizes well with Cct2-GFP, Atg11-GFP, GFP-Atg8, and Atg1-GFP under nutrient-rich conditions (Figure R2A and B, now Figure S4A and B).

Regarding whether the colocalization of Cct2 with Ape1 P22L is dependent on Atg11 or Atg8, fluorescence microscopy and statistical analysis showed that knocking out *ATG11* or *ATG8* did not affect the colocalization of Cct2 with Ape1 P22L (Figure R2C and D). This indicates that the recruitment of receptor Cct2 to the substrate Ape1 P22L occurs before the involvement of Atg8 and Atg11, which

aligns with our current understanding of the molecular regulatory mechanism of selective autophagy. Subsequently, we analyzed whether the binding of Cct2 to Atg11 is required for the colocalization of Atg11 with Ape1 P22L. We found that both the Cct2^{D473A} and Cct2^{I487E} mutants significantly decreased their colocalization under nutrient-rich conditions (Figure R2E and F, now Figure S4C and D), indicating that the binding of Cct2 to Atg11 is required for the colocalization of Atg11 with Ape1 P22L under these conditions.

Figure R2: Ape1 P22L co-localizes with autophagy-related proteins and Cct2, and Atg11-Cct2 binding is required for the co-localization of Ape1 P22L with Atg11.

(A) *ape1Δ* yeast cells co-expressing RFP-Ape1 P22L with Atg1-2×GFP, GFP-Atg8, Atg11-2×GFP, or Cct2-GFP were grown to the log phase under nutrient-rich conditions. Images of cells were obtained using an inverted fluorescence microscope. Scale bar, 2 μm. (B) Cells from (A) were quantified for the number of cells in which RFP-Ape1 P22L colocalized with the indicated GFP-fused proteins. n=300 cells were pooled from three independent experiments. Data are shown as mean ± SD. (C) Wild type, *atg1Δ*, or *atg8Δ* yeast cells co-expressing RFP-Ape1 P22L with Cct2-GFP were grown to the log phase under nutrient-rich conditions. Images of cells were obtained using an inverted fluorescence microscope. Scale bar, 2 μm. (D) Cells from (C) were

quantified for the number of cells in which RFP-Ape1 P22L colocalized with Cct2-GFP. n=300 cells were pooled from three independent experiments. Data are shown as mean \pm SD. NS, not significant; two-tailed Student's t tests were used. **(E)** *ape1* Δ yeast cells co-expressing RFP-Ape1 P22L and Atg11-2 \times GFP with HA-Cct2, HA-Cct2 D473A, or HA-Cct2 I487E were grown to log phase under nutrient-rich conditions. Images of cells were obtained using an inverted fluorescence microscope. Scale bar, 2 μ m. **(F)** Cells from (E) were quantified for the number of cells in which RFP-Ape1 P22L colocalized with Atg11-2 \times GFP. n=300 cells were pooled from three independent experiments. Data are shown as mean \pm SD. ***p < 0.001; two-tailed Student's t tests were used.

3. In figures 4 and S4 the authors show that the Cct2-2A mutant decreases Atg8 and Atg11 binding *in vivo*, however, when assessing this interaction *in vitro*, there is no defect (Figure 4F and S4D). If phosphorylation and therefore the negative charge on these serine residues was essential to promote these interactions, then an alanine mutant should abolish this interaction also *in vitro*. The authors' argument that this *in vitro* - *in vivo* difference shows that the Cct2-2A structure is not impaired, does not make sense at all! It rather indicates that the DIRECT interaction between Cct2 and Atg11/Atg8 is not affected by these two residues, which speaks against their phosphoregulation model.

Response: We thank the reviewer's comments. Regarding the issue of the Cct2 2A mutation impairing its binding with Atg8 and Atg11 *in vivo*, we were concerned that this result might be due to the mutation altering the structure of Cct2, thereby affecting its binding with Atg8 and Atg11 *in vivo*. To exclude this possibility, we performed *in vitro* pull-down assays and found that when the serine residue of Cct2 is not phosphorylated, its behavior is similar to that of the alanine mutation. This indicates that Cct2 2A mutation does not alter its structure; rather, the phosphorylation modification of Cct2 promotes the binding of Cct2 to Atg8 and Atg11.

To further confirm that Atg1-mediated Cct2 phosphorylation indeed promotes its binding with Atg8 and Atg11, we performed *in vitro* kinase reactions followed by pull-down assays. We found that the phosphorylation of Cct2 by Atg1 enhances its binding to both Atg8 and Atg11 *in vitro* (Figure R3A and B, now Figure 5H and S5F). Additionally, we conducted competitive binding experiments between Atg11 and Atg8 with Cct2. The results showed that although Atg1-mediated Cct2 phosphorylation increases its binding to both Atg8 and Atg11, the binding affinity of Cct2 to Atg11 CC4 is stronger than to Atg8 (Figure R3C, now Figure S5G). Taken together, these results suggest that Cct2 enhances its binding with Atg8 and Atg11, and subsequently initiate autophagy, when phosphorylated by Atg1.

Figure R3: Atg1-mediated Cct2 phosphorylation enhances its binding with Atg8 and Atg11. (A, B) *In vitro* kinase assays were performed using GST-Cct2 WT or 2A (S412A-S470A) purified from *E. coli* as substrates, with WT or kinase-dead (KD) Atg1-3×FLAG, purified from *ape1Δ* yeast cells expressing *Apel1* P22L under nutrient-rich conditions, served as the protein kinase. After that, *in vitro* GST pull-downs were performed using phosphorylated products enriched by GST beads *in vitro* with His-Atg8(A) or Atg11 CC4(B) protein purified from *E. coli*. Protein samples were separated by SDS-PAGE, and then detected using Coomassie blue staining. The data are representative of three independent experiments. (C) *In vitro* kinase assays were performed using GST-Cct2 WT or 2A (S412A-S470A)

purified from *E. coli* as substrates, with WT or kinase-dead (KD) Atg1-3×FLAG, purified from *ape1Δ* yeast cells expressing Ape1 P22L under nutrient-rich conditions, served as the protein kinase. After that, *in vitro* GST pulldowns were performed using phosphorylated products enriched by GST beads *in vitro* with His-Atg8 and Atg11 CC4 protein purified from *E. coli*. Protein samples were separated by SDS-PAGE, and then detected using Coomassie blue staining. The data are representative of three independent experiments.

4. The mammalian part shows similar problems in the authors' claims as for the yeast part, e.g. if ULK1 is required needs to be shown by addressing the CCT2-LC3C interaction in an ULK1 knockdown, and the presence of S458 phosphorylation needs to be verified by mass spec to be lost in the absence of ULK1.

Response: We fully agree with the reviewer's comments. To clarify, we used a ULK1 KO MEF cell line to examine whether ULK1 regulates the interaction between CCT2 and LC3C. Co-IP assays confirmed that ULK1 KO significantly decreases the binding of CCT2 to LC3C in the presence of SOD1 G93A-GFP under nutrient-rich conditions (Figure R4, now Figure 6J-L). Subsequently, we purified HA-CCT2 using anti-HA agarose beads in wild-type (WT) and ULK1 KO MEF cell lines in the presence of SOD1 G93A-GFP under nutrient-rich conditions. MS analysis revealed phosphorylation of the S458 site of CCT2 in wild type MEF cells, whereas S458 is not phosphorylated in ULK1 KO MEF cells (Dataset S2).

Figure R4: ULK1 regulates the binding of CCT2-LC3C in the presence of SOD1-

G93A-GFP under nutrient-rich conditions. (A) Western blot validation of ULK1 KO effect in MEF cell lines. The data are representative of two independent experiments. (B) WT or ULK1 KO MEF cell lines were co-transfected with FLAG-LC3C, HA-CCT2, and SOD1 G93A-GFP. Cell lysates were immunoprecipitated with anti-FLAG agarose beads and then analyzed by western blot using an anti-HA antibody. (C) Quantification of IP/Lysates HA-CCT2 from (B). Data are presented as means \pm SD (n=3). ***p < 0.001; two-tailed Student's t tests were used.

5. As solid aggregates the authors use Ape1-P22L and Q47. From their results, they claim that the turnover of these depends on Atg11 binding to Cct2. However, for both aggregates, they don't show that these aggregates fail to be turned over in an atg11-delta situation. They only did the experiments in a Cct2 point mutant, which might also lack binding to factors other than Atg11. Therefore, the aggregate degradation experiments need to be repeated in atg11-delta cells.

Response: We thank the reviewer for the insightful comments. As suggested, we performed assays on the cleavage of PrApe1 P22L and GFP-47Q, which revealed that deletion of *ATG11* completely blocked the cleavage of PrApe1 P22L and GFP-47Q under nutrient-rich conditions (Figure R5, now Figure S3I-L). Collectively, these findings indicate that Atg11 plays a crucial role in the degradation of aggregates.

Figure R5: Atg11 plays a crucial role in the degradation of aggregates. (A) The indicated yeast cells were grown to the log phase under nutrient-rich conditions. Samples were analyzed by immunoblot for detecting the maturation of PrApe1 into mApe1. Pgk1 served as a loading control. **(B)** Quantification of mApe1 to PrApe1+mApe1 ratio from (A). Data are presented as means \pm SD (n=3). ***p < 0.001; NS, not significant; two-tailed Student's t tests were used. **(C)** The indicated yeast cells were grown to an OD₆₀₀= 0.6 under nutrient-rich conditions. Subsequently, 0.1 mM CuSO₄ was added to the cells to induce GFP-47Q to form solid aggregates for 12 hours. Samples were analyzed by immunoblot for the cleavage of GFP-47Q. Pgk1 served as a loading control. **(D)** Quantification of GFP to GFP-47Q+GFP ratio from (C). Data are presented as means \pm SD (n=3). ***p < 0.001; two-tailed Student's t tests were used.

6. Fig S2A and S4A: How was vacuolar localization judged without a co-stain? The pictures shown don't allow a localization judgment of Ape1-P22L. As Ape1-P22L can get extremely big due to the strongly decreased Atg19 binding, which leads to higher oligomerization of Ape1 (Shintani et al. Dev Cell 2002, Bertipaglia et al. EMBO Rep 2016, Yamasaki et al. Mol Cell 2020), it could be misleading and a vacuolar co-stain should be used. As the effects observed by microscopy seem much more severe than

the very mild effect found by western blots, this is essential to clarify.

Response: We thank the reviewer for their comments. As suggested, we tagged the C-terminus of Vph1 with GFP to label the yeast vacuolar membrane. Using this method, we repeated the experiments from the original Figures S2A and S4A. Image data and statistical analysis indicate that the binding of Cct2 with Atg11 and Atg1-mediated Cct2 phosphorylation are required to promote the vacuolar localization of Ape1 P22L (Figure R6, now Figure 3A, 3B, 5A, and 5B).

Figure R6: Both the binding of Cct2-Atg11 and Atg1-mediated Cct2 phosphorylation are required for the maturation of PrApe1 into mApe1. (A, C) *ape1Δ* or *ape1Δ atg5Δ* yeast cells co-expressing Vph1-GFP and RFP-Ape1 P22L, along with empty vector (EV), WT 3×HA-Cct2, or 3×HA-Cct2 variants, cultured to the log phase under nutrient-rich conditions. Images of cells were obtained using an inverted fluorescence microscope. Scale bar, 2 μm. **(B, D)** Cells from (A, C) were quantified for the vacuolar localization of RFP-Ape1 P22L. n=300 cells were pooled from three independent experiments. Data are presented as means ± SD. ***p <

0.001; two-tailed Student's t tests were used.

7. Another general issue is that Cct2 is overexpressed in all experiments, and this is done on top of the endogenous Cct2, both for WT and mutants. Whether endogenous Cct2 is acting in aggregate turnover has neither been addressed in the previous work nor here. As Cct2 is essential, a temperature-sensitive mutant or degron system could be used to clarify this point. Otherwise, the authors cannot claim that Cct2 acts as a bona fide autophagy receptor, and need to state properly throughout the text that their observation "is enhanced aggregate turnover upon Cct2 overexpression."

Response : We thank the reviewer for the insightful comments. As suggested, we employed a degron system to induce the degradation of endogenous Cct2. The results reveal that degradation of Cct2 by IAA treatment significantly reduces the cleavage of PrApe1-P22L under nutrient-rich conditions (Figure R7, now Figure 1H and I), indicating that endogenous Cct2 is crucial for aggregate turnover.

Figure R7: Endogenous Cct2 plays an important role in aggregate turnover. (A) *ape1Δ atg5Δ*, *ape1Δ*, or *ape1Δ 3×HA-AID-Cct2* yeast cells were treated with 0.5 mM IAA for 0h or 2h under nutrient-rich conditions. Samples were analyzed by immunoblotting to detect the maturation of PrApe1 into mApe1 and the expression levels of 3×HA-AID-Cct2 protein. Pgk1 served as a loading control. **(B)** Quantification of the mApe1 to PrApe1+mApe1 ratio from (A). Data are presented as means ± SD(n=3). ***p < 0.001; NS, not significant; two-tailed Student's t tests were

used.

8. Most western blots are not quantified, and only shown n=1. It needs to be stated if this has been repeated and if the same result has been observed (minimum n=2). Ideally, the repetitions could be shown in the supplement, which would make the rather mild effects more convincing.

Response: We appreciate the reviewer for raising this point. As suggested, we have quantified the important IP results and provided the number of repetitions for IP experiments in the revised manuscript.

Minor points:

9. The authors use inconsistent naming of CCT2, sometimes capital CCT2 (e.g. Fig 3G), sometimes Cct2 (e.g. Fig 3F). The latter is the yeast nomenclature, the previous is the mammalian. They should do this correctly throughout the manuscript, in the figures, the text and the legends.

Response : We have corrected it in the revised manuscript, figures, and legends.

10. Unclear is why the authors do some experiments under starvation or torin treatment, although they look at selective autophagy and most experiments seem to be done under nutrient-rich conditions. This should be explained. If effects are only observed under starvation and not under rich conditions, they need to rule out bulk autophagy in these experiments.

Response: We thank the reviewer for these insightful suggestions. The main reason some experiments were performed under starvation or torin treatment conditions is that aggrephagy occurs more prominently under these conditions. For example, the regulatory mechanisms of selective autophagy, such as ribophagy and ER-phagy, are typically studied under starvation conditions. Furthermore, we fully agree with the reviewer's comments. Therefore, we have repeated these experiments under nutrient-rich conditions in the presence of

aggregated cargo. Through these experiments, we found that the results remained consistent with those obtained under starvation or torin treatment conditions, as shown in the revised manuscript and Figure.

11. The authors state: "...an increase in the binding of CCT2 with Atg1 under nitrogen starvation conditions (Fig. 3A), suggesting the potential phosphorylation of CCT2 by Atg1 in response to stress." This argument is strange as kinases usually don't stably interact with their substrate, and should at least not be used to support the claim that Atg1 phosphorylates Cct2.

Response: We appreciate the reviewer's comment. As mentioned, protein kinases sometimes do not exhibit stable binding with their substrates, but in other cases, they can indeed interact stably with their substrates. For instance, in a study titled "AMPK and mTOR regulate autophagy through direct phosphorylation of ULK1" published by Kun-Liang Guan's group in Nature Cell Biology (PMID: 21258367), they demonstrated that ULK1 is a substrate of AMPK and can stably bind to AMPK.

Furthermore, as suggested, we have revised original Figure 3A to depict the experiment detecting the interaction between ULK1 and CCT2. Co-IP assays confirm this interaction between ULK1 and CCT2 in the presence of SOD1-G93A-GFP under nutrient-rich conditions. Finally, we have adjusted the statement from "suggesting the potential phosphorylation of CCT2 by Atg1 in response to stress" to "suggesting that CCT2 may be a substrate of ULK1."

12. The authors use GFP-Q47 as a solid aggregate. They cite their previous paper for this, however, in their previous work, it has not been shown that GFP-Q47 is indeed solid. They need to cite the original literature for this if existent, or otherwise show it experimentally.

Response: We appreciate the reviewer's comment. To verify this, we treated the

cells with Cu^{2+} for 12 hours to induce GFP-47Q overexpression and aggregate formation. Subsequently, we performed a fluorescence recovery after photobleaching (FRAP) assay and observed minimal fluorescence recovery for GFP-47Q after bleaching, indicating the formation of solid aggregates.

Figure R8: GFP-47Q forms a solid aggregate. (A) BY4741 yeast cells expressing Cu^{2+} -inducible GFP-47Q plasmids were cultured for 12 h under nutrient-rich conditions. FRAP analysis was performed at the indicated time points. Scale bar, 2 μm . (B) Quantification of the normalized GFP-47Q fluorescence signal (mean \pm SD) in (A) (>30 cells from three independent experiments).

Referee #2:

The authors' groups previously discovered autophagic degradation of solid protein aggregates, which involves the chaperonin subunit CCT2 as a receptor in both yeast and mammals; CCT2 binds to both the aggregates and Atg8-family proteins anchored to the autophagosomal membrane and thereby mediates their sequestration into the autophagosome. In this study, the authors further addressed the regulation of this solid aggregate pathway. They first showed in yeast that similar to most of the yeast autophagy receptors, CCT2 interacts with Atg11 and that this interaction facilitates the CCT2-Atg8 interaction. In addition, they found that CCT2 is phosphorylated by the autophagy initiation kinase Atg1 and that this phosphorylation enhances the CCT2 interactions with Atg11 and Atg8. These interactions and phosphorylation of CCT2 were shown to be important for solid aggregate degradation. Finally, the authors showed that mammalian CCT2 is also phosphorylated by the Atg1 homolog ULK1, and this

phosphorylation is important for the CCT2 interaction with the Atg8 homolog LC3C and thus for solid aggrephagy. Thus this study reveals how solid aggrephagy is regulated but the following issues should be addressed to strengthen the authors' conclusions.

Major comments:

1. The authors showed that the CCT2-Atg11 interaction facilitates the CCT2-Atg8 interaction, but the underpinning mechanism remains unknown. To assess whether the former interaction directly increases the latter, the author should perform *in vitro* pull-down assay for the CCT2-WT/CCT2-D473A/I487E-Atg8 interaction with or without Atg11. On the other hand, another possibility would be that autophagosome formation on solid aggregates, which requires the former interaction, promotes the latter via the production of Atg8-PE in the expanding phagophore. If so, deletion of core ATG genes such as ATG2 would also decrease the CCT2-Atg8 interaction *in vivo* even in the presence of Atg11.

Response: We appreciate the reviewer for the insightful comments. As suggested, we performed *in vitro* pull-down assays and found that Cct2 can simultaneously bind to both Atg11 and Atg8. Notably, when Atg11 CC4 domain cannot bind to Cct2, the interaction between Atg8 and Cct2 increases (Figure R9A and B). This contrasts with the inhibition of Atg8 binding to Cct2 when Cct2 cannot bind to Atg11 *in vivo*. We speculate that this opposite result between *in vitro* and *in vivo* may be due to temporal and spatial regulation of selective autophagy *in vivo*.

Additionally, as suggested, we explored whether autophagic core genes *ATG2* or *ATG14* are involved in regulating the binding of Cct2 to Atg8 in the presence of Ape1 P22L under nutrient-rich conditions. Co-IP assays showed that the deletion of *ATG2* or *ATG14* did not affect the interaction between Cct2 and Atg8 (Figure R9C and D, now Figure S3I and J). Taken together, these data suggest that Atg11 is specifically required for the binding of Cct2 to Atg8.

Figure R9: The deletion of *ATG2* or *ATG14* did not affect the interaction between Cct2 and Atg8. (A) *In vitro* GST pull-down assays were performed using GST-Cct2, GST-Cct2^{D473A}, or GST-Cct2^{I487E} with His-Atg8 purified from *E. coli*, in the presence or absence of Atg11 CC4 protein. Protein samples were separated by SDS-PAGE and detected using Coomassie blue staining. The data are representative of three independent experiments. (B) Quantification of Pulldowns/lysates His-Atg8 from (A). Data are presented as means \pm SD (n=3). ***p < 0.001; NS, not significant; two-tailed Student's t tests were used. (C, D) *ape1* Δ , *atg2* Δ *ape1* Δ (C), or *atg14* Δ *ape1* Δ (D) cells co-expressing 3×HA-Cct2 with FLAG-Atg8 in the presence of Ape1 P22L were grown to the log phase under nutrient-rich conditions. Cell lysates were immunoprecipitated with anti-FLAG agarose beads and then analyzed by western blot using anti-HA antibody. The data are representative of three independent experiments.

2. The authors showed that Atg1-mediated phosphorylation of CCT2 promotes the CCT2 interactions with both Atg11 and Atg8. However, they insist that CCT2 phosphorylation increases its interaction with Atg8 Although without further dissecting the effect of CCT2 phosphorylation on these interactions; there remains the

possibility that CCT2 phosphorylation by Atg1 enhances the CCT2-Atg11 interaction, which promotes the CCT2-Atg8 interaction. The authors should clarify this point. Does abolishing CCT2 phosphorylation decrease the CCT2-Atg8 interaction in the absence of Atg11 *in vivo*? How CCT2 phosphorylation by Atg1 affect the CCT2-Atg8 and CCT2-Atg11 interactions in *in vitro* pull-down assay?

Response: We thank the reviewer's insightful comments. As suggested, we performed the relevant co-IP experiments and observed that deletion of *ATG11* did not further reduce the decrease in the Cct2-Atg8 interaction caused by abolishing Cct2 phosphorylation (Figure R10A and B). Subsequently, through *in vitro* kinase reactions followed by pull-down assays, we found that Cct2 phosphorylation by Atg1 enhances its binding to both Atg8 and Atg11, respectively (Figure R10C and D, now Figure 5H and S5F). Furthermore, we performed competitive binding assays between Atg11 and Atg8 with Cct2 under such conditions. We found that while Atg1-mediated Cct2 phosphorylation enhances its binding to both Atg8 and Atg11, the affinity of Cct2 for Atg11 CC4 is stronger than for Atg8 (Figure R10E, now Figure S5G). Collectively, these findings indicate that Atg1-mediated Cct2 phosphorylation independently promotes interactions with both Atg11 and Atg8.

Figure R10: Atg1-mediated Cct2 phosphorylation independently promotes interactions with both Atg11 and Atg8. (A) *ape1* Δ or *ape1* Δ *atg11* Δ yeast cells co-expressing HA-Cct2 or HA-Cct2 S412A-S470A(2A) with FLAG-Atg8 were grown to log phase in the presence of Ape1 P22L under nutrient-rich conditions. Cell lysates were immunoprecipitated with anti-FLAG agarose beads and then analyzed by western blot using anti-HA antibody. (B) Quantification of IP/lysates HA-CCT2 from (A). Data are presented as means \pm SD (n=3). ***p < 0.001; two-tailed Student's t tests were used. (C, D) *In vitro* kinase assays were performed using GST-Cct2 WT or 2A (S412A-S470A) purified from *E. coli* as substrates, with WT or KD Atg1-3 \times FLAG purified from yeast cells as a protein kinase. After that, *in vitro* GST pulldowns were performed using phosphorylated products enriched by GST beads *in vitro* with His-Atg8(C) or Atg11 CC4(D) protein purified from *E. coli*. Protein samples were separated by SDS-PAGE, and then detected using Coomassie blue staining. The data are representative of three independent experiments. (E) *In vitro* kinase assays were performed using GST-Cct2 WT or 2A (S412A-S470A) purified from *E. coli* as substrates, with WT or KD Atg1-3 \times FLAG purified from yeast cells as a protein kinase. After that, *in vitro* GST pulldowns were performed using phosphorylated products enriched by GST beads *in vitro* with His-Atg8 and Atg11 CC4 protein purified from *E. coli*. Protein samples were separated by SDS-PAGE, and then detected using Coomassie blue staining. The data are representative of three independent experiments.

3. In the authors' model, free CCT2 (not assembled into the chaperonine complex) acts as a solid aggrephagy receptor. Is free CCT2 more efficiently phosphorylated by Atg1 compared with that in the complex?

Response: That's a good question. To address this issue, we investigated the interaction of Atg8 with Cct2 and other subunits of the TRiC complex, such as Cct1, Cct4, Cct5, and Cct8. Consistent with results in mammalian cells (PMID: 35366418), Atg8 interacts exclusively with Cct2 and not with other TRiC complex subunits like Cct1, Cct4, Cct5, or Cct8 (Figure R11A, now Figure S1C). Additionally, we also examined the interaction of Atg11 with Cct2 and other TRiC complex subunits. Co-IP assays revealed that Atg11 interacts specifically with Cct2 and not with other TRiC complex subunits (Figure R11B). These findings indicate that autophagy-related proteins interact selectively with free Cct2 protein rather than with the entire TRiC complex. Given that Atg1 and Cct2 directly bind to both Atg8 and Atg11, it suggests that free Cct2, through its interaction with Atg8 and Atg11, more efficiently phosphorylated by Atg1 compared to when it is part of the complex.

Figure R11: Atg8 and Atg11 specifically interact with free Cct2 protein. (A) *ape1* Δ cells co-expressing HA-Cct1, HA-Cct2, HA-Cct4, HA-Cct5, or HA-Cct8 with FLAG-Atg8 and Ape1 P22L were grown to the log phase under nutrient-rich conditions. Cell lysates were immunoprecipitated with anti-FLAG agarose beads and then analyzed by western blot using anti-HA antibody. The data are representative of

three independent experiments. **(B)** *ape1*Δ cells co-expressing HA-Cct1, HA-Cct2, HA-Cct4, HA-Cct5, or HA-Cct8 with Atg11-3×FLAG and Ape1 P22L were grown to the log phase under nutrient-rich conditions. Cell lysates were immunoprecipitated with anti-FLAG agarose beads and then analyzed by western blot using anti-HA antibody. The data are representative of three independent experiments.

4. The authors should examine whether CCT2 interacts with FIP200 in mammalian cells and if this is the case how the interaction affects the CCT2-LC3C interaction.

Response: As suggested, we explored whether the mammalian homolog of Atg11, FIP200, can bind to CCT2 and regulate its association with LC3C. Co-IP assays revealed that FIP200 can indeed bind with CCT2 in the presence of SOD1-G93A-GFP under nutrient-rich conditions (Figure R12A, now Figure 6M), and FIP200 KO significantly reduces the interaction between CCT2 and LC3C (Figure R12B-D, now Figure 6N-P). Additionally, we found that WIPI2 KO does not affect the binding of LC3C to CCT2 (Figure R12E-G, now Figure S6I-K). Taken together, these findings suggest that the regulatory role of Atg11 in the binding of CCT2 to LC3C is conserved in mammals.

Figure R12: FIP200 regulates the binding of CCT2-LC3C. (A) HEK293T cell lines were co-transfected with FLAG-CCT2, HA-FIP200, and SOD1 G93A-GFP. Cell lysates were immunoprecipitated with anti-FLAG agarose beads and then analyzed by western blot using an anti-HA antibody. The data are representative of three independent experiments. (B) Western blot validation of FIP200 KO effect in MEF cell lines. The data are representative of two independent experiments. (C) WT or FIP200 KO MEF cell lines were co-transfected with FLAG-LC3C, HA-CCT2, and SOD1 G93A-GFP. Cell lysates were immunoprecipitated with anti-FLAG agarose beads and then analyzed by western blot using an anti-HA antibody. (D) Quantification of IP/Lysates HA-CCT2 from (C). Data are presented as means \pm SD (n=3). ***p < 0.001; two-tailed Student's t tests were used. (E) Western blot validation of WIPI2 KO effect in MEF cell lines. The data are representative of two independent experiments. (F) WT or WIPI2 KO MEF cell lines were co-transfected with FLAG-LC3C, HA-CCT2, and SOD1 G93A-GFP. Cell lysates were immunoprecipitated with anti-FLAG agarose beads and then analyzed by western blot using an anti-HA antibody. (G) Quantification of IP/lysates HA-CCT2 from (F). Data are presented as means \pm SD (n=3). NS, not significant; two-tailed Student's t tests

were used.

Minor comments:

1. Page 6, lines 26-27, "We proceeded to... its interaction with Atg11": This sentence should be rewritten.

Response : We have corrected this sentence in the revised manuscript.

2. Page 8, lines 7-12, "Considering...with Atg8 (Fig. 2E).": This sentence should be rewritten.

Response : We have already rewritten this sentence in the revised manuscript.

3. Figure 2B: The graph labels should be corrected.

Response : We corrected the graph labels in original Figure 2B.

4. The authors should describe why they chose LC3C among several Atg8-family proteins in mammalian cells.

Response : We thank the reviewer's comment. As suggested, we describe the reason why chose LC3C among Atg8-family proteins in mammalian cells in the revised introduction section.

Referee #3:

CCT2 was recently discovered to be a solid aggrephagy receptor that functions independently of the chaperonin TCP-1 ring complex (TRiC). Selective autophagy receptors (SARs) are known to bind to members of the ATG machinery and thereby autophagosome formation is initiated around the cargo that is bound by the SARs. In this manuscript, the authors mostly investigated the role of Cct2 in a yeast model. They show that Cct2 is phosphorylated by Atg1 on 2 sites (Ser 412 and 470) and interacts with Atg11. The phosphorylation and subsequent Atg11 binding regulate the binding to Atg8. Furthermore, the authors showed that a similar mechanism is conserved in mammalian cells. The manuscript advances the knowledge of CCT2 as a selective autophagy receptor.

Nevertheless, there are three major concerns about the interpretation of the data (that will be further detailed below): First, the authors did not establish that CCT2 works independently from the TriC complex in yeast. Second, the authors claim to investigate solid aggrephagy, although the substrates they test are mostly soluble mutant proteins. Third, most of the interaction experiments were performed in the absence of a solid aggrephagy cargo and often even under starvation conditions. Therefore, the provided data does not allow to call the selective form of autophagy, observed here, solid aggrephagy or that the modulations of CCT2 happen during solid aggrephagy.

Major points:

Figure 1: Other subunits of the TriC (e.g., CCT1, 3 or 6) needs to be tested to show that CCT2 works independently (also in yeast).

Can the authors exclude that Atg11 is a substrate of the TriC?

Response: We thank the reviewer for the insightful comments. To address this issue, we first investigated whether the overexpression of other subunits of the TRiC complex, such as Cct1 and Cct4, similar to Cct2 overexpression, could promote the maturation of Ape1 P22L. Western blot results showed that while overexpression of Cct2 promoted Ape1 P22L maturation, overexpression of Cct1 and Cct4 did not have the same effect (Figure R13A and B, now Figure 1A and B). This suggests that specifically, Cct2 overexpression among the TRiC complex subunits promotes the maturation of Ape1 P22L.

To confirm that Cct2 promotes Ape1 P22L maturation independently of the function of TRiC complex in yeast, we introduced an overexpressed Cct2 plasmid into temperature-sensitive Cct1 and Cct4 yeast strains. The cct1-ts and Cct4-ts yeast strains exhibit growth patterns similar to the wild-type yeast strain BY4741 at 30°C but are lethal at 37°C (Figure R13C, now Figure 1C). Subsequent

Western blot assays demonstrated that overexpressed Cct2 still promotes the maturation of PrApe1 P22L when Cct1 and Cct4 are inactive (Figure R13D and E, now Figure 1D and E). Furthermore, imaging data and statistical analysis indicate that overexpressed Cct2 can still facilitate the vacuolar localization of Ape1 P22L in the absence of active Cct1 and Cct4 (Figure R13F and G, now Figure 1F and G). *In vitro* pull-down assays additionally revealed that Atg8 binds exclusively to Cct2 and not to Cct1 or Cct4 (Figure R13H, now Figure S1B). Consistently, co-immunoprecipitation (co-IP) assays indicated that Atg8 binds only to free Cct2 and does not interact with other subunits of the TRiC complex (Figure R13I, now Figure S1C). These results strongly suggest that the regulation of solid aggrephagy by Cct2 in yeast is highly conserved with that in mammalian cells, and Cct2's involvement in solid aggrephagy does not depend on the TRiC complex's function.

Moreover, we investigated whether Atg11 is a substrate of the TRiC complex. Previous studies have shown that α -tubulin is a substrate of the TRiC complex. Using yeast temperature-sensitive strains for Cct1 and Cct4, we observed that when Cct1 and Cct4 were inactive, the protein level of α -tubulin significantly decreased, whereas the protein level of Atg11 did not change significantly (Figure R13J, now Figure S1A). This indicates that Atg11 is not a substrate of the TRiC complex.

Figure R13: Cct2 regulates solid aggregophagy independently of TRiC complex function. (A) *ape1Δ atg5Δ* or *ape1Δ* yeast cells co-expressing Ape1 WT or Ape1 P22L with empty vector (EV), 3×HA-Cct1, 3×HA-Cct2, or 3×HA-Cct4 were grown to log phase under nutrient-rich medium. Samples were analyzed by immunoblot for detecting the maturation of PrApe1 into mApe1. Pgk1 served as a loading control. (B) Quantification of mApe1 to PrApe1+mApe1 ratio from (A). Data are presented as means ± SD(n=3). ***p < 0.001; NS, not significant; two-tailed Student's t tests were used. (C) The indicated yeast strains were plated in 3-fold serial dilution onto YPD at 30°C or 37°C for 2 d. (D) *ape1Δ atg5Δ*, *ape1Δ*, *ape1Δ cct1-ts*, or *ape1Δ cct4-ts* yeast cells co-expressing Ape1 P22L with empty vector (EV) or 3×HA-Cct2 were grown to

log phase under nutrient-rich conditions. Samples were analyzed by immunoblot for detecting the maturation of PrApe1 into mApe1. Pgk1 served as a loading control. **(E)** Quantification of mApe1 to PrApe1+mApe1 ratio from (D). Data are presented as means \pm SD (n=3). ***p < 0.001; NS, not significant; two-tailed Student's t tests were used. **(F)** *ape1* Δ *atg5* Δ or *ape1* Δ yeast cells co-expressing RFP-Ape1 P22L and Vph1-GFP with either an empty vector (EV) or 3 \times HA-Cct2 were grown to log phase at 30 $^{\circ}$ C. These yeast strains were then cultured at 37 $^{\circ}$ C for 2 h under full medium conditions. Images of cells were obtained using an inverted fluorescence microscope. Scale bar, 2 μ m. **(G)** Cells from (F) were quantified for the vacuolar localization of RFP-Ape1 P22L. n=300 cells were pooled from three independent experiments. Data are presented as means \pm SD. ***p < 0.001; NS, not significant; two-tailed Student's t tests were used. **(H)** GST pulldowns were performed by using purified GST, GST-Cct1, GST-Cct2, or GST-Cct4 with His-Atg8 protein from *E. Coli*. The data are representative of three independent experiments. **(I)** *ape1* Δ cells co-expressing HA-Cct1, HA-Cct2, HA-Cct4, HA-Cct5, or HA-Cct8 with FLAG-Atg8 and Ape1 P22L were grown to the log phase under nutrient-rich medium. Cell lysates were immunoprecipitated with anti-FLAG agarose beads and then analyzed by western blot using anti-HA antibody. The data are representative of three independent experiments. **(J)** BY4741, *cct1-ts*, or *cct4-ts* yeast cells expressing Atg11-3 \times FLAG were grown to early log phase at 30 $^{\circ}$ C and then subjected to 37 $^{\circ}$ C for 2 hours. Samples were analyzed by immunoblot for detecting the expression of Atg11-3 \times FLAG and α -Tubulin. Pgk1 served as a loading control. The data are representative of three independent experiments.

Figure S2, S4: how did the authors determine the localization of the vacuole without staining? It needs to be indicated how the quantification was done (automatically or manual)

Response: We appreciate the reviewer's comments. Reviewer 1 also raised the

same issue. To clarify this issue, we utilized GFP tagging at the C-terminus of Vph1 to label the yeast vacuolar membrane. Using this approach, we replicated the experiments originally presented in Figures S2 and S4. The image data and statistical analyses confirm that both the binding of Cct2 with Atg11 and Atg1-mediated phosphorylation of Cct2 are crucial for the vacuolar localization of Ape1-P22L under nutrient-rich conditions (Figure R6, now Figures 3A, 3B, 5A, and 5B).

Figure R6: Both the binding of Cct2-Atg11 and Atg1-mediated Cct2 phosphorylation are required for the maturation of PrApe1 into mApe1. (A, C) *ape1Δ* or *ape1Δ atg5Δ* yeast cells co-expressing Vph1-GFP and RFP-Ape1 P22L, along with empty vector (EV), WT 3×HA-Cct2, or 3×HA-Cct2 variants, and cultured to the logarithmic phase under nutrient-rich medium. Images of cells were obtained using an inverted fluorescence microscope. Scale bar, 2 μm. **(B, D)** Cells from (A, C) were quantified for the vacuolar localization of RFP-Ape1 P22L. n=300 cells were pooled from three independent experiments. Data are presented as means ± SD. ***p < 0.001; two-tailed Student's t tests were used.

Figure 2A, B: the effect of the WT-CCT2 vs the mutants (or KO) seems rather marginal, especially given that for the WT-CCT2, the expression levels are higher compared to the mutants. Do the authors still detect this defect when the results are normalized against the CCT2 expression levels? To show that CCT2 is a solid aggrephagy receptor in yeast (independently of TriC), the authors need to show that CCT2 is delivered to the vacuole and not the Atg11 binding mutants, and that deletions of other TriC subunits do not impair the delivery of ApeP22L.

Response: We appreciate the reviewer's insightful comments. As suggested, we repeated the experiments originally presented in Figures 2A and 2B. By normalizing against the expression levels of Cct2 wild type and mutants, we confirmed that the interaction between Cct2 and Atg11 is crucial for promoting the maturation of PrApe1 P22L (Figure R14A and B, now Figure 3C and D).

Additionally, we investigated whether Cct2 enters the yeast vacuole in the presence of Ape1 P22L under nutrient-rich conditions. Imaging data indicate that Cct2 can enter the vacuole under these conditions. However, the number of cells showing vacuolar localization is significantly reduced for Cct2 mutants unable to bind Atg11 (Figure R14C and D, now Figure 3E and F). This suggests that the binding of Cct2 to Atg11 is required for Ape1 P22L to enter the vacuole. Furthermore, imaging data and statistical analysis demonstrate that overexpressed Cct2 can still facilitate the vacuolar localization of Ape1 P22L in the absence of active Cct1 and Cct4 (Figure R14E-G, now Figure 1C, 1F and G). These findings collectively indicate that Cct2 serves as a solid aggrephagy receptor in yeast, independently of its function in the TriC complex.

Figure R14: CCT2 is a solid aggregophagy receptor in yeast independently of TriC complex function. (A) *ape1Δ atg5Δ* or *ape1Δ* yeast cells co-expressing Ape1 WT or Ape1 P22L with empty vector (EV), 3×HA-Cct1, 3×HA-Cct2, or 3×HA-Cct4 were grown to log phase under nutrient-rich conditions. Samples were analyzed by immunoblot for detecting the maturation of PrApe1 into mApe1. Pgk1 served as a loading control. (B) Quantification of mApe1 to PrApe1+mApe1 ratio from (A). Data are presented as means ± SD (n=3). ***p < 0.001; NS, not significant; two-tailed Student's t tests were used. (C) *ape1Δ* Ape1 P22L yeast cells co-expressing Vph1-mCherry with Cct2-GFP, Cct2^{D473A}-GFP, or Cct2^{I487E}-GFP were grown to OD₆₀₀=0.8 and then further cultured in nutrient-rich medium for 6 h. Images of cells were obtained using an inverted fluorescence microscope. Scale bar, 2 μm. (D) Cells from (C) were quantified for the vacuolar localization of Cct2-GFP. n=300 cells were pooled from three independent experiments. Data are presented as means ± SD. ***p < 0.001; two-tailed Student's t tests were used. (E) The indicated yeast strains were plated in 3-fold serial dilution onto YPD at 30°C or 37°C for 2 d. (F) *ape1Δ atg5Δ* or *ape1Δ* yeast cells co-expressing RFP-Ape1 P22L and Vph1-GFP with either an empty vector (EV) or 3×HA-Cct2 were grown to log phase at 30°C. These yeast strains were

then cultured at 37°C for 2 h under full medium conditions. Images of cells were obtained using an inverted fluorescence microscope. Scale bar, 2 μ m. (G) Cells from (F) were quantified for the vacuolar localization of RFP-Ape1 P22L. n=300 cells were pooled from three independent experiments. Data are presented as means \pm SD. ***p < 0.001; NS, not significant; two-tailed Student's t tests were used.

Figure 2C, D: similar controls as suggested for figure 2A, B should also tested here. The authors analyzing here yeast cells that have been starved for 1-2 days, thereby massively inducing bulk autophagy. How can the authors conclude that the effect is specific for selective autophagy? These experiments need to be repeated under non-starving conditions.

Response: We appreciate the reviewer's comments. As suggested, we performed the experiment in the presence of GFP-47Q under nutrient-rich conditions. Western blot results demonstrated that both the Cct2^{D473A} and Cct2^{I487E} mutants significantly decreased the generation of free GFP (Figure R15, now Figure S3E and F), indicating that the binding of Cct2 with Atg11 is required for the degradation of GFP-poly 47Q under nutrient-rich conditions.

Figure R15: The binding of Cct2 with Atg11 is required for the cleavage of GFP-47Q under nutrient-rich conditions. (A) Yeast cells co-expressing Cu²⁺-inducible GFP-47Q plasmids with an empty vector (EV), 3 \times HA-Cct2, or 3 \times HA-Cct2 variants were grown to an OD₆₀₀ = 0.6. Subsequently, 0.1 mM CuSO₄ was added to the cells to induce GFP-47Q to form solid aggregates for 12 h. Samples were analyzed by immunoblot for the cleavage of GFP-47Q. Pgk1 served as a loading control. **(B)** Quantification of GFP to GFP-47Q+GFP ratio from (A). Data are presented as means

± SD(n=3). ***p < 0.001; two-tailed Student's t tests were used.

Figure 3: The authors claim that have identified the two phosphorylation sites in CCT2 by Atg1 during solid aggrephagy. This interpretation of these data bears multiple problems. First, the authors have not tested all the phosphorylation site. They focused on the D3 region and did not consider the other 2 regions, where there was phosphorylation detected (albeit less strong). Second, in the dataset table S1 there are many more phosphorylation site detected and the description why exactly the mentioned 2 phospho site are important is missing. Third, there is no evidence that Atg1 is phosphorylating CCT2 in vivo during solid aggrephagy. On the contrary, the authors show that CCT2 is phosphorylated during starvation induced autophagy in vivo, not solid aggrephagy. To prove that Atg1 is the responsible kinase in vivo, the CCT2 phosphorylation needs to be tested in the absence of Atg1. And to be able to claim that this phosphorylation happens during solid aggrephagy specifically, the experiment must be done under non-starved conditions with an appropriated aggregated cargo.

Response: We appreciate the reviewer's valuable comments. Regarding the first issue, the lack of a negative control (Atg1 kinase-dead) might have contributed to this problem. Additionally, it is possible that the *in vitro* purified protein was not entirely pure or that other factors in the reaction system contributed to this issue. We subsequently optimized the protein purification and *in vitro* kinase reaction system by adding two negative controls: one without ATP and one with Atg1 kinase-dead (Atg1 KD). Meanwhile, we also repurified these proteins. Western blot results indicated that the D3 domain of Cct2 is indeed phosphorylated by Atg1, while the D1 and D2 domains of Cct2 are not phosphorylated by Atg1(Figure R16A, now Figure 4D).

Regarding the reviewer's second and third points, we fully agree with the comments. These suggestions greatly enhance the rigor of our conclusions.

Indeed, it is necessary to identify the phosphorylation sites of Cct2 in wild-type and *ATG1* knockout yeast cells in the presence of aggregated cargo under nutrient-rich conditions. To this end, we purified the Cct2-3×FLAG protein using anti-FLAG agarose beads from both wild-type and *atg1Δ* yeast cells in the presence of Ape1 P22L under nutrient-rich conditions. Through MS analysis, we found that the S412 and S470 sites of Cct2 were phosphorylated in wild-type yeast cells, whereas these phosphorylation modifications were absent *atg1Δ* yeast cells (Dataset S1). Additionally, previous studies have reported that Atg1 substrates have specific phosphorylation motifs (PMID: 24440502), although we cannot exclude the possibility that Atg1 phosphorylate some substrates with non-classical motifs. However, our *in vitro* kinase and Cct2 S412A-S470A mutant experiments confirm that S412 and S470 are indeed the two primary phosphorylation sites on Cct2 by Atg1 (Figure 4F and G). We have discussed this point in the revised manuscript. Similarly, we purified HA-CCT2 using anti-HA agarose beads in wild-type (WT) and ULK1 KO MEF cell lines in the presence of SOD1 G93A-GFP under nutrient-rich conditions. MS analysis revealed phosphorylation of the S458 site of CCT2 in wild-type MEF cells, whereas S458 is not phosphorylated in ULK1 KO MEF cells (Dataset S2).

To further confirm Atg1-mediated Cct2 phosphorylation is crucial for its binding with Atg8 and Atg1, we explored whether the deletion of *ATG1* regulates the binding of Cct2 with Atg8 and Atg11. Consistent with the results of Cct2 S412A-S417A mutant, *ATG1* KO significantly decrease the association of Cct2 with Atg8 and Atg11 in the presence of Ape1 P22L under nutrient-rich conditions (Figure R16B and C, now Figure 5F and S5D). Subsequently, we employed a ULK1 KO cell line to examine whether ULK1 regulates the interaction between CCT2 and LC3C. Co-IP assays confirmed that ULK1 KO dramatically decreases the binding of CCT2 to LC3C in the presence of SOD1 G93A-GFP under nutrient-rich conditions (Figure R16D and E, now Figure 6J and K). Collectively, these results indicate that Atg1/ULK1-mediated Cct2 phosphorylation is required for

the association of Cct2 with Atg8/LC3 during the process of solid aggregaphy.

Figure R16: Atg1/ULK1-mediated Cct2 phosphorylation is required for the association of Cct2 with Atg8/LC3 during the process of solid aggregaphy. (A) *In vitro* kinase assays were performed using the GST-Cct2 or its variants purified from *E. coli* as substrates with WT or kinase-dead (KD) Atg1-3xFLAG, purified from yeast cells expressing Ape1 P22L under nutrient-rich conditions, served as the protein kinase. Phosphorylation of GST-Cct2 was detected using anti-thioP antibody. The data are representative of two independent experiments. (B) *ape1Δ atg1Δ atg8Δ* cells co-expressing an empty vector (EV), 3×HA-Cct2, 3×HA-Cct2^{S412A-S470A} (2A) with 3×FLAG-Atg8 were grown into log phase under nutrient-rich medium. Cell lysates were immunoprecipitated with anti-FLAG agarose beads and then analyzed by western blot using anti-HA antibody. The data are representative of three independent experiments. (C) *ape1Δ atg1Δ* cells co-expressing 3×HA-Cct2 with Atg11-3×FLAG were grown to log phase in the presence of Ape1 P22L under nutrient-rich medium. Cell lysates were immunoprecipitated with anti-FLAG agarose beads and then

analyzed by western blot using anti-HA antibody. The data are representative of three independent experiments. **(D)** Western blot validation of ULK1 KO effect in MEF cell lines. The data are representative of two independent experiments. **(E)** WT or ULK1 KO MEF cell lines were co-transfected with FLAG-LC3C, HA-CCT2, and SOD1 G93A-GFP. Cell lysates were immunoprecipitated with anti-FLAG agarose beads and then analyzed by western blot using an anti-HA antibody. The data are representative of three independent experiments.

Figure 4C, D: WT-CCT2, the expression levels of the WT-CCT2 are much higher compared to the mutants. Do the authors still detect this defect when the results are normalized against the CCT2 expression levels? The authors analyzing here yeast cells that have been starved for 1-2 days, thereby massively inducing bulk autophagy. How can the authors conclude that the effect is specific for selective autophagy? These experiments need to be repeated under non-starving conditions.

Response: We fully agree with the reviewer's comments. As suggested, we performed the experiment in the presence of GFP-47Q under nutrient-rich conditions. Western blot results showed that the Cct2^{S412A-S470A} mutant significantly decreased the generation of free GFP (Figure R17, now Figure S5A and B), indicating that Atg1-mediated Cct2 phosphorylation is required for the degradation of GFP-poly 47Q under nutrient-rich conditions.

Figure R17: Atg1-mediated Cct2 phosphorylation is required for the cleavage of GFP-47Q under nutrient-rich conditions. (A) Yeast cells co-expressing Cu²⁺-inducible GFP-47Q plasmids with an empty vector (EV), 3×HA-Cct2, or CCT2^{S412A-S470A}

^{S470A} (2A) were grown to an OD₆₀₀=0.6. Subsequently, 0.1 mM CuSO₄ was added to the cells to induce GFP-47Q to form solid aggregates for 12 h. Samples were analyzed by immunoblot for the cleavage of GFP-47Q. Pgk1 served as a loading control. **(B)** Quantification of GFP to GFP-47Q+GFP ratio from (A). Data are presented as means ± SD(n=3). ***p < 0.001; **p < 0.01; two-tailed Student's t tests were used.

Figure 4E, S4C: The authors must not talk about solid aggrephagy when the experiment was performed during starvation conditions. This is misleading. Moreover, in the experiments described in Figure 4, there is even no solid aggrephagy cargo present showing that the interaction/phosphorylation occurs independently of the substrate. Therefore, to claim that the Atg1 mediated CCT2 phosphorylation is solid aggrephagy specific, the authors need to show the selectivity only in the presence of a substrate and/or re-phrase their conclusion.

Response : We completely agree with the reviewer's comments. As suggested, we performed Co-IP assays for the original Figure 4E and S4C in the presence of Ape1-P22L under nutrient-rich conditions. The results indicate that Atg1-mediated Cct2 phosphorylation is required for the association of Cct2 with Atg8 and Atg11 under nutrient-rich conditions (Figure R18, and Figure 5E and S5C).

Figure R18: Atg1-mediated Cct2 phosphorylation is required for the association of Cct2 with Atg8 and Atg11. (A) *atg8Δ ape1Δ* cells co-expressing 3×HA-Cct2 or

3×HA-Cct2^{S412A-S470A} (2A) with FLAG-Atg8 in the presence of Ape1 P22L were grown to the log phase under nutrient-rich conditions. Cell lysates were immunoprecipitated with anti-FLAG agarose beads and then analyzed by western blot using anti-HA antibody. The data are representative of three independent experiments. (B) *ape1*Δ cells co-expressing 3×HA-Cct2 or 3×HA-Cct2^{S412A-S470A} (2A) with Atg11-3×FLAG in the presence of Ape1 P22L were grown to the log phase under nutrient-rich conditions. Cell lysates were immunoprecipitated with anti-FLAG agarose beads and then analyzed by western blot using anti-HA antibody. The data are representative of three independent experiments.

Figure 5D-G: For all the solid aggregating substrates (Q47, FUS, alpha-syn, ...in yeast and HEK) the authors did not show that they analyse the aggregated (non-soluble) form of the mutant proteins. The protocol in the methods section does also not allow to judge whether they examine the insoluble or soluble fraction. The authors must clearly show that CCT2 has no effect on the WT protein versions of their mutated counterparts. Furthermore, if the authors cannot clearly show that aggregated species are analysed, then the phrasing solid aggregating needs to be changed.

Response: We appreciate the reviewer's suggestions and apologize for not detailing how we detect the insoluble fraction of SOD1 G93A-GFP, GFP-FUS P525L, and GFP- α -Synuclein A53T in mammalian cells. In our revised Materials and Methods section, we have provided a detailed method for examining their insoluble fractions. As suggested, we investigated whether the degradation of their corresponding wild-type proteins depends on CCT2. Western blotting data confirm that the degradation of these wild-type proteins is not dependent on CCT2 (Figure R19A-F, now Figure S6D-G).

In yeast cells, reverting Ape1 P22L to its wild-type form Ape1 revealed CCT2-independent maturation (Figure R19G and H, now Figure 1A and B), whereas experiments for GFP-47Q were not performed due to its inability to revert to the

wild-type form. Collectively, these findings support our conclusion that CCT2 is a receptor for solid aggregophagy.

Figure R19: CCT2 is a receptor for solid aggregophagy. (A, C, E) Turnover of SOD1-GFP(A), GFP- α -Synuclein(C), or GFP-Fus (E) in CHX chase assay with empty vector (EV), HA-CCT2, HA-CCT2^{S458A}, or HA-CCT2-mVL(I)L (LIR motif mutant) expression in HeLa cell lines at 24 h (A and C) or 72 h (E) after transfection. The protein sample were analyzed by western-blot. GAPDH served as a loading control. (B, D, F) Quantification of normalized SOD1-GFP from (A), GFP- α -Synuclein (C), and GFP-Fus (E). Data are presented as means \pm SD(n=3). NS, not significant. Two-tailed Student's t tests were used. (G) *ape1Δ atg5Δ* or *ape1Δ* yeast cells co-expressing Ape1 WT or Ape1 P22L with empty vector (EV), 3 \times HA-Cct1, 3 \times HA-Cct2, or 3 \times HA-Cct4 were grown to log phase under nutrient-rich medium. Samples were analyzed by immunoblot for detecting the maturation of PrApe1 into

mApe1. Pgk1 served as a loading control. **(H)** Quantification of mApe1 to PrApe1+mApe1 ratio from (G). Data are presented as means \pm SD(n=3). ***p < 0.001; NS, not significant; two-tailed Student's t tests were used.

Many interaction experiments between CCT2 and the autophagy machinery proteins were done in the absence of solid aggrephagy cargos or during starvation conditions. Can the authors exclude that CCT2 is a substrate of autophagy and how are these interactions then to be explained?

Response: We appreciate the reviewer's insightful suggestions, which have helped us avoid inappropriate conclusions or over-interpretations. As suggested, we performed co-IP experiments between Cct2 and the autophagy machinery proteins in the presence of solid aggrephagy cargos under nutrient-rich conditions. Based on the findings presented in the revised Figure, we concluded that Atg1-mediated Cct2 phosphorylation and the Cct2-Atg11 interaction are crucial for solid aggrephagy by regulating the binding of Cct2 to Atg8.

Quantification for all IP results are missing and if not done, repeats of these experiments needs to be provided.

Response: We thank the reviewer's comment. As suggested, we quantified the IP results or provided the number of repetitions for IP experiments in the revised manuscript.

Minor points:

The dataset tables should be better explained.

Response: We thank the reviewer's comment. As suggested, we have modified the explanation of the dataset tables in the revised manuscript.

In yeast, the authors show that Atg1 phosphorylation of Cct2 regulates the binding of Cct2 and Atg11 and Atg11 then binds Atg8. Can the authors speculate whether in mammalian cells such an adaptor protein that links CCT2 and LC3 does also exist (e.g., FIP200) or is the LC3-CCT2 interaction direct?

Response: As suggested, we investigated whether the mammalian homolog of Atg11, FIP200, can bind to CCT2 and regulate its association with LC3C. Co-IP assays revealed that FIP200 can indeed bind with CCT2 in the presence of SOD1-G93A-GFP under nutrient-rich conditions (Figure R12A, now Figure 6M), and FIP200 KO significantly decreases the interaction between CCT2 and LC3C (Figure R12B-D, now Figure 6N-P). Additionally, we found that WIPI2 KO does not affect the binding of LC3C to CCT2 (Figure R12E-G, now Figure S6I-K). Taken together, these findings suggest that the regulatory role of Atg11 in the binding of CCT2 to LC3C is conserved in mammals.

Figure R12: FIP200 regulates the binding of CCT2-LC3C. (A) HEK293T cell lines were co-transfected with FLAG-CCT2, HA-FIP200 and SOD1 G93A-GFP. Cell

lysates were immunoprecipitated with anti-FLAG agarose beads and then analyzed by western blot using an anti-HA antibody. The data are representative of three independent experiments. **(B)** Western blot validation of FIP200 KO effect in MEF cell lines. The data are representative of two independent experiments. **(C)** WT or FIP200 KO MEF cell lines were co-transfected with FLAG-LC3C, HA-CCT2 and SOD1 G93A-GFP. Cell lysates were immunoprecipitated with anti-FLAG agarose beads and then analyzed by western blot using an anti-HA antibody. **(D)** Quantification of IP/Lysates HA-CCT2 from (C). Data are presented as means \pm SD (n=3). ***p < 0.001; two-tailed Student's t tests were used. **(E)** Western blot validation of WIPI2 KO effect in MEF cell lines. The data are representative of two independent experiments. **(F)** WT or WIPI2 KO MEF cell lines were co-transfected with FLAG-LC3C, HA-CCT2 and SOD1 G93A-GFP. Cell lysates were immunoprecipitated with anti-FLAG agarose beads and then analyzed by western blot using an anti-HA antibody. **(G)** Quantification of IP/lysates HA-CCT2 from (F). Data are presented as means \pm SD (n=3). NS, not significant; two-tailed Student's t tests were used.

Overall, we thank the reviewers for these insightful and constructive comments, which have helped us strengthen the rigor of our study and clarified the conclusions of our manuscript. We hope that, following their guidance, our paper is sufficiently improved to meet the appropriately high standards necessary for publication in *EMBO reports*.

Best wishes,

Cong

Dear Prof. Yi

Thank you for the submission of your revised manuscript to EMBO reports. We have now received the full set of referee reports that is copied below.

As you will see, all referees are very positive about the study and request only minor textual changes regarding the conclusiveness of 'solid aggrephagy' in the mammalian system.

Your manuscript was a re-submission and you had therefore not received our general information on formatting. I am now pasting this information below. Please re-format your manuscript along these general guidelines. You will also be contacted by our Source Data Coordinator, Hannah Sonntag, with a checklist detailing which figure panels we would need source data for and useful tips how to structure it. See also point 8 below.

A few specific points I would like to emphasize:

- You need to define the exact p-values in the figure legends not the range.
- The Supplemental Data PDF you supplied should be reformatted as Appendix. The nomenclature is Appendix Figure S# and Appendix Table S#. It needs a table of content with page numbers.
- Datasets are called Dataset EV#. They need a legend in a separate tab of the .xls file which should also state the file name, i.e., Dataset EV#.
- We need a Reagents and Tools Table (point 12).

General formatting guidelines

2) individual production quality figure files as .eps, .tif, .jpg (one file per figure).

Please download our Figure Preparation Guidelines (figure preparation pdf) from our Author Guidelines pages <https://www.embopress.org/page/journal/14693178/authorguide> for more info on how to prepare your figures.

4) a complete author checklist, which you can download from our author guidelines

(<https://www.embopress.org/page/journal/14693178/authorguide>). Please insert information in the checklist that is also reflected in the manuscript. The completed author checklist will also be part of the RPF.

5) Please note that all corresponding authors are required to supply an ORCID ID for their name upon submission of a revised manuscript (<https://orcid.org/>). Please find instructions on how to link your ORCID ID to your account in our manuscript tracking system in our Author guidelines

(<https://www.embopress.org/page/journal/14693178/authorguide#authorshipguidelines>)

6) We replaced Supplementary Information with Expanded View (EV) Figures and Tables that are collapsible/expandable online. A maximum of 5 EV Figures can be typeset. EV Figures should be cited as 'Figure EV1, Figure EV2' etc... in the text and their respective legends should be included in the main text after the legends of regular figures.

7) Please note that a Data Availability section at the end of Materials and Methods is now mandatory. In case you have no data that requires deposition in a public database, please state so instead of refereeing to the database.

See also < <https://www.embopress.org/page/journal/14693178/authorguide#dataavailability>>. Please note that the Data

Availability Section is restricted to new primary data that are part of this study.

Additional information on source data and instruction on how to label the files are available
<<https://www.embopress.org/page/journal/14693178/authorguide#sourcedata>>.

10) Figure legends and data quantification:
The following points must be specified in each figure legend:

- the name of the statistical test used to generate error bars and P values,
 - the number (n) of independent experiments (please specify technical or biological replicates) underlying each data point,
 - the nature of the bars and error bars (s.d., s.e.m.)
- If the data are obtained from n {less than or equal to} 5, show the individual data points in addition to the SD or SEM.
 - If the data are obtained from n {less than or equal to} 2, use scatter blots showing the individual data points.

See also the guidelines for figure legend preparation:
<https://www.embopress.org/page/journal/14693178/authorguide#figureformat>

11) Our journal encourages inclusion of *data citations in the reference list* to directly cite datasets that were re-used and obtained from public databases. Data citations in the article text are distinct from normal bibliographical citations and should directly link to the database records from which the data can be accessed. In the main text, data citations are formatted as follows: "Data ref: Smith et al, 2001" or "Data ref: NCBI Sequence Read Archive PRJNA342805, 2017". In the Reference list, data citations must be labeled with "[DATASET]". A data reference must provide the database name, accession number/identifiers and a resolvable link to the landing page from which the data can be accessed at the end of the reference. Further instructions are available at <<https://www.embopress.org/page/journal/14693178/authorguide#referencesformat>>.

12) All Materials and Methods need to be described in the main text using our 'Structured Methods' format, which is required for all research articles. According to this format, the Methods section includes a Reagents and Tools Table (listing key reagents, experimental models, software and relevant equipment and including their sources and relevant identifiers) followed by a Methods and Protocols section describing the methods using a step-by-step protocol format. The aim is to facilitate adoption of the methodologies across labs. More information on how to adhere to this format as well as a downloadable template (.docx) for the Reagents and Tools Table can be found in our author guidelines:
<https://www.embopress.org/page/journal/14693178/authorguide#structuredmethods>.

An example of a Method paper with Structured Methods can be found here:
<https://www.embopress.org/doi/10.15252/msb.20178071>.

13) As part of the EMBO publication's Transparent Editorial Process, EMBO Reports publishes online a Review Process File to accompany accepted manuscripts. This File will be published in conjunction with your paper and will include the referee reports, your point-by-point response and all pertinent correspondence relating to the manuscript.

Kind regards,

Martina

Referee #1:

The authors have substantially revised the manuscript and added convincing data to support their claims. All my points have been addressed.

Referee #2:

The authors have satisfactorily addressed all the concerns I raised in the review of the original manuscript.

Referee #3:

The authors addressed all my concerns satisfactorily, except for one point.

One of my comments regarding former Fig. 5D-G was: "Furthermore, if the authors cannot clearly show that aggregated species are analysed, then the phrasing solid aggrephagy needs to be changed."

The authors now specified the protocol in the M&M section. This protocol describes that the cells were treated with digitonin before lysis and the assumption is that this treatment would lead to a loss of all soluble species. But that is not necessarily the case, therefore, for the mammalian part, there is no proof that the disease related protein degradation, which is observed here, is concerning already aggregated (solid) species. In my opinion, this is an over-interpretation of the data on the mammalian part, and therefore the wording "solid aggrephagy" should be changed into "aggrephagy" for this part (e.g., p2, line 14; p5, line 5; p14, line 8 and 12).

Page 14, line 7, 11: the figures citation is not updated (it should be figure 6 now).

Dear Martina,

Thank you very much for giving us the opportunity to resubmit. Please find below our point-by-point responses to Reviewer 3's comments. Thank you once again for allowing us to address these issues.

Point-by-point responses to the reviewer:

Referee #3:

The authors addressed all my concerns satisfactorily, except for one point.

One of my comments regarding former Fig. 5D-G was: "Furthermore, if the authors cannot clearly show that aggregated species are analysed, then the phrasing solid aggrephagy needs to be changed."

The authors now specified the protocol in the M&M section. This protocol describes that the cells were treated with digitonin before lysis and the assumption is that this treatment would lead to a loss of all soluble species. But that is not necessarily the case, therefore, for the mammalian part, there is no proof that the disease related protein degradation, which is observed here, is concerning already aggregated (solid) species. In my opinion, this is an over-interpretation of the data on the mammalian part, and therefore the wording "solid aggrephagy" should be changed into "aggrephagy" for this part (e.g., p2, line 14; p5, line 5; p14, line 8 and 12).

Response: We thank the reviewer's comments. As suggested, we have revised "solid aggrephagy" to "aggrephagy" in the mammalian system.

Page 14, line 7, 11: the figures citation is not updated (it should be figure 6 now).

Response: We thank the reviewer for pointing it out. It has now been corrected.

We thank the reviewer for the insightful and constructive comments, which have helped us to strengthen the rigor of our study in this manuscript. Furthermore, we have adjusted the manuscript according to the formatting requirements of *EMBO reports*. We hope that following their guidance, our paper is sufficiently improved to meet the appropriately high standards necessary for publication in *EMBO reports*.

Best wishes,

Cong

Dear Cong,

Thank you for the submission of the revised and re-formatted manuscript files. Our editorial assistants and I have done some final checks on them and I note below a few points that still need your attention, before we can proceed with the official acceptance of your manuscript.

- Please provide an ORCID ID for Dr. Du Feng. All corresponding authors must provide this information. Please find instructions on how to link your ORCID ID to your account in our manuscript tracking system in our Author guidelines (<https://www.embopress.org/page/journal/14693178/authorguide#authorshipguidelines>)
- Please provide up to 5 keywords.
- Please provide callouts for Fig. 6D-G. There is a reference to Fig. 5D-G on page 14 that might be meant to refer to Fig. 6D-G? Please check.
- The legends for the EV figures should be included in the manuscript file, below the main figure legends.
- Please provide the Appendix as PDF file. The .docx file is not needed in this case.
- We need the Reagents and Tools table as separate .docx file that is uploaded to the system as file type "Reagent table". Please also remove the table from the manuscript file.
- The Data Availability statement should only refer to datasets deposited (or not) in public repositories. Therefore, please remove the statement "The related yeast strains, mammalian cell lines, and plasmids will be made available upon reasonable request." from that section.
- LC-MS data: Is it possible to deposit the mass spectrometry data in a public repository in addition to listing the results in the EV Datasets? Moreover, I could not find a description of the LC-MS experiments in the methods section, unless I missed it?
- Please include a statement on blinding in the methods section, as indicated in the Author Checklist.
- Source data:
 - 1) The following Source data appears to be missing: Fig. 1C, 3GHI. Please provide these.
 - 2) The Source Data folder 'Figure 3 - Western blots' seems to contain a duplication of the Western blots for Figure 2, which explains the missing data listed above for Fig. 3G, H, I.
 - 3) The Source data should be re-grouped so that each folder contains one folder per panel; SD for the EV and Appendix figures need to be grouped into one folder and zipped.
 - 4) We perform a routine screen on all numerical source data supplied. Here I noticed that the FRAP quantification for the last two timepoints (Figure EV2B and D) appear to be the same. Was the signal saturated/stable? Please check these quantifications (the seemingly duplicate rows are color-coded in the attached files).
- Our production/data editors have asked you to clarify several points in the figure legends (see below). Please incorporate these changes in the manuscript and return the revised file with tracked changes with your final manuscript submission.
 - A) Figure legend text:
 - Please note that the legends for figures 6e-f is not provided in the sequential manner (legend for figure 6f is provided before legend of figure 6e). This needs to be rectified.
 - Please note that the legends for figures EV 5e-f is not provided in the sequential manner (legend for figure EV 5f is provided before legend of figure EV 5e). This needs to be rectified.
 - B) Replicates and error bars:
 - Although 'n' is provided, please describe the nature of entity for 'n' in the legends of figures 1b, e, i; 6e, g, i, l, p; EV 2f; EV 3b, d; EV 4b; EV 5e, g, k.
 - Please note that the red arrowheads are not defined in the legend of figure 2e. This needs to be rectified.
- Dataset EV3 lists the strains used. This could be changed to Table EV1 instead, as it is not that complex as a dataset.
- Please remove the Synopsis text from the manuscript file.
- As a standard procedure, we edit the title and abstract of manuscripts to make them more accessible to a general readership. Please find my suggested abstract below my signature for your approval or further editing, as needed.
- On a different note, I would like to alert you that EMBO Press offers a new format for a video-synopsis of work published with us, which essentially is a short, author-generated film explaining the core findings in hand drawings, and, as we believe, can be

very useful to increase visibility of the work. This has proven to offer a nice opportunity for exposure i.p. for the first author(s) of the study. Please see the following link for representative examples and their integration into the article web page:

<https://www.embopress.org/doi/full/10.15252/emj.2019103932>

Kind regards,

Martina

Abstract

CCT2 serves as an aggrephagy receptor that plays a crucial role in the clearance of solid aggregates, yet the underlying molecular mechanisms by which CCT2 regulates solid aggrephagy are not fully understood. Here we report that the binding of Cct2 to Atg8 is governed by two distinct regulatory mechanisms: Atg1-mediated Cct2 phosphorylation and the interaction between Cct2 and Atg11. Atg1 phosphorylates Cct2 at Ser412 and Ser470, and disruption of these phosphorylation sites impairs solid aggrephagy by hindering Cct2-Atg8 binding. Additionally, we observe that Atg11, an adaptor protein involved in selective autophagy, directly associates with Cct2 through its CC4 domain. Deficiency in this interaction significantly weakens the association of Cct2 with Atg8. The requirement of Atg1-mediated Cct2 phosphorylation and of Atg11 for CCT2-LC3C binding and subsequent aggrephagy is conserved in mammalian cells. These findings provide insights into the crucial roles of Atg1-mediated Cct2 phosphorylation and Atg11-Cct2 binding as key mediators governing the interaction between Cct2 and Atg8 during the process of solid aggrephagy.

Dear Martina,

Please find below our point-by-point responses to the editor's comments. Thank you once again for allowing us to address these issues.

Point-by-point responses to the editor:

- Please provide an ORCID ID for Dr. Du Feng. All corresponding authors must provide this information. Please find instructions on how to link your ORCID ID to your account in our manuscript tracking system in our Author guidelines (<<https://www.embopress.org/page/journal/14693178/authorguide#authorshipguidelines>>);

Response: Dr. Du Feng's ORCID ID is 0000-0002-2489-4702.

- Please provide up to 5 keywords.

Response: We have provided 5 keywords.

- Please provide callouts for Fig. 6D-G. There is a reference to Fig. 5D-G on page 14 that might be meant to refer to Fig. 6D-G? Please check.

Response: We thank the editor for pointing this out. We have now revised it to Fig. 6D–G.

- The legends for the EV figures should be included in the manuscript file, below the

main figure legends.

Response: We have added the EV figure legends below the main figure legends.

- Please provide the Appendix as PDF file. The .docx file is not needed in this case.

Response: We have provided a PDF file of the Appendix.

- We need the Reagents and Tools table as separate .docx file that is uploaded to the system as file type "Reagent table". Please also remove the table from the manuscript file.

Response: We have now addressed this issue as requested.

- The Data Availability statement should only refer to datasets deposited (or not) in public repositories. Therefore, please remove the statement "The related yeast strains, mammalian cell lines, and plasmids will be made available upon reasonable request." from that section.

Response: We have now figured out this issue as requested.

- LC-MS data: Is it possible to deposit the mass spectrometry data in a public repository in addition to listing the results in the EV Datasets? Moreover, I could not find a description of the LC-MS experiments in the methods section, unless I missed it?

Response : We have now included a description of the LC-MS experiments in

the revised methods section. Additionally, we do not intend to deposit the mass spectrometry data in a public repository.

- Please include a statement on blinding in the methods section, as indicated in the Author Checklist.

Response: We have included this statement in the revised methods section.

- Source data:

1) The following Source data appears to be missing: Fig. 1C, 3GHI. Please provide these.

Response: We have now provided the source data for Figs. 1C and 3G-I.

2) The Source Data folder "Figure 3 - Western blots" seems to contain a duplication of the Western blots for Figure 2, which explains the missing data listed above for Fig. 3G, H, I.

Response: We thank the editor for pointing it out, and we have now addressed it as requested.

3) The Source data should be re-grouped so that each folder contains one folder per panel; SD for the EV and Appendix figures need to be grouped into one folder and zipped.

Response: We have now figured out this issue as requested.

4) We perform a routine screen on all numerical source data supplied. Here I noticed that the FRAP quantification for the last two timepoints (Figure EV2B and D) appear to be the same. Was the signal saturated/stable? Please check these quantifications (the seemingly duplicate rows are color-coded in the attached files).

Response: We thank the editor for pointing these out. After reviewing the original data and statistical analysis, we found that the imaging duration for all experiments was 190 s after quenching. However, due to equal spacing on the x-axis during graphing, the final time point was shown as 200 s. This led to duplicate statistical results for the 190 s and 200 s time points. We have now removed the incorrect 200 s data point.

For the other time points with repeated results, we examined the original fluorescence intensity data and the statistical outcomes at different time points. We found that the statistical results were normalized based on the comparison of fluorescence intensities at each time point with the intensities before quenching. The occasional repeated results for GFP-47Q and RFP-Ape1^{P22L} at other time points were due to rounding the fluorescence intensity values to two decimal places during analysis. To ensure the reliability of the results, we have re-analyzed the original data (now Figure EV3B and 3D).

- Our production/data editors have asked you to clarify several points in the figure

legends (see below). Please incorporate these changes in the manuscript and return the revised file with tracked changes with your final manuscript submission.

A) Figure legend text:

- Please note that the legends for figures 6e-f is not provided in the sequential manner (legend for figure 6f is provided before legend of figure 6e). This needs to be rectified.

Response: We thank the reviewer for pointing this out. Figures 6D and 6F both display Western blot images, and their figure legends were initially combined. Similarly, the quantitative analyses for Figures 6E and 6G correspond to Figures 6D and 6F, respectively, which resulted in the legend for Figure 6F appearing before the legend for Figure 6E. We have now separated the legends and arranged them in the correct sequential order.

- Please note that the legends for figures EV 5e-f is not provided in the sequential manner (legend for figure EV 5f is provided before legend of figure EV 5e). This needs to be rectified.

Response: We thank the reviewer for pointing this out. Figures EV5D and EV5F both present Western blot images, and their figure legends were initially combined. Similarly, the quantitative analyses for Figures EV5E and EV5G correspond to Figures EV5D and EV5F, respectively, which caused the legend for Figure EV5F to appear before the legend for Figure EV5E. We have now separated the legends and arranged them in the correct sequential order.

Replicates and error bars: m- Although 'n' is provided, please describe the nature of entity for 'n' in the legends of figures 1b, e, i; 6e, g, i, l, p; EV 2f; EV 3b, d; EV 4b; EV 5e, g, k.

Response: As requested, we have revised "n=3" to "Data represent the results of three independent experiments" in the updated figure legends.

- Please note that the red arrowheads are not defined in the legend of figure 2e. This needs to be rectified.

Response : We have clarified the meaning of the red arrows in the figure 2e legend.

- Dataset EV3 lists the strains used. This could be changed to Table EV1 instead, as it is not that complex as a dataset.

Response: We have revised Dataset EV3 to Table EV1.

- Please remove the Synopsis text from the manuscript file.

Response : We have removed the Synopsis text from the manuscript file and have provided it as a separate Word file.

- As a standard procedure, we edit the title and abstract of manuscripts to make them more accessible to a general readership. Please find my suggested abstract below my

signature for your approval or further editing, as needed.

Response: We greatly appreciate the editor's revisions to our abstract, which have indeed made it much more accessible to a general readership.

- On a different note, I would like to alert you that EMBO Press offers a new format for a video-synopsis of work published with us, which essentially is a short, author-generated film explaining the core findings in hand drawings, and, as we believe, can be very useful to increase visibility of the work. This has proven to offer a nice opportunity for exposure i.p. for the first author(s) of the study. Please see the following link for representative examples and their integration into the article web page:

<https://www.embopress.org/doi/full/10.15252/emj.2019103932>

Response: We thank the editor for the suggestion. However, we currently do not plan to provide a video synopsis.

Best wishes,

Cong

Prof. Cong Yi
Zhejiang University
866 Yuhangtang Road
Hangzhou, orcid||| 310058
China

Dear Cong,

Thank you for implementing the final minor corrections. I am very pleased to accept your manuscript for publication in the next available issue of EMBO reports. Thank you for your contribution to our journal.

Kind regards,

Martina
